# Regularized Gradient Descent Ascent for Two-Player Zero-Sum Markov Games

**Sihan Zeng**
Dept. of Electrical and Computer Engineering
Georgia Institute of Technology
Atlanta, GA 30318
szeng30@gatech.edu

**Thinh Doan**
Dept. of Electrical and Computer Engineering
Virginia Tech
Blacksburg, VA 24061
thinhdoan@vt.edu

**Justin Romberg**
Dept. of Electrical and Computer Engineering
Georgia Institute of Technology
Atlanta, GA 30318
jrom@ece.gatech.edu

## Abstract

We study the problem of finding the Nash equilibrium in a two-player zero-sum Markov game. Due to its formulation as a minimax optimization program, a natural approach to solve the problem is to perform gradient descent/ascent with respect to each player in an alternating fashion. However, due to the non-convexity/non-concavity of the underlying objective function, theoretical understandings of this method are limited. In our paper, we consider solving an entropy-regularized variant of the Markov game. The regularization introduces structure into the optimization landscape that make the solutions more identifiable and allow the problem to be solved more efficiently. Our main contribution is to show that under proper choices of the regularization parameter, the gradient descent ascent algorithm converges to the Nash equilibrium of the original unregularized problem. We explicitly characterize the finite-time performance of the last iterate of our algorithm, which vastly improves over the existing convergence bound of the gradient descent ascent algorithm without regularization. Finally, we complement the analysis with numerical simulations that illustrate the accelerated convergence of the algorithm.

## 1 Introduction

The two-player zero-sum Markov game is a special case of competitive multi-agent reinforcement learning where two agents driven by opposite reward functions jointly determine the state transition in an environment. Usually cast as a non-convex non-concave minimax optimization program, this framework finds applications in many practical problems including game playing [Lanctot et al., 2019, Vinyals et al., 2019], robotics [Riedmiller and Gabel, 2007, Shalev-Shwartz et al., 2016], and robust policy optimization [Pinto et al., 2017].

A convenient class of methods frequently used to solve multi-agent reinforcement learning problems is the independent learning approach. Independent learning algorithms proceed iteratively with each player taking turns to optimize its own objective while pretending that the other players' policies are fixed to their current iterates. In the context of two-player zero-sum Markov games, the independent learning algorithm performs gradient descent ascent (GDA), which alternates between the gradient updates of the two agents that seek to maximize and minimize the same value function. Despite

36th Conference on Neural Information Processing Systems (NeurIPS 2022).

the popularity of such algorithms in practice, their theoretical understandings are sparse and do not follow from those in the single-agent case as the environment is not stationary from the eye of any agent. [Daskalakis et al., 2017] shows that iterates of GDA can possibly diverge or be trapped in limit cycles even in the simplest single-state case when the two players learn with the same rate.

It may be tempting to analyze the two-player zero-sum Markov game by applying the existing theoretical results on minimax optimization. However, as the objective function in a Markov game is not convex or concave, current analytical tools in minimax optimization that require the objective function to be convex/concave at least on one side are inapplicable. Fortunately, the Markov game has its own structure: it exhibits a "gradient domination" condition with respect to each player, which essentially guarantees that every stationary point of the value function is globally optimal. Exploiting this property, Daskalakis et al. [2020] builds on the theory of Lin et al. [2020a] and shows that a two-time-scale GDA algorithm converges to the Nash equilibrium of the Markov game with a complexity that depends polynomially on the specified precision. However, deriving an explicit finite-time convergence rate is still an open problem. In addition, the analysis in Daskalakis et al. [2020] does not guarantee the convergence of the last iterate; convergence is shown on the average of all past iterates.

In this paper, we show that introducing an entropy regularizer into the value function significantly accelerates the convergence of GDA to the Nash equilibrium. By dynamicially adjusting the regularization weight towards zero, we are able to give a finite-time last-iterate convergence guarantee to the Nash equilibrium of the original Markov game.

**Main Contributions**

• We show that the entropy-regularized Markov game is highly structured; in particular, it obeys a condition similar to the well-known Polyak-Łojasiewicz condition, which allows linear convergence of GDA to the (unique) equilibrium point of the regularized game with fixed regularization weight. We also show that the distance of the equilibrium point of the regularized game to the equilibrium point of the original game can be bounded in terms of the regularizing weight.

• We show that by dynamically driving the regularization weight towards zero, we can solve the original Markov game. We propose two approaches to reduce the regularization weight and study their finite-time convergence. The first approach uses a piecewise constant weight that decays geometrically fast, and its analysis follows as a straightforward consequence of our analysis for the case of fixed regularization weight. To reach a Nash equilibrium of the Markov game up to error $\epsilon$, we find that this approach requires at most $\mathcal{O}(\epsilon^{-3})$ gradient updates, where $\mathcal{O}$ only hides structural constants. The second approach reduces the regularization weight online along with the gradient updates. Through a multi-time-scale analysis, we optimize the regularization weight sequence along with the step size as polynomial functions of $k$, where $k$ is the iteration index. We show that the last iterate of the GDA algorithm converges to the Nash equilibrium of the original Markov game at a rate of $\mathcal{O}(k^{-1/3})$. Compared with the state-of-the-art analysis of the GDA algorithm without regularization which shows that the convergence rate of the averaged iterates is polynomial in the desired precision and all related parameters, our algorithms enjoy faster last-iterate convergence guarantees.

## 1.1 Related Work

A Markov game reduces to a standard Markov Decision Process (MDP) with respect to one player if the policy of the other player is fixed. This is an important observation that allows our work to exploit the recent advances in the analysis of policy gradient methods for MDPs [Nachum et al., 2017, Neu et al., 2017, Agarwal et al., 2020, Mei et al., 2020, Lan, 2022]. Various entropy-based regularizers are introduced in these works that inspire the regularization of this paper. Our particular regularization is also considered by Cen et al. [2021], but we discuss and leverage structure in the regularized Markov game that was previously unknown.

As the two-player zero-sum Markov game can be formulated a minimax optimization problem, our work relates to the vast volume of literature in this domain. Minimax optimization has been extensively studied in the case where the objective function is convex/concave with respect to at least one variable [Lin et al., 2020a,b, Wang and Li, 2020, Ostrovskii et al., 2021]. In the general non-convex non-concave setting, the problem becomes much more challenging as even the notion of stationarity is unclear [Jin et al., 2020]. In Nouiehed et al. [2019], non-convex non-concave

objective functions obeying a one–sided PŁ condition are considered, which the authors use to show the convergence of GDA. Yang et al. [2020] analyzes GDA under a two-sided PŁ condition and has a tight connection to our work as the value function of our regularized Markov game also has structure that is similar to, but weaker than, the PŁ condition on two sides.

By exploiting the gradient domination condition of a Markov game with respect to each player, Daskalakis et al. [2020] is the first to show that the GDA algorithm provably converges to a Nash equilibrium of a Markov game. A finite-time complexity is not derived in Daskalakis et al. [2020], but their analysis and choice of step sizes indicate that the convergence rate is at least worse than $\mathcal{O}(k^{-1/10.5})$. Additionally, Daskalakis et al. [2020] does not guarantee the convergence of the last iterate, but rather analyzes the average of all iterates. In contrast, our work provides a finite-time convergence analysis on the last iterate of the GDA algorithm.

While our work treats the Markov game purely from the optimization perspective, we would like to point out another related line of works that consider value-based methods [Perolat et al., 2015, Bai and Jin, 2020, Xie et al., 2020, Cen et al., 2021, Sayin et al., 2022]. In particular, Perolat et al. [2015] is among the first works to extend value-based methods from single-agent MDP to two-player Markov games. Since then, the basic techniques for analyzing value-based methods for Markov games are relatively well-known. Bai and Jin [2020] considers a value iteration algorithm with confidence bounds. In Cen et al. [2021], a nested-loop algorithm is designed where the outer loop employs value iteration and the inner loop runs a gradient-descent-ascent-flavored algorithm to solve a regularized bimatrix game. In comparison, pure policy optimization algorithms are much less understood for Markov games, but this is an important subject to study due to their wide use in practice. In single-agent MDPs, value-based methods and policy optimization methods enjoy comparable convergence guarantees today, and our work aims to narrow the gap between the understanding of these two classes of algorithms in two-player Markov games.

Finally, we note the recent surge of interest in solving two-player games and minimax optimization programs with extragradient or optimistic gradient methods in the cases where vanilla gradient algorithms often cannot be shown to converge [Chavdarova et al., 2019, Mokhtari et al., 2020, Li et al., 2022, Wei et al., 2021, Zhao et al., 2021, Cen et al., 2021, Chen et al., 2021]. These methods typically require multiple gradient evaluations at each iteration and are more complicated to implement. Most related to our work, Cen et al. [2021] shows the linear convergence of an extragradient algorithm for solving regularized bilinear matrix games. They also show that a regularized Markov game can be decomposed into a series of regularized matrix games and present a nested-loop extragradient algorithm which solves these games successively and eventually converges to the Nash equilibrium of the regularized Markov game. The regularization weight can then be selected based on the desired precision of the unregularized problem. Although our overall goal of finding the Nash equilibrium of a general Markov game is the same, the manner in which we decompose and analyze the problem is different. Our analysis here is based on GDA applied directly to a general regularized Markov game. We show that for a fixed regularization parameter for a general Markov game, GDA has linear convergence to the modified equilibrium point. We also give a scheduling scheme for adjusting the regularization parameter as the GDA iterations proceed, making them converge to the solution to the original problem.

## 2   Preliminaries

We consider a two-player Markov game characterized by $\mathcal{M} = (\mathcal{S}, \mathcal{A}, \mathcal{B}, \mathcal{P}, \gamma, r)$. Here, $\mathcal{S}$ is the finite state space, $\mathcal{A}$ and $\mathcal{B}$ are the finite action spaces of the two players, $\gamma \in (0, 1)$ is the discount factor, and $r : \mathcal{S} \times \mathcal{A} \times \mathcal{B} \to [0, 1]$ is the reward function. Let $\Delta_{\mathcal{F}}$ denote the probability simplex over a set $\mathcal{F}$, and $\mathcal{P} : \mathcal{S} \times \mathcal{A} \times \mathcal{B} \to \Delta_{\mathcal{S}}$ be the transition probability kernel, with $\mathcal{P}(s' \mid s, a, b)$ specifying the probability of the game transitioning from state $s$ to $s'$ when the first player selects action $a \in \mathcal{A}$ and the second player selects $b \in \mathcal{B}$. The policies of the two players are denoted by $\pi \in \Delta_{\mathcal{A}}^{\mathcal{S}}$ and $\phi \in \Delta_{\mathcal{B}}^{\mathcal{S}}$, with $\pi(a \mid s), \phi(b \mid s)$ denoting the probability of selecting action $a, b$ in state $s$ according to $\pi, \phi$. Given a policy pair $(\pi, \phi)$, we measure its performance in state $s \in \mathcal{S}$ by the value function

$$V^{\pi,\phi}(s) = \mathbb{E}_{a_k \sim \pi(\cdot|s_k), b_k \sim \phi(\cdot|s_k), s_{k+1} \sim \mathcal{P}(\cdot|s_k, a_k, b_k)} \left[ \sum_{k=0}^{\infty} \gamma^k r\left(s_k, a_k, b_k\right) \mid s_0 = s \right].$$

Under a fixed initial distribution $\rho \in \Delta_{\mathcal{S}}$, we define the discounted cumulative reward under $(\pi, \phi)$

$$J(\pi, \phi) \triangleq \mathbb{E}_{s_0 \sim \rho}[V^{\pi,\phi}(s_0)],$$

where the dependence on $\rho$ is dropped for simplicity. It is known that the Nash equilibrium always exists in two-player zero-sum Markov games [Shapley, 1953], i.e. there exists an optimal policy pair $(\pi^\star, \phi^\star)$ such that

$$\max_{\pi \in \Delta_{\mathcal{A}}^{\mathcal{S}}} \min_{\phi \in \Delta_{\mathcal{B}}^{\mathcal{S}}} J(\pi, \phi) = \min_{\phi \in \Delta_{\mathcal{B}}^{\mathcal{S}}} \max_{\pi \in \Delta_{\mathcal{A}}^{\mathcal{S}}} J(\pi, \phi) = J(\pi^\star, \phi^\star). \tag{1}$$

However, as $J$ is generally non-concave with respect to the policy of the first player and non-convex with respect to that of the second player, direct GDA updates may not find $(\pi^\star, \phi^\star)$ and usually exhibit an oscillation behavior, which we illustrate through numerical simulations in Section 5. Our approach to address this issue is to enhance the structure of the Markov game through regularization.

## 2.1 Entropy-Regularized Two-Player Zero-Sum Markov Games

In this section we define the entropy regularization and discuss structure of the regularized objective function and its connection to the original problem. Let the regularizers be

$$\mathcal{H}_\pi(s, \pi, \phi) \triangleq \mathbb{E}_{a_k \sim \pi(\cdot|s_k), b_k \sim \phi(\cdot|s_k), s_{k+1} \sim \mathcal{P}(\cdot|s_k, a_k, b_k)} \left[ \sum_{k=0}^{\infty} -\gamma^k \log \pi(a_k \mid s_k) \mid s_0 = s \right],$$

$$\mathcal{H}_\phi(s, \pi, \phi) \triangleq \mathbb{E}_{a_k \sim \pi(\cdot|s_k), b_k \sim \phi(\cdot|s_k), s_{k+1} \sim \mathcal{P}(\cdot|s_k, a_k, b_k)} \left[ \sum_{k=0}^{\infty} -\gamma^k \log \phi(b_k \mid s_k) \mid s_0 = s \right].$$

We define the regularized value function

$$V_\tau^{\pi, \phi}(s) \triangleq V^{\pi, \phi}(s) + \tau \mathcal{H}_\pi(s, \pi, \phi) - \tau \mathcal{H}_\phi(s, \pi, \phi)$$

$$= \mathbb{E}_{\pi, \phi, \mathcal{P}} \left[ \sum_{k=0}^{\infty} \gamma^k \left( r(s_k, a_k, b_k) - \tau \log \pi(a_k \mid s_k) + \tau \log \phi(b_k \mid s_k) \right) \mid s_0 = s \right],$$

where $\tau \geq 0$ is a weight parameter. Again under a fixed initial distribution $\rho \in \Delta_{\mathcal{S}}$ we denote $J_\tau(\pi, \phi) \triangleq \mathbb{E}_{s \sim \rho}[V_\tau^{\pi, \phi}(s)]$. The regularized advantage function is

$$A_\tau^{\pi, \phi}(s, a, b) \triangleq r(s, a, b) - \tau \log \pi(a \mid s) + \tau \log \phi(b \mid s) + \gamma \mathbb{E}_{s' \sim \mathcal{P}(\cdot|s, a, b)} \left[ V_\tau^{\pi, \phi}(s') \right] - V_\tau^{\pi, \phi}(s),$$

which later helps us to express the policy gradient.

We use $d_\rho^{\pi, \phi} \in \Delta_{\mathcal{S}}$ to denote the discounted visitation distribution under any policy pair $(\pi, \phi)$ and the initial state distribution $\rho$

$$d_\rho^{\pi, \phi}(s) \triangleq (1 - \gamma) \mathbb{E}_{\pi, \phi, \mathcal{P}} \left[ \sum_{k=0}^{\infty} \gamma^k \mathbf{1}(s_k = s) \mid s_0 \sim \rho \right]$$

For sufficient state visitation, we assume that the initial state distribution is bounded away from zero. This is a standard assumption in the entropy-regularized MDP literature [Mei et al., 2020, Ying et al., 2022].

**Assumption 1.** *The initial state distribution $\rho$ is strictly positive for any state, and we denote $\rho_{\min} = \min_{s \in \mathcal{S}} \rho(s) > 0$.*

When the policy of the first player is fixed to $\pi \in \Delta_{\mathcal{A}}^{\mathcal{S}}$, the Markov game reduces to an MDP for the second player with state transition probability $\widetilde{\mathcal{P}}_\phi(s' \mid s, b) = \sum_{a \in \mathcal{A}} \mathcal{P}(s' \mid s, a, b) \pi(a \mid s)$ and reward function $\widetilde{r}_\phi(s, b) = \sum_{a \in \mathcal{A}} r(s, a, b) \pi(a \mid s)$. A similar argument holds for the first player if the second player's policy is fixed. To denote the operators that map one player's policy to the best response of the other player and the corresponding value function, we define

$$\pi_\tau(\phi) \triangleq \operatorname*{argmax}_{\pi \in \Delta_{\mathcal{A}}^{\mathcal{S}}} J_\tau(\pi, \phi), \quad \phi_\tau(\pi) \triangleq \operatorname*{argmin}_{\phi \in \Delta_{\mathcal{B}}^{\mathcal{S}}} J_\tau(\pi, \phi),$$

$$g_\tau(\pi) \triangleq \min_{\phi \in \Delta_{\mathcal{B}}^{\mathcal{S}}} J_\tau(\pi, \phi) = J_\tau(\pi, \phi_\tau(\pi)). \tag{2}$$

For any $\tau > 0$, the following lemma bounds the performance difference between optimal and suboptimal policies and establishes the uniqueness of $\pi_\tau(\phi)$ and $\phi_\tau(\pi)$. When $\tau = 0$, we use $\pi_0(\phi)$ and $\phi_0(\pi)$ to denote one of the maximizers and minimizers since they may not be unique.

**Lemma 1** (Performance Difference). *Under Assumption 1 and given $\tau > 0$, $\pi_\tau(\phi)$ is unique for any $\phi \in \Delta_{\mathcal{B}}^{\mathcal{S}}$, and $\phi_\tau(\pi)$ is unique for any $\pi \in \Delta_{\mathcal{A}}^{\mathcal{S}}$. Given any min player policy $\phi \in \Delta_{\mathcal{B}}^{\mathcal{S}}$,*

$$J_\tau(\pi_\tau(\phi), \phi) - J_\tau(\pi, \phi) \geq \frac{\tau \rho_{\min}}{2 \log(2)} \|\pi_\tau(\phi) - \pi\|^2, \quad \forall \pi \in \Delta_{\mathcal{A}}^{\mathcal{S}}. \tag{3}$$

*Given any max player policy $\pi \in \Delta_{\mathcal{A}}^{\mathcal{S}}$,*

$$J_\tau(\pi, \phi_\tau(\pi)) - J_\tau(\pi, \phi) \leq -\frac{\tau \rho_{\min}}{2 \log(2)} \|\phi_\tau(\pi) - \phi\|^2, \quad \forall \phi \in \Delta_{\mathcal{B}}^{\mathcal{S}}. \tag{4}$$

The Nash equilibrium of the regularized problem is sometimes referred to as the quantal response equilibrium [McKelvey and Palfrey, 1995] and is known to exist under any $\tau$. Leveraging Lemma 1, we formally state the conditions guaranteeing its existence and affirm that it is unique.

**Lemma 2** (Minimax Theorem for Entropy-Regularized Markov Game). *Under Assumption 1, for any regularization weight $\tau > 0$, there exists a unique Nash equilibrium policy pair $(\pi_\tau^\star, \phi_\tau^\star)$ such that*

$$\max_{\pi \in \Delta_{\mathcal{A}}^{\mathcal{S}}} \min_{\phi \in \Delta_{\mathcal{B}}^{\mathcal{S}}} J_\tau(\pi, \phi) = \min_{\phi \in \Delta_{\mathcal{B}}^{\mathcal{S}}} \max_{\pi \in \Delta_{\mathcal{A}}^{\mathcal{S}}} J_\tau(\pi, \phi) = J_\tau(\pi_\tau^\star, \phi_\tau^\star). \tag{5}$$

We are only interested in the solution of the regularized Markov game if it gives us knowledge of the original problem in (1). In the following lemma, we show that the distance between the Nash equilibrium of the regularized game and that of the original one is bounded by the regularization weight. This is an important condition guaranteeing that we can find an approximate solution to the original Markov game by solving the regularized problem. In addition, this lemma also shows that the same policy pair produces value functions with bounded distance under two regularization weights.

**Lemma 3.** *For any $\tau \geq \tau' \geq 0$ and policy $\pi$,*

$$-(\tau - \tau') \log |\mathcal{B}| \leq J_\tau(\pi_\tau^\star, \phi_\tau^\star) - J_{\tau'}(\pi_{\tau'}^\star, \phi_{\tau'}^\star) \leq (\tau - \tau') \log |\mathcal{A}|. \tag{6}$$

$$-(\tau - \tau') \log |\mathcal{B}| \leq g_\tau(\pi) - g_{\tau'}(\pi) = J_\tau(\pi, \phi_\tau(\pi)) - J_{\tau'}(\pi, \phi_{\tau'}(\pi)) \leq (\tau - \tau') \log |\mathcal{A}|. \tag{7}$$

$$-\frac{\tau - \tau'}{1 - \gamma} \log |\mathcal{B}| \leq J_\tau(\pi, \phi) - J_{\tau'}(\pi, \phi) \leq \frac{\tau - \tau'}{1 - \gamma} \log |\mathcal{A}|. \tag{8}$$

## 2.2 Softmax Parameterization

In this work we use a tabular softmax policy parameterization and maintain two tables $\theta \in \mathbb{R}^{\mathcal{S} \times \mathcal{A}}$, $\psi \in \mathbb{R}^{\mathcal{S} \times \mathcal{B}}$ that parameterize the policies of the two players according to

$$\pi_\theta(a \mid s) = \frac{\exp(\theta(s, a))}{\sum_{a' \in \mathcal{A}} \exp(\theta(s, a'))}, \quad \text{and} \quad \phi_\psi(b \mid s) = \frac{\exp(\psi(s, b))}{\sum_{b' \in \mathcal{A}} \exp(\psi(s, b'))}.$$

The gradients of the regularized value function with respect to the policy parameters admit closed-form expressions

$$\frac{\partial J_\tau(\pi_\theta, \phi_\psi)}{\partial \theta(s, a)} = \frac{1}{1 - \gamma} d_\rho^{\pi_\theta, \phi_\psi}(s) \pi_\theta(a \mid s) \sum_{b \in \mathcal{B}} \phi_\psi(b \mid s) A_\tau^{\pi_\theta, \phi_\psi}(s, a, b),$$

$$\frac{\partial J_\tau(\pi_\theta, \phi_\psi)}{\partial \psi(s, b)} = \frac{1}{1 - \gamma} d_\rho^{\pi_\theta, \phi_\psi}(s) \phi_\psi(b \mid s) \sum_{a \in \mathcal{A}} \pi_\theta(a \mid s) A_\tau^{\pi_\theta, \phi_\psi}(s, a, b),$$

and computing them exactly requires knowledge of the dynamics of the environment. Note that the gradients of value function and the regularizer are Lipschitz with respect to the policy parameters with constants $L_V = \frac{8}{(1-\gamma)^3}$ and $L_{\mathcal{H}} = \frac{4+8 \log |\mathcal{A}|}{(1-\gamma)^3}$. This property is more formally stated and proved in Lemmas 5 and 6 of the appendix.

We next present an important property that we will later exploit to study the convergence of the GDA updates to the solution of the regularized Markov game. Under the softmax parameterization, the regularized value function enjoys a gradient domination condition with respect to the policy parameter that resembles the PŁ condition.

**Lemma 4** (PL-Type Condition). *Under Assumption 1, we have for any $\theta \in \mathbb{R}^{\mathcal{S} \times \mathcal{A}}$ and $\psi \in \mathbb{R}^{\mathcal{S} \times \mathcal{B}}$*

$$\|\nabla_\theta J_\tau(\pi_\theta, \phi_\psi)\|^2 \geq \frac{2(1-\gamma)\tau\rho_{\min}^2}{|\mathcal{S}|} \left(\min_{s,a} \pi_\theta(a \mid s)\right)^2 (J_\tau(\pi_\tau(\phi_\psi), \phi_\psi) - J_\tau(\pi_\theta, \phi_\psi)),$$

$$\|\nabla_\psi J_\tau(\pi_\theta, \phi_\psi)\|^2 \geq \frac{2(1-\gamma)\tau\rho_{\min}^2}{|\mathcal{S}|} \left(\min_{s,b} \phi_\psi(b \mid s)\right)^2 (J_\tau(\pi_\theta, \phi_\psi) - J_\tau(\pi_\theta, \phi_\tau(\pi_\theta))).$$

The PŁ condition is a tool commonly used in the optimization community to show the linear convergence of the gradient descent algorithm [Karimi et al., 2016, Yu and Jin, 2019, Khaled and Richtárik, 2020, Zeng et al., 2021b]. The condition in Lemma 4 is weaker than the common PŁ condition in two aspects. First, our PŁ coefficient is a function of the smallest policy entry. When we seek to bound the gradient of the iterates $\|\nabla_\theta J_\tau(\pi_{\theta_k}, \phi_{\psi_k})\|^2$ and $\|\nabla_\psi J_\tau(\pi_{\theta_k}, \phi_{\psi_k})\|^2$ later in the analysis, the PŁ coefficients will depend on $\min_{s,a} \pi_{\theta_k}(a \mid s)$ and $\min_{s,b} \phi_{\psi_k}(b \mid s)$, which may not be lower bounded by any positive constant. Second, the coefficients involve $\tau$, which is not a constant but needs to be carefully chosen to control the error between the regularized problem and the original one.

## 3 Solving Regularized Markov Games

Leveraging the structure introduced in Section 2, our first aim is to establish the finite-time convergence of the GDA algorithm to the Nash equilibrium of the regularized Markov game under a fixed regularization weight $\tau > 0$. The GDA algorithm executes the updates

$$\theta_{k+1} = \theta_k + \alpha_k \nabla_\theta J_\tau(\pi_{\theta_k}, \phi_{\psi_k}), \qquad \psi_{k+1} = \psi_k - \beta_k \nabla_\psi J_\tau(\pi_{\theta_{k+1}}, \phi_{\psi_k}). \tag{9}$$

The convergence bound we will derive reflects a trade-off for the regularization weight $\tau$: when $\tau$ is large, we get faster convergence to the Nash equilibrium of the regularized problem, but it is farther away from the Nash equilibrium of the original one. The result in this section will inspire the $\tau$ adjustment schemes designed later in the paper to achieve the best possible convergence to the Nash equilibrium of the original unregularized Markov game.

It can be shown that the Nash equilibrium of the regularized Markov game is a pair of completely mixed policies, i.e. $\forall \tau > 0$ there exists $c_\tau > 0$ such that $\min_{s,a} \pi_\tau^\star(a \mid s) \geq c_\tau$, and $\min_{s,b} \phi_\tau^\star(b \mid s) \geq c_\tau$ [Nachum et al., 2017]. In this work, we further assume the existence of a uniform lower bound on the entries of $(\pi_\tau^\star, \phi_\tau^\star)$ across $\tau$. We provide more explanation of the assumption in Remark 1.

**Assumption 2.** *There exists a positive constant $c$ (independent of $\tau$) such that for any $\tau > 0$*

$$\min_{s,a} \pi_\tau^\star(a \mid s) \geq c, \quad \min_{s,b} \phi_\tau^\star(b \mid s) \geq c.$$

To measure the convergence of the iterates to the Nash equilibrium of the regularized Markov game, we recall the definition of $g_\tau$ in (2) and define

$$\delta_k^\pi = J_\tau(\pi_\tau^\star, \phi_\tau^\star) - g_\tau(\pi_{\theta_k}), \quad \delta_k^\phi = J_\tau(\pi_{\theta_k}, \phi_{\psi_k}) - g_\tau(\pi_{\theta_k}). \tag{10}$$

The convergence metric is asymmetric for two players: the first player is quantified by its performance when the second player takes the most adversarial policy, while the second player is evaluated under the current policy iterate of the first player. We note that $\delta_k^\pi$ and $\delta_k^\phi$ are non-negative, and $\delta_k^\pi = \delta_k^\phi = 0$ implies that $(\pi_{\theta_k}, \phi_{\psi_k})$ is the Nash equilibrium. Under this convergence metric, the following theorem states that the GDA updates in (9) solve the regularized Markov game linearly fast. The proofs of the theoretical results of this paper are presented in Section A of the appendix.

**Theorem 1.** *We define $L = 3L_{\mathcal{H}} \max\{\tau, 1\}$, $C_1 = \frac{\rho_{\min}c^2}{64\log(2)}$, and $C_2 = \frac{2\sqrt{|\mathcal{S}|}}{\sqrt{(1-\gamma)\rho_{\min}c}}$, and choose the initial policy parameters to be $\theta_0 = 0 \in \mathbb{R}^{|\mathcal{S}| \times |\mathcal{A}|}$ and $\psi_0 = 0 \in \mathbb{R}^{|\mathcal{S}| \times |\mathcal{B}|}$ (the initial policies $\pi_{\theta_0}$ and $\phi_{\psi_0}$ are uniform). Let the step sizes of (9) be*

$$\alpha_k = \alpha, \quad \beta_k = \beta,$$

*with $\alpha, \beta$ satisfying*

$$\max\{\alpha, \beta\} \leq \frac{1}{L}, \quad \frac{\alpha}{\beta} \leq \min\{\frac{(1-\gamma)\rho_{\min}^3 c^2 \tau^2}{152\log(2)|\mathcal{S}|L^2}, 8\}, \quad \alpha \leq \min\{(L + \frac{C_2 L^2}{\tau})^{-1}, \frac{16|\mathcal{S}|}{(1-\gamma)\rho_{\min}^2 c^2 \tau}\}.$$

*If Assumption 1 holds and*

$$3\delta_0^\pi + \delta_0^\phi \le C_1\tau, \tag{11}$$

*then the iterates of (9) satisfy for all $k \ge 0$*

$$3\delta_k^\pi + \delta_k^\phi \le (1 - \frac{(1-\gamma)\alpha\tau\rho_{\min}^2 c^2}{32|\mathcal{S}|})^k (3\delta_0^\pi + \delta_0^\phi).$$

Theorem 1 establishes the linear convergence of the iterates of (9) to the Nash equilibrium of (5), provided that the initial condition (11) is satisfied. The convergence is faster when $\tau$ is large and slower when $\tau$ is small. Choosing $\tau$ to be large enough guarantees the initial condition (see Section C of the appendix for more discussion) but causes the Nash equilibrium of the regularized Markov game to be distant from that of the original Markov game. This motivates us to make the regularization weight a decaying sequence that starts off large enough to meet the initial condition and becomes smaller over time to narrow the gap between the regularized Markov game and the original one. We discuss two such schemes of reducing the regularization weight in the next section.

## 4 Main Results - Solving the Original Markov Game

This section presents two approaches to adjust the regularization weight that allow the GDA algorithm to converge to the Nash equilibrium of the original Markov game. The first approach uses a piecewise constant weight and results in the nested-loop updates stated in Algorithm 1. In the inner loop the regularization weight and step sizes are fixed, and the two players update their policy iterates towards the Nash equilibrium of the regularized Markov game. The outer loop iteration reduces the regularization weight to make the regularized Markov game approach the original one. The regularization weight decays geometrically in the outer loop, i.e. $\tau_{t+1} = \eta\tau_t$, where $\eta \in (0, 1)$ must be carefully balanced. On the one hand, recalling the definition of $g_\tau$ in (2) and defining

$$\delta_{t,k}^\pi = J_{\tau_t}(\pi_{\tau_t}^\star, \phi_{\tau_t}^\star) - g_{\tau_t}(\pi_{\theta_{t,k}}), \quad \delta_{t,k}^\phi = J_{\tau_t}(\pi_{\theta_{t,k}}, \phi_{\psi_{t,k}}) - g_{\tau_t}(\pi_{\theta_{t,k}}),$$

we need $\eta$ to be large enough that if $\theta_{t,0}$ and $\psi_{t,0}$ observe the initial condition $3\delta_{t,0}^\pi + \delta_{t,0}^\phi \le C_1\tau_t$, then so do $\theta_{t+1,0}$ and $\psi_{t+1,0}$ in the worst case. On the other hand, an $\eta$ selected excessively large makes the reduction of $\tau_t$ too slow to achieve the best possible convergence rate. Our next theoretical result, as a corollary of Theorem 1, properly chooses $\eta$ and $K_t$ and establishes the convergence of Algorithm 1 to the Nash equilibrium of the original original problem.

---

**Algorithm 1:** Nested-Loop Policy Gradient Descent Ascent Algorithm with Piecewise Constant Regularization Weight

---

**Initialize:** Policy parameters $\theta_{0,0} = 0 \in \mathbb{R}^{\mathcal{S} \times \mathcal{A}}$ and $\psi_{0,0} = 0 \in \mathbb{R}^{\mathcal{S} \times \mathcal{B}}$, step size sequences $\{\alpha_t\}$
  and $\{\beta_t\}$, an initial regularization parameter $\tau_0$
**for** $t = 0, 1, \cdots, T$ **do**
  **for** $k = 0, 1, \cdots, K_t - 1$ **do**
    1) Max player update:

$$\theta_{t,k+1} = \theta_{t,k} + \alpha_t \nabla_\theta J_\tau(\pi_{\theta_{t,k}}, \phi_{\psi_{t,k}})$$

    2) Min player update:

$$\psi_{t,k+1} = \psi_{t,k} - \beta_t \nabla_\psi J_\tau(\pi_{\theta_{t,k+1}}, \phi_{\psi_{t,k}})$$

  **end**
  Set initial policies for next outer loop iteration $\theta_{t+1,0} = \theta_{t,K_t}, \psi_{t+1,0} = \psi_{t,K_t}$
  Reduce regularization weight $\tau_{t+1} = \eta\tau_t$ and properly adjust $\alpha_t, \beta_t$
**end**

---

**Corollary 1.** *Suppose that Assumption 1-2 hold and $\tau_0$ is chosen such that $3\delta_{0,0}^\pi + \delta_{0,0}^\phi \le C_1\tau_0$[1]. We choose $\eta = \frac{C_1 + 2L_\delta}{2C_1 + 2L_\delta}$, where $L_\delta = 4\log|\mathcal{A}| + 3\log|\mathcal{B}| + \frac{\log|\mathcal{B}|}{1-\gamma}$ and $C_1$ is defined in Theorem 1.*

---

[1]This inequality is guaranteed to hold with a large enough $\tau_0$ if $\pi_{\theta_0}$ and $\phi_{\psi_0}$ are initialized to be uniform. See Section C of the appendix for more discussion.

*Then, under proper choices of $\alpha_t$ and $\beta_t$, the iterates of Algorithm 1 converge to a point such that*

$$J(\pi^\star, \phi^\star) - g_0(\pi_{\theta_{T,0}}) \leq \epsilon \quad \text{and} \quad J(\pi_{\theta_{T,0}}, \phi_{\psi_{T,0}}) - g_0(\pi_{\theta_{T,0}}) \leq \epsilon \tag{12}$$

*in at most $T = \mathcal{O}(\log(\epsilon^{-1}))$ outer loop iterations. The total number of gradient updates required is $\sum_{t=0}^{T} K_t = \mathcal{O}(\epsilon^{-3})$.*

Corollary 1 guarantees that $(\pi_{\theta_T}, \phi_{\psi_T})$ converge to an $\epsilon$-approximate Nash equilibrium of the original Markov game in $T = \mathcal{O}(\epsilon^{-3})$ gradient steps. In order to achieve this rate, $K_t$ has to be adjusted along with $\tau_t$: we need $K_t = \mathcal{O}(\tau_t^{-3})$ when $\tau_t$ becomes smaller than 1. The varying number of inner loop iterations may cause inconvenience for practical implementation. To address this issue, we next propose another scheme of adjusting the regularization weight that is carried out online along with the update of the policy iterates.

---

**Algorithm 2:** Policy Gradient Descent Ascent Algorithm with Diminishing Regularization Weight

---

**Initialize:** Policy parameters $\theta_0 = 0 \in \mathbb{R}^{\mathcal{S} \times \mathcal{A}}$ and $\psi_0 = 0 \in \mathbb{R}^{\mathcal{S} \times \mathcal{B}}$, step size sequences $\{\alpha_k\}$ and $\{\beta_k\}$, regularization parameter sequence $\{\tau_k\}$

**for** $k = 0, 1, \cdots, K$ **do**

    1) Max player update:

$$\theta_{k+1} = \theta_k + \alpha_k \nabla_\theta J_{\tau_k}(\pi_{\theta_k}, \phi_{\psi_k})$$

    2) Min player update:

$$\psi_{k+1} = \psi_k - \beta_k \nabla_\psi J_{\tau_k}(\pi_{\theta_{k+1}}, \phi_{\psi_k})$$

**end**

---

Presented in Algorithm 2, the second approach is a single-loop algorithm that reduces the regularization weight as a polynomial function of the iteration $k$. We define the auxiliary convergence metrics

$$\delta_k^\pi = J_{\tau_k}(\pi_{\tau_k}^\star, \phi_{\tau_k}^\star) - g_{\tau_k}(\pi_{\theta_k}), \quad \delta_k^\phi = J_{\tau_k}(\pi_{\theta_k}, \phi_{\psi_k}) - g_{\tau_k}(\pi_{\theta_k}),$$

which measure the convergence of $(\pi_{\theta_k}, \phi_{\psi_k})$ to the Nash equilibrium of the Markov game regularized with weight $\tau_k$. To judge the performance of the iterates in the original Markov game, we are ultimately interested in bounding $J(\pi^\star, \phi^\star) - g_0(\pi_{\theta_k})$ and $J(\pi_{\theta_k}, \phi_{\psi_k}) - g_0(\pi_{\theta_k})$. Thanks to Lemma 3, we can quantify how fast $\delta_k^\pi$ and $\delta_k^\phi$ approach these desired quantities as $\tau_k$ decays to 0. Under an initial condition on $\delta_k^\pi$ and $\delta_k^\phi$, we now establish the convergence rate of Algorithm 2 to $(\pi^\star, \phi^\star)$ of (1) through a multi-time-scale analysis.

**Theorem 2.** *Let the step sizes and regularization parameter be*

$$\alpha_k = \frac{\alpha_0}{(k+h)^{2/3}}, \quad \beta_k = \beta_0, \quad \tau_k = \frac{\tau_0}{(k+h)^{1/3}},$$

*with $\alpha_0$, $\beta_0$, $\tau_0$, and $h \geq 1$ satisfying a system of inequalities discussed in details in the analysis. Under Assumption 1-2, the iterates of Algorithm 2 satisfy for all $k \geq 0$*

$$J(\pi^\star, \phi^\star) - g_0(\pi_{\theta_k}) \leq \frac{C_1 \tau_0 + 3(\log|\mathcal{A}| + \log|\mathcal{B}|)\tau_0}{3(k+h)^{1/3}}, \tag{13}$$

$$J(\pi_{\theta_k}, \phi_{\psi_k}) - g_0(\pi_{\theta_k}) \leq \frac{(1-\gamma)C_1\tau_0 + (\log|\mathcal{A}| + \log|\mathcal{B}|)\tau_0}{(1-\gamma)(k+h)^{1/3}}, \tag{14}$$

*where the constant $C_1$ is defined in Theorem 1.*

Theorem 2 states that the last iterate of Algorithm 2 converges to an $\mathcal{O}(k^{-1/3})$-approximate Nash equilibrium of the original Markov game in $k$ iterations. This translates to the same sample complexity as Algorithm 1 derived in Corollary 1. Compared with Algorithm 1, reducing $\tau_k$ online along with the gradient updates in a single loop simplifies the algorithm and makes tracking the regularization weight, step sizes, and policy iterates simpler and more convenient. We note that the techniques in Daskalakis et al. [2020] may be used to analyze the finite-time performance of GDA for Markov games and lead to a convergence rate at least worse than $\mathcal{O}(k^{-1/10.5})$, which we improve over.

**Remark 1.** *Assumption 2 is a restrictive assumption that does not seem necessary but rather arises as an artifact of the current analysis. When we apply the weaker PL-type condition (Lemma 4) in the analysis, the entries of the iterates $\pi_{\theta_k}, \phi_{\psi_k}$ need to be uniformly lower bounded, which is difficult to establish using the game structure. We come up with an innovative induction approach to quantify the connection between $\min_{s,a} \pi_{\theta_k}(a \mid s), \min_{s,b} \phi_{\psi_k}(b \mid s)$ and the optimal gap $\delta_k^\pi, \delta_k^\phi$. This approach allows us to transform the uniform lower bound requirement on $\pi_{\theta_k}, \phi_{\psi_k}$ to that on the Nash equilibrium, leading to Assumption 2. It is a future work to remove/relax this assumption.*

*A Markov game is said to be completely mixed if every Nash equilibrium of the game consists of a pair of completely mixed policies, i.e. $\min_{s,a} \pi^\star(a \mid s) > 0, \min_{s,b} \phi^\star(b \mid s) > 0$ for any Nash equilibrium $(\pi^\star, \phi^\star)$ of the Markov game (if more than one exists). Assumption 2 intuitively seems no stronger than requiring the original Markov game to be completely mixed. If the original Markov game has at least one completely mixed Nash equilibrium, the Nash equilibrium of the regularized Markov game should also be completely mixed even when the regularization weight is small, since the regularization encourages the solution to be more uniform. The reward function that results in completely mixed Markov games is well studied in Raghavan [1978], Kaplansky [1995], Das et al. [2017].*

## 5   Numerical Simulations

In this section, we numerically verify the convergence of Algorithm 2 on small-scale synthetic Markov games. Our aim is to confirm that the algorithm indeed converges rather than to visualize the exact convergence rate, as achieving the theoretical rate derived in Theorem 2 requires very careful selection of all involved parameters. Considering an environment with $|\mathcal{S}| = 2$ and $|\mathcal{A}| = |\mathcal{B}| = 2$, we first choose the reward and transition probability kernel such that the Markov game is completely mixed[2].

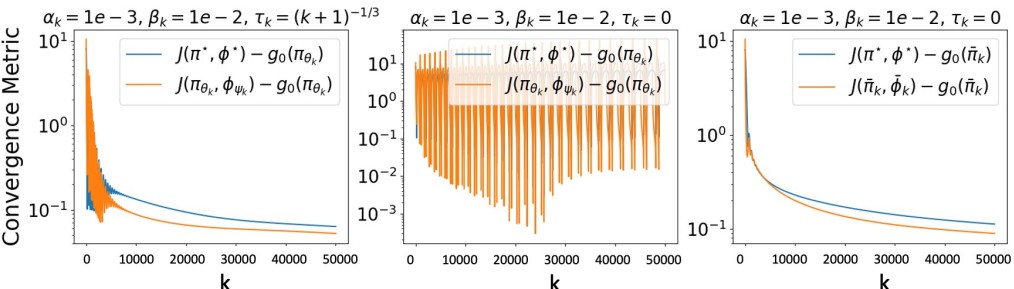

Figure 1: Convergence of GDA for a Completely Mixed Markov game

We run Algorithm 2 for 50000 iterations with $\alpha_k = 10^{-3}$, $\beta_k = 10^{-2}$, $\tau_k = (k+1)^{-1/3}$, and measure the convergence of $\pi_k$ and $\phi_k$ by metrics considered in (13) and (14) of Theorem 2. As shown in the first plot of Figure 1, the last iterate exhibits an initial oscillation behavior but converge smoothly after 10000 iterations. In comparison, we visualize the convergence of the last iterate and averaged iterate of the GDA algorithm without any regularization (second and third plots of Figure 1), where the average is computed with equal weights as $\bar{\pi}_k = \frac{1}{k+1} \sum_{t=0}^{k} \pi_{\theta_t}$, $\bar{\phi}_k = \frac{1}{k+1} \sum_{t=0}^{k} \phi_{\psi_t}$. The existing theoretical results in this case guarantee the convergence of the averaged iterate but not the last iterate [Daskalakis et al., 2020]. According to our simulations, the last iterate indeed does not converge, while the averaged iterate does, but at a slower rate than the convergence of the last iterate of the GDA algorithm under the decaying regularization.

The theoretical results derived in this paper rely on Assumption 2. To investigate whether this assumption is truly necessary, we also apply Algorithm 2 to a Markov game that has a deterministic

---

[2]To create a completely mixed game with $|\mathcal{A}| = |\mathcal{B}| = 2$, we simply need to choose the reward function such that $r(s, \cdot, \cdot)$ as a 2x2 matrix is diagonal dominant or sub-diagonal dominant for any state $s \in \mathcal{S}$, and we can use an arbitrary transition probability kernel. The exact choice of the reward function and transition kernel as well as the Nash equilibrium of this Markov game are presented in Section D of the appendix.

Nash equilibrium and does not observe Assumption 2[3]. As illustrated in Figure 2, the experiment shows that Algorithm 2 still converges correctly to $(\pi^\star, \phi^\star)$ of (1). This observation suggests that Assumption 2 may be an artifact of the current analysis and motivates for us to investigate ways to remove/relax this assumption in the future. We note that the pure GDA approach without regularization also has a last-iterate convergence and does not exhibit the oscillation behavior observed in Figure 1, since the gradients of both players never change signs regardless of the policy of the opponent in this Markov game.

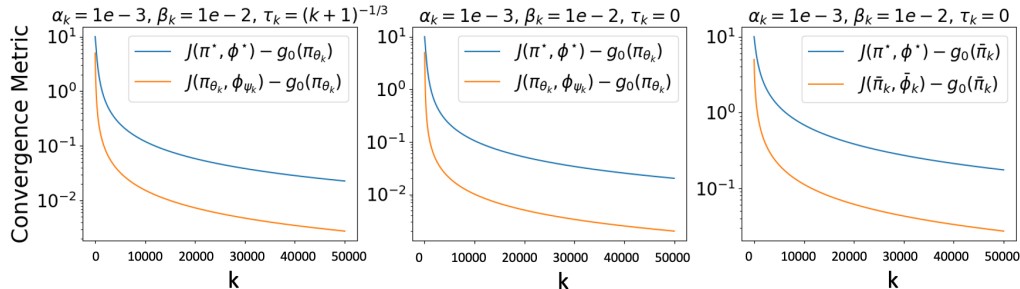

Figure 2: Convergence of GDA for a Deterministic Markov game

## 6 Conclusion & Future Work

In this paper, we present the finite-time analysis of two GDA algorithms that provably find the Nash equilibrium of a Markov game with the help of a structured entropy regularization. Future directions of this work include formalizing the link between Assumption 2 and completely mixed Markov games, investigating the possibility of relaxing this assumption, and characterizing the convergence of the stochastic GDA algorithm where the players do not have knowledge of the environment dynamics and can only take samples to estimate the gradients.

## Acknowledgement

Sihan Zeng and Justin Romberg were supported in part by ARL DCIST CRA W911NF-17-2-0181. The work of Thinh T. Doan was supported in part by the Commonwealth Cyber Initiative.

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
