# Supplementary Material - NeurIPS 2022

For convenience, we include a table of contents for the supplementary material below.

## Table of Contents

## A  Proof of Theorems and Corollaries

We frequently use the following inequalities which hold for all $\tau \geq 0$, $\pi \in \Delta_{\mathcal{S}}^{\mathcal{A}}$, and $\phi \in \Delta_{\mathcal{S}}^{\mathcal{B}}$,

$$J_\tau(\pi, \phi_\tau(\pi)) \leq J_\tau(\pi, \phi), \quad J_\tau(\pi_\tau(\phi), \phi) \geq J_\tau(\pi, \phi).$$

We use $H(\cdot)$ to denote the entropy of a distribution. For example,

$$H(\pi(\cdot \mid s)) = -\sum_a \pi(a \mid s) \log \pi(a \mid s), \quad H(\phi(\cdot \mid s)) = -\sum_b \phi(b \mid s) \log \phi(b \mid s). \quad (15)$$

Due to the uniqueness of $\phi_\tau(\cdot)$, Danskin's Theorem guarantees that $g_\tau(\pi_\theta)$ defined in (2) is differentiable with respect to $\theta$ [Bernhard and Rapaport, 1995]

$$\nabla_\theta g_\tau(\pi_\theta) = \nabla_\theta J_\tau(\pi_\theta, \phi), \quad \phi = \phi_\tau(\pi_\theta), \quad \forall \theta \in \mathbb{R}^{|\mathcal{S}| \times |\mathcal{A}|}. \quad (16)$$

We also introduce a few lemmas that will be applied regularly in the rest of the paper.

**Lemma 5.** *Let $L_V = \frac{8}{(1-\gamma)^3}$. The value function $J$ is $L_V$-Lipschitz continuous and has $L_V$-Lipschitz gradients, i.e. we have for all $\theta_1, \theta_2 \in \mathbb{R}^{|\mathcal{S}| \times |\mathcal{A}|}$ and $\psi_1, \psi_2 \in \mathbb{R}^{|\mathcal{S}| \times |\mathcal{B}|}$*

$$\|\nabla_\theta J(\pi_{\theta_1}, \phi_{\psi_1}) - \nabla_\theta J(\pi_{\theta_2}, \phi_{\psi_2})\| \le L_V(\|\theta_1 - \theta_2\| + \|\psi_1 - \psi_2\|),$$
$$\|\nabla_\psi J(\pi_{\theta_1}, \phi_{\psi_1}) - \nabla_\psi J(\pi_{\theta_2}, \phi_{\psi_2})\| \le L_V(\|\theta_1 - \theta_2\| + \|\psi_1 - \psi_2\|),$$
$$\|J(\pi_{\theta_1}, \phi_{\psi_1}) - J(\pi_{\theta_2}, \phi_{\psi_2})\| \le L_V(\|\theta_1 - \theta_2\| + \|\psi_1 - \psi_2\|).$$

**Lemma 6.** *Let $L_\mathcal{H} = \frac{4 + 8\log|\mathcal{A}|}{(1-\gamma)^3}$. The regularization functions $\mathcal{H}_\pi$ and $\mathcal{H}_\phi$ are $L_\mathcal{H}$-Lipschitz continuous and has $L_\mathcal{H}$-Lipschitz gradients.*

Lemmas 5 and 6 imply that $\forall \tau \ge 0$, $\nabla_\theta J_\tau$ is Lipschitz continuous, i.e. for any $\theta_1, \theta_2 \in \mathbb{R}^{|\mathcal{S}| \times |\mathcal{A}|}$, $\psi_1, \psi_2 \in \mathbb{R}^{|\mathcal{S}| \times |\mathcal{B}|}$

$$
\begin{aligned}
\|\nabla_\theta J_\tau(\pi_{\theta_1}, \phi_{\psi_1}) - \nabla_\theta J_\tau(\pi_{\theta_2}, \phi_{\psi_2})\| &\le \|\nabla_\theta J(\pi_{\theta_1}, \phi_{\psi_1}) - \nabla_\theta J(\pi_{\theta_2}, \phi_{\psi_2})\| \\
&\quad + \tau\|\nabla_\theta \mathcal{H}_\pi(s, \pi_{\theta_1}, \phi_{\psi_1}) - \nabla_\theta \mathcal{H}_\pi(s, \pi_{\theta_2}, \phi_{\psi_2})\| \\
&\quad + \tau\|\nabla_\theta \mathcal{H}_\phi(s, \pi_{\theta_1}, \phi_{\psi_1}) - \nabla_\theta \mathcal{H}_\phi(s, \pi_{\theta_2}, \phi_{\psi_2})\| \\
&\le (L_V + 2\tau L_\mathcal{H})(\|\theta_1 - \theta_2\| + \|\psi_1 - \psi_2\|). \quad (17)
\end{aligned}
$$

**Lemma 7.** *For any $0 \le a \le 1$ and integer $k > 0$, we have*

$$\frac{1}{(k+h)^a} - \frac{1}{(k+1+h)^a} \le \frac{8}{3(k+h)^{a+1}}.$$

### A.1 Proof of Theorem 1

The definition of the constant $L$ and Eq.(17) imply for any $\theta_1, \theta_2 \in \mathbb{R}^{|\mathcal{S}| \times |\mathcal{A}|}$, $\psi_1, \psi_2 \in \mathbb{R}^{|\mathcal{S}| \times |\mathcal{B}|}$

$$\|\nabla_\theta J_\tau(\pi_{\theta_1}, \phi_{\psi_1}) - \nabla_\theta J_\tau(\pi_{\theta_2}, \phi_{\psi_2})\| \le L(\|\theta_1 - \theta_2\| + \|\psi_1 - \psi_2\|). \quad (18)$$

We will use an induction argument to prove the convergence of $3\delta_k^\pi + \delta_k^\phi$. The base case is $3\delta_0^\pi + \delta_0^\phi \le 3\delta_0^\pi + \delta_0^\phi$, which obviously holds. Now, suppose

$$3\delta_k^\pi + \delta_k^\phi \le (1 - \frac{\alpha(1-\gamma)\tau\rho_{\min}^2 c^2}{32|\mathcal{S}|})^k (3\delta_0^\pi + \delta_0^\phi) \quad (19)$$

holds. We aim to show

$$3\delta_{k+1}^\pi + \delta_{k+1}^\phi \le (1 - \frac{\alpha(1-\gamma)\tau\rho_{\min}^2 c^2}{32|\mathcal{S}|})^{k+1} (3\delta_0^\pi + \delta_0^\phi).$$

We introduce the following technical lemmas.

**Lemma 8.** *Suppose* (19) *holds. Then, we have*

$$-\left(\min_{s,a} \pi_{\theta_k}(a \mid s)\right)^2 \le -\frac{3c^2}{8}, \quad (20)$$

$$-\left(\min_{s,b} \phi_{\psi_k}(b \mid s)\right)^2 \le -\frac{3c^2}{8}. \quad (21)$$

**Lemma 9.** *Suppose* (19) *holds. Under Assumption 1 and the step size $\alpha_k \le (L + \frac{2\sqrt{|\mathcal{S}|}L^2}{\sqrt{(1-\gamma)\rho_{\min}\tau c}})^{-1}$, we have*

$$
\begin{aligned}
&g_\tau(\theta_k) - g_\tau(\theta_{k+1}) \\
&= J_\tau(\pi_{\theta_k}, \phi_\tau(\pi_{\theta_k})) - J_\tau(\pi_{\theta_{k+1}}, \phi_\tau(\pi_{\theta_{k+1}})) \\
&= \frac{\alpha_k}{2} \left(\|\nabla_\theta J_\tau(\pi_{\theta_k}, \phi_\tau(\pi_{\theta_k})) - \nabla_\theta J_\tau(\pi_{\theta_k}, \phi_{\psi_k})\|^2 - \|\nabla_\theta J_\tau(\pi_{\theta_k}, \phi_\tau(\pi_{\theta_k}))\|^2\right).
\end{aligned}
$$

By the lemma above, we have

$$\delta_{k+1}^\pi - \delta_k^\pi$$

$$= J_\tau(\pi_{\theta_k}, \phi_\tau(\pi_{\theta_k})) - J_\tau(\pi_{\theta_{k+1}}, \phi_\tau(\pi_{\theta_{k+1}}))$$

$$\le \frac{\alpha_k}{2} \left( \|\nabla_\theta J_\tau(\pi_{\theta_k}, \phi_\tau(\pi_{\theta_k})) - \nabla_\theta J_\tau(\pi_{\theta_k}, \phi_{\psi_k})\|^2 - \|\nabla_\theta J_\tau(\pi_{\theta_k}, \phi_\tau(\pi_{\theta_k}))\|^2 \right). \tag{22}$$

Similarly, we consider the decay of $\delta_k^\phi$.

$$\delta_{k+1}^\phi - \delta_k^\phi = J_\tau(\pi_{\theta_{k+1}}, \phi_{\psi_{k+1}}) - g_\tau(\pi_{\theta_{k+1}}) - J_\tau(\pi_{\theta_k}, \phi_{\psi_k}) + g_\tau(\pi_{\theta_k})$$

$$= \left( J_\tau(\pi_{\theta_{k+1}}, \phi_{\psi_{k+1}}) - J_\tau(\pi_{\theta_{k+1}}, \phi_{\psi_k}) \right)$$

$$+ \left( J_\tau(\pi_{\theta_{k+1}}, \phi_{\psi_k}) - J_\tau(\pi_{\theta_k}, \phi_{\psi_k}) \right) + \left( g_\tau(\pi_{\theta_k}) - g_\tau(\pi_{\theta_{k+1}}) \right). \tag{23}$$

Using the $L$-smoothness of the value function derived in (18)

$$J_\tau(\pi_{\theta_{k+1}}, \phi_{\psi_{k+1}}) - J_\tau(\pi_{\theta_{k+1}}, \phi_{\psi_k})$$

$$\le \langle \nabla_\psi J_\tau(\pi_{\theta_{k+1}}, \phi_{\psi_k}), \psi_{k+1} - \psi_k \rangle + \frac{L}{2} \|\psi_{k+1} - \psi_k\|^2$$

$$= -\beta_k \|\nabla_\psi J_\tau(\pi_{\theta_{k+1}}, \phi_{\psi_k})\|^2 + \frac{L\beta_k^2}{2} \|\nabla_\psi J_\tau(\pi_{\theta_{k+1}}, \phi_{\psi_k})\|^2$$

$$\le -\frac{\beta_k}{2} \|\nabla_\psi J_\tau(\pi_{\theta_{k+1}}, \phi_{\psi_k})\|^2$$

$$\le -\frac{(1-\gamma)\beta_k \tau \rho_{\min}^2}{|\mathcal{S}|} \left( \min_{s,b} \phi_{\psi_k}(b \mid s) \right)^2 \left( J_\tau(\pi_{\theta_k}, \phi_{\psi_k}) - J_\tau(\pi_{\theta_k}, \phi_\tau(\pi_{\theta_k})) \right)$$

$$= -\frac{(1-\gamma)\beta_k \tau \rho_{\min}^2}{|\mathcal{S}|} \left( \min_{s,b} \phi_{\psi_k}(b \mid s) \right)^2 \delta_k^\phi,$$

where the second inequality uses $\beta_k \le \frac{1}{L}$ and the third inequality follows from Lemma 4 and the fact that $d_\rho^{\pi,\phi}(s) \le 1$ for all $s \in \mathcal{S}$ and policies $\pi, \phi$.

Using Eq. (21) of Lemma 8 to further simplify this inequality,

$$J_\tau(\pi_{\theta_{k+1}}, \phi_{\psi_{k+1}}) - J_\tau(\pi_{\theta_{k+1}}, \phi_{\psi_k}) \le -\frac{3(1-\gamma)\beta_k \tau \rho_{\min}^2 c^2}{8|\mathcal{S}|} \delta_k^\phi. \tag{24}$$

For the second term of (23), we have from the $L$-smoothness of the value function derived in (18)

$$J_\tau(\pi_{\theta_{k+1}}, \phi_{\psi_k}) - J_\tau(\pi_{\theta_k}, \phi_{\psi_k}) \le \langle \nabla_\theta J_\tau(\pi_{\theta_k}, \phi_{\psi_k}), \theta_{k+1} - \theta_k \rangle + \frac{L}{2} \|\theta_{k+1} - \theta_k\|^2$$

$$= \alpha_k \|\nabla_\theta J_\tau(\pi_{\theta_k}, \phi_{\psi_k})\|^2 + \frac{L\alpha_k^2}{2} \|\nabla_\theta J_\tau(\pi_{\theta_k}, \phi_{\psi_k})\|^2$$

$$\le \frac{3\alpha_k}{2} \|\nabla_\theta J_\tau(\pi_{\theta_k}, \phi_{\psi_k})\|^2, \tag{25}$$

where in the last inequality we use $\alpha_k L \le 1$.

Similarly to (22), the last term of (23) is bounded as

$$g_\tau(\pi_{\theta_k}) - g_\tau(\pi_{\theta_{k+1}}) = g_\tau(\pi_{\theta_k}) - g_\tau(\pi_{\theta_{k+1}}) + g_\tau(\pi_{\theta_{k+1}}) - g_\tau(\pi_{\theta_{k+1}})$$

$$\le \frac{\alpha_k}{2} \left( \|\nabla_\theta J_\tau(\pi_{\theta_k}, \phi_\tau(\pi_{\theta_k})) - \nabla_\theta J_\tau(\pi_{\theta_k}, \phi_{\psi_k})\|^2 - \|\nabla_\theta J_\tau(\pi_{\theta_k}, \phi_\tau(\pi_{\theta_k}))\|^2 \right) \tag{26}$$

Using (24)-(26) in (23), we have

$$\delta_{k+1}^\phi = \left( J_\tau(\pi_{\theta_{k+1}}, \phi_{\psi_{k+1}}) - J_\tau(\pi_{\theta_{k+1}}, \phi_{\psi_k}) \right)$$

$$+ \left( J_\tau(\pi_{\theta_{k+1}}, \phi_{\psi_k}) - J_\tau(\pi_{\theta_k}, \phi_{\psi_k}) \right) + \left( g_\tau(\pi_{\theta_k}) - g_\tau(\pi_{\theta_{k+1}}) \right)$$

$$\le (1 - \frac{3(1-\gamma)\beta_k \tau \rho_{\min}^2 c^2}{8|\mathcal{S}|}) \delta_k^\phi + \frac{3\alpha_k}{2} \|\nabla_\theta J_\tau(\pi_{\theta_k}, \phi_{\psi_k})\|^2$$

$$+ \frac{\alpha_k}{2} \left( \|\nabla_\theta J_\tau(\pi_{\theta_k}, \phi_\tau(\pi_{\theta_k})) - \nabla_\theta J_\tau(\pi_{\theta_k}, \phi_{\psi_k})\|^2 - \|\nabla_\theta J_\tau(\pi_{\theta_k}, \phi_\tau(\pi_{\theta_k}))\|^2 \right). \tag{27}$$

Combining (22) and (27),

$$3\delta_{k+1}^\pi + \delta_{k+1}^\phi \le 3\delta_k^\pi + \frac{3\alpha_k}{2}\left(\|\nabla_\theta J_\tau(\pi_{\theta_k},\phi_\tau(\pi_{\theta_k})) - \nabla_\theta J_\tau(\pi_{\theta_k},\phi_{\psi_k})\|^2 - \|\nabla_\theta J_\tau(\pi_{\theta_k},\phi_\tau(\pi_{\theta_k}))\|^2\right)$$

$$+ (1 - \frac{3(1-\gamma)\beta_k\tau\rho_{\min}^2 c^2}{8|\mathcal{S}|})\delta_k^\phi + 2\delta_k^\phi)\delta_k^\phi + \frac{3\alpha_k}{2}\|\nabla_\theta J_\tau(\pi_{\theta_k},\phi_{\psi_k})\|^2$$

$$+ \frac{\alpha_k}{2}\left(\|\nabla_\theta J_\tau(\pi_{\theta_k},\phi_\tau(\pi_{\theta_k})) - \nabla_\theta J_\tau(\pi_{\theta_k},\phi_{\psi_k})\|^2 - \|\nabla_\theta J_\tau(\pi_{\theta_k},\phi_\tau(\pi_{\theta_k}))\|^2\right)$$

$$\le 3\delta_k^\pi + (1 - \frac{3(1-\gamma)\beta_k\tau\rho_{\min}^2 c^2}{8|\mathcal{S}|})\delta_k^\phi + \frac{3\alpha_k}{2}\|\nabla_\theta J_\tau(\pi_{\theta_k},\phi_{\psi_k})\|^2$$

$$+ 2\alpha_k\|\nabla_\theta J_\tau(\pi_{\theta_k},\phi_\tau(\pi_{\theta_k})) - \nabla_\theta J_\tau(\pi_{\theta_k},\phi_{\psi_k})\|^2 - 2\alpha_k\|\nabla_\theta J_\tau(\pi_{\theta_k},\phi_\tau(\pi_{\theta_k}))\|^2.$$

Simplifying this inequality with

$$\|\nabla_\theta J_\tau(\pi_{\theta_k},\phi_{\psi_k})\|^2 = \|\nabla_\theta J_\tau(\pi_{\theta_k},\phi_\tau(\pi_{\theta_k})) - (\nabla_\theta J_\tau(\pi_{\theta_k},\phi_\tau(\pi_{\theta_k})) - \nabla_\theta J_\tau(\pi_{\theta_k},\phi_{\psi_k}))\|^2$$

$$\le \|\nabla_\theta J_\tau(\pi_{\theta_k},\phi_\tau(\pi_{\theta_k}))\|^2 + \|\nabla_\theta J_\tau(\pi_{\theta_k},\phi_\tau(\pi_{\theta_k})) - \nabla_\theta J_\tau(\pi_{\theta_k},\phi_{\psi_k})\|^2$$

$$+ 2\langle\nabla_\theta J_\tau(\pi_{\theta_k},\phi_\tau(\pi_{\theta_k})), \nabla_\theta J_\tau(\pi_{\theta_k},\phi_\tau(\pi_{\theta_k})) - \nabla_\theta J_\tau(\pi_{\theta_k},\phi_{\psi_k})\rangle$$

$$\le \frac{5}{4}\|\nabla_\theta J_\tau(\pi_{\theta_k},\phi_\tau(\pi_{\theta_k}))\|^2 + 5\|\nabla_\theta J_\tau(\pi_{\theta_k},\phi_\tau(\pi_{\theta_k})) - \nabla_\theta J_\tau(\pi_{\theta_k},\phi_{\psi_k})\|^2,$$

we have

$$3\delta_{k+1}^\pi + \delta_{k+1}^\phi \le 3\delta_k^\pi + (1 - \frac{3(1-\gamma)\beta_k\tau\rho_{\min}^2 c^2}{8|\mathcal{S}|})\delta_k^\phi - \frac{\alpha_k}{8}\|\nabla_\theta J_\tau(\pi_{\theta_k},\phi_\tau(\pi_{\theta_k}))\|^2$$

$$+ \frac{19\alpha_k}{2}\|\nabla_\theta J_\tau(\pi_{\theta_k},\phi_\tau(\pi_{\theta_k})) - \nabla_\theta J_\tau(\pi_{\theta_k},\phi_{\psi_k})\|^2. \tag{28}$$

Using Lemma 4 to bound $-\|\nabla_\theta J_\tau(\pi_{\theta_k},\phi_\tau(\pi_{\theta_k}))\|^2$,

$$- \|\nabla_\theta J_\tau(\pi_{\theta_k},\phi_\tau(\pi_{\theta_k}))\|^2$$

$$\le -\frac{2(1-\gamma)\tau\rho_{\min}^2}{|\mathcal{S}|}\left(\min_{s,a}\pi_{\theta_k}(a\mid s)\right)^2 (J_\tau(\pi_\tau(\phi_\tau(\pi_{\theta_k})),\phi_\tau(\pi_{\theta_k})) - J_\tau(\pi_{\theta_k},\phi_\tau(\pi_{\theta_k})))$$

$$\le -\frac{2(1-\gamma)\tau\rho_{\min}^2}{|\mathcal{S}|}\left(\min_{s,a}\pi_{\theta_k}(a\mid s)\right)^2 (J_\tau(\pi_\tau^\star,\phi_\tau^\star) - J_\tau(\pi_{\theta_k},\phi_\tau(\pi_{\theta_k}))), \tag{29}$$

where the second inequality follows from

$$J_\tau(\pi_\tau(\phi_\tau(\pi_{\theta_k})),\phi_\tau(\pi_{\theta_k})) = \max_\pi J_\tau(\pi,\phi_\tau(\pi_{\theta_k})) \ge \max_\pi \min_\phi J_\tau(\pi,\phi) = J_\tau(\pi_\tau^\star,\phi_\tau^\star).$$

From Lemma 8 Eq. (20), $-\left(\min_{s,a}\pi_{\theta_k}(a\mid s)\right)^2 \le -\frac{3c^2}{8}$, which further simplifies (29)

$$-\|\nabla_\theta J_\tau(\pi_{\theta_k},\phi_\tau(\pi_{\theta_k}))\|^2 \le -\frac{2(1-\gamma)\tau\rho_{\min}^2}{|\mathcal{S}|}\left(\min_{s,a}\pi_{\theta_k}(a\mid s)\right)^2 (J_\tau(\pi_\tau^\star,\phi_\tau^\star) - J_\tau(\pi_{\theta_k},\phi_\tau(\pi_{\theta_k})))$$

$$= -\frac{2(1-\gamma)\tau\rho_{\min}^2}{|\mathcal{S}|}\left(\min_{s,a}\pi_{\theta_k}(a\mid s)\right)^2 \delta_k^\pi \le -\frac{3(1-\gamma)\tau\rho_{\min}^2 c^2}{4|\mathcal{S}|}\delta_k^\pi.$$

For $\|\nabla_\theta J_\tau(\pi_{\theta_k},\phi_\tau(\pi_{\theta_k})) - \nabla_\theta J_\tau(\pi_{\theta_k},\phi_{\psi_k})\|^2$, we have from the $L$-smoothness of the value function derived in (18)

$$\|\nabla_\theta J_\tau(\pi_{\theta_k},\phi_\tau(\pi_{\theta_k})) - \nabla_\theta J_\tau(\pi_{\theta_k},\phi_{\psi_k})\|^2 \le L^2\|\phi_\tau(\pi_{\theta_k}) - \phi_{\psi_k}\|^2$$

$$\le \frac{2\log(2)L^2}{\tau\rho_{\min}}(J_\tau(\pi_{\theta_k},\phi_{\psi_k}) - J_\tau(\pi_{\theta_k},\phi_\tau(\pi_{\theta_k})))$$

$$= \frac{2\log(2)L^2}{\tau\rho_{\min}}\delta_k^\phi$$

Using the bound on $-\|\nabla_\theta J_\tau(\pi_{\theta_k}, \phi_\tau(\pi_{\theta_k}))\|^2$ and $\|\nabla_\theta J_\tau(\pi_{\theta_k}, \phi_\tau(\pi_{\theta_k})) - \nabla_\theta J_\tau(\pi_{\theta_k}, \phi_{\psi_k})\|^2$ in (28),

$$3\delta_{k+1}^\pi + \delta_{k+1}^\phi \le 3\delta_k^\pi + (1 - \frac{3(1-\gamma)\beta_k\tau\rho_{\min}^2 c^2}{8|\mathcal{S}|})\delta_k^\phi - \frac{\alpha_k}{8}\|\nabla_\theta J_\tau(\pi_{\theta_k}, \phi_\tau(\pi_{\theta_k}))\|^2$$

$$+ \frac{19\alpha_k}{2}\|\nabla_\theta J_\tau(\pi_{\theta_k}, \phi_\tau(\pi_{\theta_k})) - \nabla_\theta J_\tau(\pi_{\theta_k}, \phi_{\psi_k})\|^2$$

$$\le 3\delta_k^\pi + (1 - \frac{3(1-\gamma)\beta_k\tau\rho_{\min}^2 c^2}{8|\mathcal{S}|})\delta_k^\phi - \frac{3\alpha_k(1-\gamma)\tau\rho_{\min}^2 c^2}{32|\mathcal{S}|}\delta_k^\pi + \frac{19\log(2)L^2\alpha_k}{\tau\rho_{\min}}\delta_k^\phi$$

$$= 3(1 - \frac{\alpha_k(1-\gamma)\tau\rho_{\min}^2 c^2}{32|\mathcal{S}|})\delta_k^\pi + (1 - \frac{3(1-\gamma)\beta_k\tau\rho_{\min}^2 c^2}{8|\mathcal{S}|} + \frac{19\log(2)L^2\alpha_k}{\tau\rho_{\min}})\delta_k^\phi.$$

With the step sizes $\alpha_k = \alpha$, $\beta_k = \beta$ such that $\frac{\alpha}{\beta} \le \min\{\frac{(1-\gamma)\tau^2\rho_{\min}^3 c^2}{152|\mathcal{S}|\log(2)L^2}, 8\}$, we can simplify the inequality above

$$3\delta_{k+1}^\pi + \delta_{k+1}^\phi \le 3(1 - \frac{\alpha_k(1-\gamma)\tau\rho_{\min}^2 c^2}{32|\mathcal{S}|})\delta_k^\pi + (1 - \frac{3(1-\gamma)\beta_k\tau\rho_{\min}^2 c^2}{8|\mathcal{S}|} + \frac{19\log(2)L^2\alpha_k}{\tau\rho_{\min}})\delta_k^\phi$$

$$\le 3(1 - \frac{\alpha(1-\gamma)\tau\rho_{\min}^2 c^2}{32|\mathcal{S}|})\delta_k^\pi + (1 - \frac{(1-\gamma)\beta\tau\rho_{\min}^2 c^2}{4|\mathcal{S}|})\delta_k^\phi$$

$$\le (1 - \frac{\alpha(1-\gamma)\tau\rho_{\min}^2 c^2}{32|\mathcal{S}|})(3\delta_k^\pi + \delta_k^\phi)$$

$$\le (1 - \frac{\alpha(1-\gamma)\tau\rho_{\min}^2 c^2}{32|\mathcal{S}|})^{k+1}(3\delta_0^\pi + \delta_0^\phi).$$

$\square$

## A.2   Proof of Corollary 1

As a result of Lemma 3, it is easy to verify

$$(3\delta_{t+1,0}^\pi + \delta_{t+1,0}^\phi) - (3\delta_{t,K_t}^\pi + \delta_{t,K_t}^\phi)$$

$$= (3J_{\tau_{t+1}}(\pi_{\tau_{t+1}}^\star, \phi_{\tau_{t+1}}^\star) - 3J_{\tau_{t+1}}(\pi_{\theta_{t+1,0}}, \phi_{\tau_{t+1}}(\pi_{\theta_{t+1,0}}))$$

$$+ J_{\tau_{t+1}}(\pi_{\theta_{t+1,0}}, \phi_{\psi_{t+1,0}}) - J_{\tau_{t+1}}(\pi_{\theta_{t+1,0}}, \phi_{\tau_{t+1}}(\pi_{\theta_{t+1,0}})))$$

$$- (3J_{\tau_t}(\pi_{\tau_t}^\star, \phi_{\tau_t}^\star) - 3J_{\tau_t}(\pi_{\theta_{t,K_t}}, \phi_{\tau_t}(\pi_{\theta_{t,K_t}}))$$

$$+ J_{\tau_t}(\pi_{\theta_{t,K_t}}, \phi_{\psi_{t,K_t}}) - J_{\tau_t}(\pi_{\theta_{t,K_t}}, \phi_{\tau_t}(\pi_{\theta_{t,K_t}})))$$

$$= (3J_{\tau_{t+1}}(\pi_{\tau_{t+1}}^\star, \phi_{\tau_{t+1}}^\star) - 3J_{\tau_{t+1}}(\pi_{\theta_{t+1,0}}, \phi_{\tau_{t+1}}(\pi_{\theta_{t+1,0}}))$$

$$+ J_{\tau_{t+1}}(\pi_{\theta_{t+1,0}}, \phi_{\psi_{t+1,0}}) - J_{\tau_{t+1}}(\pi_{\theta_{t+1,0}}, \phi_{\tau_{t+1}}(\pi_{\theta_{t+1,0}})))$$

$$- (3J_{\tau_t}(\pi_{\tau_t}^\star, \phi_{\tau_t}^\star) - 3J_{\tau_t}(\pi_{\theta_{t+1,0}}, \phi_{\tau_t}(\pi_{\theta_{t+1,0}}))$$

$$+ J_{\tau_t}(\pi_{\theta_{t+1,0}}, \phi_{\psi_{t+1,0}}) - J_{\tau_t}(\pi_{\theta_{t+1,0}}, \phi_{\tau_t}(\pi_{\theta_{t+1,0}})))$$

$$= 3(J_{\tau_{t+1}}(\pi_{\tau_{t+1}}^\star, \phi_{\tau_{t+1}}^\star) - J_{\tau_t}(\pi_{\tau_t}^\star, \phi_{\tau_t}^\star))$$

$$- 4(J_{\tau_{t+1}}(\pi_{\theta_{t+1,0}}, \phi_{\tau_{t+1}}(\pi_{\theta_{t+1,0}})) - J_{\tau_t}(\pi_{\theta_{t+1,0}}, \phi_{\tau_t}(\pi_{\theta_{t+1,0}})))$$

$$+ (J_{\tau_{t+1}}(\pi_{\theta_{t+1,0}}, \phi_{\psi_{t+1,0}}) - J_{\tau_t}(\pi_{\theta_{t+1,0}}, \phi_{\psi_{t+1,0}}))$$

$$\le L_\delta(\tau_t - \tau_{t+1}). \tag{30}$$

We can choose $\tau_0$ large enough that

$$3\delta_{0,0}^\pi + \delta_{0,0}^\phi \le C_1\tau_0$$

holds. For any $t \ge 0$, if we run the inner loop for $K_t$ iterations such that

$$3\delta_{t,K_t}^\pi + \delta_{t,K_t}^\phi \le \frac{1}{2}(3\delta_{t,0}^\pi + \delta_{t,0}^\phi) \le \frac{C_1\tau_t}{2},$$

then we have

$$3\delta_{t+1,0}^{\pi} + \delta_{t+1,0}^{\phi} \leq 3\delta_{t,K_t}^{\pi} + \delta_{t,K_t}^{\phi} + L_\delta(\tau_t - \tau_{t+1}) \leq \frac{C_1\tau_t}{2} + L_\delta(\tau_t - \tau_{t+1})$$

$$= \frac{(C_1 + L_\delta)C_1}{C_1 + 2L_\delta}\tau_{t+1} + \frac{C_1 L_\delta}{C_1 + 2L_\delta}\tau_{t+1} = C_1\tau_{t+1},$$

where the first equality plugs in $\tau_t = \frac{2C_1 + 2L_\delta}{C_1 + 2L_\delta}\tau_{t+1}$. This means that the initial condition (11) is observed at the beginning of the every outer loop iteration.

Applying the inequality recursively,

$$3\delta_{T,0}^{\pi} + \delta_{T,0}^{\phi} \leq C_1\tau_T.$$

With an argument similar to the one in (30), we can show

$$(3(J(\pi^\star, \phi^\star) - J(\pi_{\theta_{T,0}}, \phi_0(\pi_{\theta_{T,0}}))))$$
$$+ (J(\pi_{\theta_{T,0}}, \phi_{\psi_{T,0}}) - J(\pi_{\theta_{T,0}}, \phi_0(\pi_{\theta_{T,0}})))) - (3\delta_{T,0}^{\pi} + \delta_{T,0}^{\phi}) \leq L_\delta\tau_T.$$

In order to achieve (12), it suffices to guarantee $3\delta_{T,0}^{\pi} + \delta_{T,0}^{\phi} + L_\delta\tau_T \leq \epsilon$, or $(C_1 + L_\delta)\tau_T \leq \epsilon$. This implies that we need $\tau_T = \mathcal{O}(\epsilon)$, or equivalently, $T = \mathcal{O}(\log(\epsilon^{-1}))$ since $\tau_T = \left(\frac{C_1 + 2L_\delta}{2C_1 + 2L_\delta}\right)^T \tau_0$.

Ultimately we are interested in bounding $\sum_{t=0}^{T} K_t$. Note that $K_t$ needs to be at most

$$K_t \leq \lceil \frac{\log(\frac{1}{2})}{\log(1 - \frac{\alpha_t(1-\gamma)\tau_t\rho_{\min}^2 c^2}{32|\mathcal{S}|})} \rceil.$$

To apply Theorem 1, we need to select the step sizes that satisfy the required condition. Since $\{\tau_t\}$ is a decaying sequence, the smoothness constant $L = 3L_\mathcal{H}\max\{\tau_0, 1\}$ is valid across all outer loop iterations $t$.

We use $L_t = 3L_\mathcal{H}\max\{\tau_t, 1\}$ to denote the smoothness constant of the regularized value function in outer loop iteration $t$ and use $T_1$ to denote the index of the outer loop iteration such that $\tau_{T_1} \geq 1$ and $\tau_{T_1+1} < 1$. Note that $T_1$ is an absolute constant that only depends on the structure of the Markov game. From iterations $t = 0$ to $t = T_1$, the smoothness constant is proportional to regularization weight $L_t = 3L_\mathcal{H}\max\{\tau_t, 1\} = 3L_\mathcal{H}\tau_t$. We need to choose $\alpha_t, \beta_t$ such that

$$\beta_t \leq \frac{1}{L_t} = \frac{1}{3L_\mathcal{H}\tau_t}, \quad \frac{\alpha_t}{\beta_t} \leq \min\{\frac{(1-\gamma)\rho_{\min}^3 c^2\tau_t^2}{152\log(2)|\mathcal{S}|L_t^2}, 8\} = \min\{\frac{(1-\gamma)\rho_{\min}^3 c^2}{1368\log(2)|\mathcal{S}|L_\mathcal{H}^2}, 8\},$$

$$\alpha_t \leq \min\{(L_t + \frac{2\sqrt{|\mathcal{S}|L_t^2}}{\sqrt{(1-\gamma)\rho_{\min}\tau_t}c})^{-1}, \frac{16|\mathcal{S}|}{(1-\gamma)\rho_{\min}^2 c^2\tau_t}\}$$

$$= \min\{(3L_\mathcal{H}\tau_t + \frac{18\sqrt{|\mathcal{S}|L_\mathcal{H}^2\tau_t}}{\sqrt{(1-\gamma)\rho_{\min}}c})^{-1}, \frac{16|\mathcal{S}|}{(1-\gamma)\rho_{\min}^2 c^2\tau_t}\}.$$

Then it is obvious that we can choose $\alpha_t = \mathcal{O}(\tau_t^{-1})$, implying $\alpha_t\tau_t = \mathcal{O}(1)$. Therefore, for all $0 \leq t \leq T_1$,

$$K_t \leq \lceil \frac{\log(\frac{1}{2})}{\log(1 - \frac{\alpha_t(1-\gamma)\tau_t\rho_{\min}^2 c^2}{32|\mathcal{S}|})} \rceil = \mathcal{O}(1). \tag{31}$$

From iterations $t = T_1$ until $t = T$, the smoothness constant is $L_t = 3L_\mathcal{H}\max\{\tau_t, 1\} = 3L_\mathcal{H}$. Note that there is an upper and lower bound on $\beta_t$. In order for the upper bound to be no smaller than the lower bound, we need

$$\frac{152\log(2)|\mathcal{S}|L^2\alpha_t}{(1-\gamma)\rho_{\min}^3 c^2\tau_t^2} \leq \frac{1}{L}.$$

This means that we should choose $\alpha_t = \mathcal{O}(\tau_t^2)$, implying $\alpha_t \tau_t = \mathcal{O}(\tau_t^3)$. Plugging it in (31),

$$K_t = \lceil \frac{\log(\frac{1}{2})}{\log(1 - \frac{\alpha_t(1-\gamma)\tau_t \rho_{\min}^2 c^2}{32|\mathcal{S}|})} \rceil = \mathcal{O}(\frac{1}{\log(1 - \tau_t^3)}) \leq \mathcal{O}(\tau_t^{-3}),$$

where the last inequality follows from the fact that $1 + x \leq \exp(x)$ for any scalar $x$.

Since $\tau_t = \tau_T(\frac{2C_1 + 2L_\delta}{C_1 + 2L_\delta})^{T-t}$,

$$\sum_{t=0}^{T} K_t = \sum_{t=0}^{T_1} K_t + \sum_{t=T_1}^{T} K_t \leq \sum_{t=0}^{T} \mathcal{O}(\tau_t^{-3}) = \mathcal{O}(1) + \sum_{t=T_1}^{T} \mathcal{O}(\tau_T^{-3}(\frac{2C_1 + 2L_\delta}{C_1 + 2L_\delta})^{-3(T-t)})$$

$$\leq \mathcal{O}(\tau_T^{-3} \sum_{t=0}^{T}(\frac{C_1 + 2L_\delta}{2C_1 + 2L_\delta})^{3(T-t)}) = \mathcal{O}(\tau_T^{-3} \sum_{t=0}^{T}(\frac{C_1 + 2L_\delta}{2C_1 + 2L_\delta})^{3t})$$

$$\leq \mathcal{O}(\tau_T^{-3} \frac{1}{1 - (\frac{C_1 + 2L_\delta}{2C_1 + 2L_\delta})^3}) = \mathcal{O}(\tau_T^{-3}).$$

Since $\tau_T = \mathcal{O}(\epsilon)$,

$$\sum_{t=0}^{T} K_t \leq \mathcal{O}(\tau_T^{-3}) = \mathcal{O}(\epsilon^{-3}).$$

$\square$

### A.3 Proof of Theorem 2

Define $L_0 = L_{\mathcal{H}}(2\tau_0 + 1)$. The exact conditions on the initial step sizes, regularization weight, and $h$ are

$$\delta_0^\pi + \delta_0^\phi \leq \frac{C_1 \tau_0}{h^{\frac{1}{3}}}, \tag{32}$$

$$\alpha_0 = \frac{65536 \log(2)(\log|\mathcal{A}| + \log|\mathcal{B}|) + 96(1-\gamma)\rho_{\min}c^2}{3(1-\gamma)^2 \rho_{\min}^3 c^4 \tau_0}, \tag{33}$$

$$\frac{\alpha_0}{h^{\frac{2}{3}}} \leq (2L_{\mathcal{H}} + 4L_{\mathcal{H}}^2 C_2)\frac{\tau_0}{h^{\frac{1}{3}}} + (L_{\mathcal{H}} + 4L_{\mathcal{H}}^2 C_2) + \frac{L_{\mathcal{H}}^2 C_2 h^{\frac{1}{3}}}{\tau_0}, \tag{34}$$

$$\beta_0 \leq \frac{1}{L_0}, \quad \frac{\alpha_0}{\beta_0} \leq \min\{\frac{(1-\gamma)\tau_0^2 \rho_{\min}^3 c^2}{152 \log(2)|\mathcal{S}|L_0^2}, 1\}. \tag{35}$$

In Remark 2 at the end of this section, we show that there always exist $\alpha_0, \beta_0, \tau_0$, and $h$ that observe the conditions.

(17) implies that for any $\theta_1, \theta_2 \in \mathbb{R}^{|\mathcal{S}| \times |\mathcal{A}|}$, $\psi_1, \psi_2 \in \mathbb{R}^{|\mathcal{S}| \times |\mathcal{B}|}$, and $k \geq 0$,

$$\|\nabla_\theta J_{\tau_k}(\pi_{\theta_1}, \phi_{\psi_1}) - \nabla_\theta J_{\tau_k}(\pi_{\theta_2}, \phi_{\psi_2})\| \leq (L_V + 2\tau_k L_{\mathcal{H}})(\|\theta_1 - \theta_2\| + \|\psi_1 - \psi_2\|)$$
$$\leq L_0(\|\theta_1 - \theta_2\| + \|\psi_1 - \psi_2\|), \tag{36}$$

where the last inequality follows from $\tau_k \leq \tau_0$.

**Convergence of $3\delta_k^\pi + \delta_k^\phi$:**

We will first use an induction argument to prove

$$3\delta_k^\pi + \delta_k^\phi \leq \frac{\rho_{\min}\tau_0 c^2}{64 \log(2)(k+h)^{1/3}}, \quad \forall k \geq 0.$$

The base case is $3\delta_0^\pi + \delta_0^\phi \leq \frac{\rho_{\min}c^2\tau_0}{64 \log(2)h^{\frac{1}{3}}}$, which holds by the initial condition. Now, suppose

$$3\delta_k^\pi + \delta_k^\phi \leq \frac{\rho_{\min}\tau_0 c^2}{64 \log(2)(k+h)^{1/3}} \tag{37}$$

holds. We aim to show

$$3\delta_{k+1}^{\pi} + \delta_{k+1}^{\phi} \leq \frac{\rho_{\min}\tau_0 c^2}{64\log(2)(k+1+h)^{1/3}}.$$

We introduce the following technical lemmas.

**Lemma 10.** *Suppose* (37) *holds. Then, we have*

$$-\left(\min_{s,a}\pi_{\theta_k}(a\mid s)\right)^2 \leq -\frac{3c^2}{8}, \tag{38}$$

$$-\left(\min_{s,b}\phi_{\psi_k}(b\mid s)\right)^2 \leq -\frac{3c^2}{8}. \tag{39}$$

**Lemma 11.** *Suppose* (37) *holds. Under Assumption 1 and 2 and the step sizes of Theorem 2, we have*

$$g_{\tau_k}(\theta_k) - g_{\tau_k}(\theta_{k+1})$$
$$= J_{\tau_k}(\pi_{\theta_k}, \phi_{\tau_k}(\pi_{\theta_k})) - J_{\tau_k}(\pi_{\theta_{k+1}}, \phi_{\tau_k}(\pi_{\theta_{k+1}}))$$
$$\leq \frac{\alpha_k}{2}\left(\|\nabla_\theta J_{\tau_k}(\pi_{\theta_k}, \phi_{\tau_k}(\pi_{\theta_k})) - \nabla_\theta J_{\tau_k}(\pi_{\theta_k}, \phi_{\psi_k})\|^2 - \|\nabla_\theta J_{\tau_k}(\pi_{\theta_k}, \phi_{\tau_k}(\pi_{\theta_k}))\|^2\right).$$

We perform the following decomposition

$$\delta_{k+1}^{\pi} - \delta_k^{\pi}$$
$$= J_{\tau_k}(\pi_{\theta_k}, \phi_{\tau_k}(\pi_{\theta_k})) - J_{\tau_{k+1}}(\pi_{\theta_{k+1}}, \phi_{\tau_{k+1}}(\pi_{\theta_{k+1}})) + J_{\tau_{k+1}}(\pi_{\tau_{k+1}}^{\star}, \phi_{\tau_{k+1}}^{\star}) - J_{\tau_k}(\pi_{\tau_k}^{\star}, \phi_{\tau_k}^{\star})$$
$$= J_{\tau_k}(\pi_{\theta_k}, \phi_{\tau_k}(\pi_{\theta_k})) - J_{\tau_k}(\pi_{\theta_{k+1}}, \phi_{\tau_k}(\pi_{\theta_{k+1}}))$$
$$\quad + J_{\tau_k}(\pi_{\theta_{k+1}}, \phi_{\tau_k}(\pi_{\theta_{k+1}})) - J_{\tau_k}(\pi_{\theta_{k+1}}, \phi_{\tau_{k+1}}(\pi_{\theta_{k+1}}))$$
$$\quad + J_{\tau_k}(\pi_{\theta_{k+1}}, \phi_{\tau_{k+1}}(\pi_{\theta_{k+1}})) - J_{\tau_{k+1}}(\pi_{\theta_{k+1}}, \phi_{\tau_{k+1}}(\pi_{\theta_{k+1}}))$$
$$\quad + J_{\tau_{k+1}}(\pi_{\tau_{k+1}}^{\star}, \phi_{\tau_{k+1}}^{\star}) - J_{\tau_k}(\pi_{\tau_k}^{\star}, \phi_{\tau_k}^{\star})$$
$$\leq J_{\tau_k}(\pi_{\theta_k}, \phi_{\tau_k}(\pi_{\theta_k})) - J_{\tau_k}(\pi_{\theta_{k+1}}, \phi_{\tau_k}(\pi_{\theta_{k+1}})) + \frac{\tau_k - \tau_{k+1}}{1-\gamma}\log|\mathcal{A}| + (\tau_k - \tau_{k+1})\log|\mathcal{B}|$$
$$\leq \frac{\alpha_k}{2}\left(\|\nabla_\theta J_{\tau_k}(\pi_{\theta_k}, \phi_{\tau_k}(\pi_{\theta_k})) - \nabla_\theta J_{\tau_k}(\pi_{\theta_k}, \phi_{\psi_k})\|^2 - \|\nabla_\theta J_{\tau_k}(\pi_{\theta_k}, \phi_{\tau_k}(\pi_{\theta_k}))\|^2\right)$$
$$\quad + \frac{\tau_k - \tau_{k+1}}{1-\gamma}(\log|\mathcal{A}| + \log|\mathcal{B}|) \tag{40}$$

where the first inequality comes from $J_{\tau_k}(\pi_{\theta_{k+1}}, \phi_{\tau_k}(\pi_{\theta_{k+1}})) - J_{\tau_k}(\pi_{\theta_{k+1}}, \phi_{\tau_{k+1}}(\pi_{\theta_{k+1}})) \leq 0$ by the definition of $\phi_\tau(\cdot)$ and the bound on $J_{\tau_k}(\pi_{\theta_{k+1}}, \phi_{\tau_{k+1}}(\pi_{\theta_{k+1}})) - J_{\tau_{k+1}}(\pi_{\theta_{k+1}}, \phi_{\tau_{k+1}}(\pi_{\theta_{k+1}}))$ and $J_{\tau_{k+1}}(\pi_{\tau_{k+1}}^{\star}, \phi_{\tau_{k+1}}^{\star}) - J_{\tau_k}(\pi_{\tau_k}^{\star}, \phi_{\tau_k}^{\star})$ from Lemma 3 Eqs. (8) and (6). The second inequality uses Lemma 11.

Similarly, we consider the decay of $\delta_k^{\phi}$.

$$\delta_{k+1}^{\phi} - \delta_k^{\phi} = J_{\tau_{k+1}}(\pi_{\theta_{k+1}}, \phi_{\psi_{k+1}}) - g_{\tau_{k+1}}(\pi_{\theta_{k+1}}) - J_{\tau_k}(\pi_{\theta_k}, \phi_{\psi_k}) + g_{\tau_k}(\pi_{\theta_k})$$
$$= \left(J_{\tau_{k+1}}(\pi_{\theta_{k+1}}, \phi_{\psi_{k+1}}) - J_{\tau_k}(\pi_{\theta_{k+1}}, \phi_{\psi_{k+1}})\right) + \left(J_{\tau_k}(\pi_{\theta_{k+1}}, \phi_{\psi_{k+1}}) - J_{\tau_k}(\pi_{\theta_{k+1}}, \phi_{\psi_k})\right)$$
$$\quad + \left(J_{\tau_k}(\pi_{\theta_{k+1}}, \phi_{\psi_k}) - J_{\tau_k}(\pi_{\theta_k}, \phi_{\psi_k})\right) + \left(g_{\tau_k}(\pi_{\theta_k}) - g_{\tau_{k+1}}(\pi_{\theta_{k+1}})\right). \tag{41}$$

By Lemma 3 Eq. (8),

$$J_{\tau_{k+1}}(\pi_{\theta_{k+1}}, \phi_{\psi_{k+1}}) - J_{\tau_k}(\pi_{\theta_{k+1}}, \phi_{\psi_{k+1}}) \leq \frac{\tau_k - \tau_{k+1}}{1-\gamma}\log|\mathcal{B}|. \tag{42}$$

Using the $L_0$-smoothness of the value function derived in (36)

$$J_{\tau_k}(\pi_{\theta_{k+1}}, \phi_{\psi_{k+1}}) - J_{\tau_k}(\pi_{\theta_{k+1}}, \phi_{\psi_k})$$
$$\leq \langle \nabla_\psi J_{\tau_k}(\pi_{\theta_{k+1}}, \phi_{\psi_k}), \psi_{k+1} - \psi_k \rangle + \frac{L_0}{2}\|\psi_{k+1} - \psi_k\|^2$$
$$= -\beta_k\|\nabla_\psi J_{\tau_k}(\pi_{\theta_{k+1}}, \phi_{\psi_k})\|^2 + \frac{L_0\beta_k^2}{2}\|\nabla_\psi J_{\tau_k}(\pi_{\theta_{k+1}}, \phi_{\psi_k})\|^2$$

$$\leq -\frac{\beta_k}{2}\|\nabla_\psi J_{\tau_k}(\pi_{\theta_{k+1}}, \phi_{\psi_k})\|^2$$

$$\leq -\frac{(1-\gamma)\beta_k\tau_k\rho_{\min}^2}{|\mathcal{S}|}\left(\min_{s,b}\phi_{\psi_k}(b\mid s)\right)^2(J_{\tau_k}(\pi_{\theta_k}, \phi_{\psi_k}) - J_{\tau_k}(\pi_{\theta_k}, \phi_{\tau_k}(\pi_{\theta_k})))$$

$$= -\frac{(1-\gamma)\beta_k\tau_k\rho_{\min}^2}{|\mathcal{S}|}\left(\min_{s,b}\phi_{\psi_k}(b\mid s)\right)^2\delta_k^\phi,$$

where the second inequality uses $\beta_k \leq \frac{1}{L_0}$ and the third inequality follows from Lemma 4.

Using Eq. (39) of Lemma 10 to further simplify this inequality,

$$J_{\tau_k}(\pi_{\theta_{k+1}}, \phi_{\psi_{k+1}}) - J_{\tau_k}(\pi_{\theta_{k+1}}, \phi_{\psi_k}) \leq -\frac{3(1-\gamma)\beta_k\tau_k\rho_{\min}^2 c^2}{8|\mathcal{S}|}\delta_k^\phi. \tag{43}$$

For the third term of (41), we have from the $L_0$-smoothness of the value function derived in (36)

$$J_{\tau_k}(\pi_{\theta_{k+1}}, \phi_{\psi_k}) - J_{\tau_k}(\pi_{\theta_k}, \phi_{\psi_k}) \leq \langle\nabla_\theta J_{\tau_k}(\pi_{\theta_k}, \phi_{\psi_k}), \theta_{k+1}-\theta_k\rangle + \frac{L_0}{2}\|\theta_{k+1}-\theta_k\|^2$$

$$= \alpha_k\|\nabla_\theta J_{\tau_k}(\pi_{\theta_k}, \phi_{\psi_k})\|^2 + \frac{L_0\alpha_k^2}{2}\|\nabla_\theta J_{\tau_k}(\pi_{\theta_k}, \phi_{\psi_k})\|^2$$

$$\leq \frac{3\alpha_k}{2}\|\nabla_\theta J_{\tau_k}(\pi_{\theta_k}, \phi_{\psi_k})\|^2, \tag{44}$$

where in the last inequality we use $\alpha_k L_0 \leq 1$.

Using Lemma 11 and Lemma 3 (7), we bound the last term of (41)

$$g_{\tau_k}(\pi_{\theta_k}) - g_{\tau_{k+1}}(\pi_{\theta_{k+1}})$$
$$= g_{\tau_k}(\pi_{\theta_k}) - g_{\tau_k}(\pi_{\theta_{k+1}}) + g_{\tau_k}(\pi_{\theta_{k+1}}) - g_{\tau_{k+1}}(\pi_{\theta_{k+1}})$$
$$\leq \frac{\alpha_k}{2}\left(\|\nabla_\theta J_{\tau_k}(\pi_{\theta_k}, \phi_{\tau_k}(\pi_{\theta_k})) - \nabla_\theta J_{\tau_k}(\pi_{\theta_k}, \phi_{\psi_k})\|^2 - \|\nabla_\theta J_{\tau_k}(\pi_{\theta_k}, \phi_{\tau_k}(\pi_{\theta_k}))\|^2\right)$$
$$\quad + (\tau_k - \tau_{k+1})\log|\mathcal{A}| \tag{45}$$

Using (42)-(45) in (41), we have

$$\delta_{k+1}^\phi = \delta_k^\phi + \left(J_{\tau_{k+1}}(\pi_{\theta_{k+1}}, \phi_{\psi_{k+1}}) - J_{\tau_k}(\pi_{\theta_{k+1}}, \phi_{\psi_{k+1}})\right) + \left(J_{\tau_k}(\pi_{\theta_{k+1}}, \phi_{\psi_{k+1}}) - J_{\tau_k}(\pi_{\theta_{k+1}}, \phi_{\psi_k})\right)$$
$$\quad + \left(J_{\tau_k}(\pi_{\theta_{k+1}}, \phi_{\psi_k}) - J_{\tau_k}(\pi_{\theta_k}, \phi_{\psi_k})\right) + \left(g_{\tau_k}(\pi_{\theta_k}) - g_{\tau_{k+1}}(\pi_{\theta_{k+1}})\right)$$
$$\leq \delta_k^\phi + \frac{\tau_k - \tau_{k+1}}{1-\gamma}\log|\mathcal{B}| - \frac{3(1-\gamma)\beta_k\tau_k\rho_{\min}^2 c^2}{8|\mathcal{S}|}\delta_k^\phi + \frac{3\alpha_k}{2}\|\nabla_\theta J_{\tau_k}(\pi_{\theta_k}, \phi_{\psi_k})\|^2$$
$$\quad + \frac{\alpha_k}{2}\left(\|\nabla_\theta J_{\tau_k}(\pi_{\theta_k}, \phi_{\tau_k}(\pi_{\theta_k})) - \nabla_\theta J_{\tau_k}(\pi_{\theta_k}, \phi_{\psi_k})\|^2 - \|\nabla_\theta J_{\tau_k}(\pi_{\theta_k}, \phi_{\tau_k}(\pi_{\theta_k}))\|^2\right)$$
$$\quad + (\tau_k - \tau_{k+1})\log|\mathcal{A}|$$
$$\leq (1 - \frac{3(1-\gamma)\beta_k\tau_k\rho_{\min}^2 c^2}{8|\mathcal{S}|})\delta_k^\phi + \frac{3\alpha_k}{2}\|\nabla_\theta J_{\tau_k}(\pi_{\theta_k}, \phi_{\psi_k})\|^2$$
$$\quad + \frac{\alpha_k}{2}\left(\|\nabla_\theta J_{\tau_k}(\pi_{\theta_k}, \phi_{\tau_k}(\pi_{\theta_k})) - \nabla_\theta J_{\tau_k}(\pi_{\theta_k}, \phi_{\psi_k})\|^2 - \|\nabla_\theta J_{\tau_k}(\pi_{\theta_k}, \phi_{\tau_k}(\pi_{\theta_k}))\|^2\right)$$
$$\quad + \frac{\tau_k - \tau_{k+1}}{1-\gamma}(\log|\mathcal{A}| + \log|\mathcal{B}|). \tag{46}$$

Combining (40) and (46),

$$3\delta_{k+1}^\pi + \delta_{k+1}^\phi$$
$$\leq 3\delta_k^\pi + \frac{3\alpha_k}{2}\left(\|\nabla_\theta J_{\tau_k}(\pi_{\theta_k}, \phi_{\tau_k}(\pi_{\theta_k})) - \nabla_\theta J_{\tau_k}(\pi_{\theta_k}, \phi_{\psi_k})\|^2 - \|\nabla_\theta J_{\tau_k}(\pi_{\theta_k}, \phi_{\tau_k}(\pi_{\theta_k}))\|^2\right)$$
$$\quad + \frac{3(\tau_k - \tau_{k+1})}{1-\gamma}(\log|\mathcal{A}| + \log|\mathcal{B}|) + (1 - \frac{3(1-\gamma)\beta_k\tau_k\rho_{\min}^2 c^2}{8|\mathcal{S}|})\delta_k^\phi$$

$$+ \frac{\alpha_k}{2} \left( \|\nabla_\theta J_{\tau_k}(\pi_{\theta_k}, \phi_{\tau_k}(\pi_{\theta_k})) - \nabla_\theta J_{\tau_k}(\pi_{\theta_k}, \phi_{\psi_k})\|^2 - \|\nabla_\theta J_{\tau_k}(\pi_{\theta_k}, \phi_{\tau_k}(\pi_{\theta_k}))\|^2 \right)$$

$$+ \frac{3\alpha_k}{2} \|\nabla_\theta J_{\tau_k}(\pi_{\theta_k}, \phi_{\psi_k})\|^2 + \frac{\tau_k - \tau_{k+1}}{1-\gamma} (\log |\mathcal{A}| + \log |\mathcal{B}|)$$

$$\leq 3\delta_k^\pi + (1 - \frac{3(1-\gamma)\beta_k \tau_k \rho_{\min}^2 c^2}{8|\mathcal{S}|})\delta_k^\phi + \frac{3\alpha_k}{2} \|\nabla_\theta J_{\tau_k}(\pi_{\theta_k}, \phi_{\psi_k})\|^2$$

$$+ 2\alpha_k \|\nabla_\theta J_{\tau_k}(\pi_{\theta_k}, \phi_{\tau_k}(\pi_{\theta_k})) - \nabla_\theta J_{\tau_k}(\pi_{\theta_k}, \phi_{\psi_k})\|^2 - 2\alpha_k \|\nabla_\theta J_{\tau_k}(\pi_{\theta_k}, \phi_{\tau_k}(\pi_{\theta_k}))\|^2$$

$$+ \frac{4(\tau_k - \tau_{k+1})}{1-\gamma} (\log |\mathcal{A}| + \log |\mathcal{B}|).$$

Simplifying this inequality with

$$\|\nabla_\theta J_{\tau_k}(\pi_{\theta_k}, \phi_{\psi_k})\|^2 = \|\nabla_\theta J_{\tau_k}(\pi_{\theta_k}, \phi_{\tau_k}(\pi_{\theta_k})) - (\nabla_\theta J_{\tau_k}(\pi_{\theta_k}, \phi_{\tau_k}(\pi_{\theta_k})) - \nabla_\theta J_{\tau_k}(\pi_{\theta_k}, \phi_{\psi_k}))\|^2$$

$$\leq \|\nabla_\theta J_{\tau_k}(\pi_{\theta_k}, \phi_{\tau_k}(\pi_{\theta_k}))\|^2 + \|\nabla_\theta J_{\tau_k}(\pi_{\theta_k}, \phi_{\tau_k}(\pi_{\theta_k})) - \nabla_\theta J_{\tau_k}(\pi_{\theta_k}, \phi_{\psi_k})\|^2$$

$$+ 2\langle \nabla_\theta J_{\tau_k}(\pi_{\theta_k}, \phi_{\tau_k}(\pi_{\theta_k})), \nabla_\theta J_{\tau_k}(\pi_{\theta_k}, \phi_{\tau_k}(\pi_{\theta_k})) - \nabla_\theta J_{\tau_k}(\pi_{\theta_k}, \phi_{\psi_k})\rangle$$

$$\leq \frac{5}{4} \|\nabla_\theta J_{\tau_k}(\pi_{\theta_k}, \phi_{\tau_k}(\pi_{\theta_k}))\|^2 + 5\|\nabla_\theta J_{\tau_k}(\pi_{\theta_k}, \phi_{\tau_k}(\pi_{\theta_k})) - \nabla_\theta J_{\tau_k}(\pi_{\theta_k}, \phi_{\psi_k})\|^2$$

we have

$$3\delta_{k+1}^\pi + \delta_{k+1}^\phi \leq 3\delta_k^\pi + (1 - \frac{3(1-\gamma)\beta_k \tau_k \rho_{\min}^2 c^2}{8|\mathcal{S}|})\delta_k^\phi - \frac{\alpha_k}{8} \|\nabla_\theta J_{\tau_k}(\pi_{\theta_k}, \phi_{\tau_k}(\pi_{\theta_k}))\|^2$$

$$+ \frac{19\alpha_k}{2} \|\nabla_\theta J_{\tau_k}(\pi_{\theta_k}, \phi_{\tau_k}(\pi_{\theta_k})) - \nabla_\theta J_{\tau_k}(\pi_{\theta_k}, \phi_{\psi_k})\|^2$$

$$+ \frac{4(\tau_k - \tau_{k+1})}{1-\gamma} (\log |\mathcal{A}| + \log |\mathcal{B}|) \tag{47}$$

Using Lemma 4 to bound $-\|\nabla_\theta J_{\tau_k}(\pi_{\theta_k}, \phi_{\tau_k}(\pi_{\theta_k}))\|^2$,

$$- \|\nabla_\theta J_{\tau_k}(\pi_{\theta_k}, \phi_{\tau_k}(\pi_{\theta_k}))\|^2$$

$$\leq -\frac{2(1-\gamma)\tau_k \rho_{\min}^2}{|\mathcal{S}|} \left( \min_{s,a} \pi_{\theta_k}(a \mid s) \right)^2 (J_{\tau_k}(\pi_{\tau_k}(\phi_{\tau_k}(\pi_{\theta_k})), \phi_{\tau_k}(\pi_{\theta_k})) - J_{\tau_k}(\pi_{\theta_k}, \phi_{\tau_k}(\pi_{\theta_k})))$$

$$\leq -\frac{2(1-\gamma)\tau_k \rho_{\min}^2}{|\mathcal{S}|} \left( \min_{s,a} \pi_{\theta_k}(a \mid s) \right)^2 (J_{\tau_k}(\pi_{\tau_k}^\star, \phi_{\tau_k}^\star) - J_{\tau_k}(\pi_{\theta_k}, \phi_{\tau_k}(\pi_{\theta_k}))) , \tag{48}$$

where the second inequality follows from

$$J_{\tau_k}(\pi_{\tau_k}(\phi_{\tau_k}(\pi_{\theta_k})), \phi_{\tau_k}(\pi_{\theta_k})) = \max_\pi J_{\tau_k}(\pi, \phi_{\tau_k}(\pi_{\theta_k})) \geq \max_\pi \min_\phi J_{\tau_k}(\pi, \phi) = J_{\tau_k}(\pi_{\tau_k}^\star, \phi_{\tau_k}^\star).$$

From Lemma 10 Eq. (38), $-\left( \min_{s,a} \pi_{\theta_k}(a \mid s) \right)^2 \leq -\frac{3c^2}{8}$, which further simplifies (48)

$$- \|\nabla_\theta J_{\tau_k}(\pi_{\theta_k}, \phi_{\tau_k}(\pi_{\theta_k}))\|^2$$

$$\leq -\frac{2(1-\gamma)\tau_k \rho_{\min}^2}{|\mathcal{S}|} \left( \min_{s,a} \pi_{\theta_k}(a \mid s) \right)^2 (J_{\tau_k}(\pi_{\tau_k}^\star, \phi_{\tau_k}^\star) - J_{\tau_k}(\pi_{\theta_k}, \phi_{\tau_k}(\pi_{\theta_k})))$$

$$= -\frac{2(1-\gamma)\tau_k \rho_{\min}^2}{|\mathcal{S}|} \left( \min_{s,a} \pi_{\theta_k}(a \mid s) \right)^2 \delta_k^\pi \leq -\frac{3(1-\gamma)\tau_k \rho_{\min}^2 c^2}{4|\mathcal{S}|} \delta_k^\pi. \tag{49}$$

For $\|\nabla_\theta J_{\tau_k}(\pi_{\theta_k}, \phi_{\tau_k}(\pi_{\theta_k})) - \nabla_\theta J_{\tau_k}(\pi_{\theta_k}, \phi_{\psi_k})\|^2$, we have from the $L_0$-smoothness of the value function derived in (36)

$$\|\nabla_\theta J_{\tau_k}(\pi_{\theta_k}, \phi_{\tau_k}(\pi_{\theta_k})) - \nabla_\theta J_{\tau_k}(\pi_{\theta_k}, \phi_{\psi_k})\|^2 \leq L_0^2 \|\phi_{\tau_k}(\pi_{\theta_k}) - \phi_{\psi_k}\|^2$$

$$\leq \frac{2\log(2)L_0^2}{\tau_k \rho_{\min}} (J_{\tau_k}(\pi_{\theta_k}, \phi_{\psi_k}) - J_{\tau_k}(\pi_{\theta_k}, \phi_{\tau_k}(\pi_{\theta_k})))$$

$$= \frac{2\log(2)L_0^2}{\tau_k \rho_{\min}} \delta_k^\phi, \tag{50}$$

where the second inequality follows from Lemma 1 Eq. (4).

Using (49) and (50) in (47),

$$3\delta_{k+1}^{\pi} + \delta_{k+1}^{\phi}$$

$$\leq 3\delta_k^{\pi} + (1 - \frac{3(1-\gamma)\beta_k\tau_k\rho_{\min}^2 c^2}{8|\mathcal{S}|})\delta_k^{\phi} - \frac{\alpha_k}{8}\|\nabla_\theta J_{\tau_k}(\pi_{\theta_k}, \phi_{\tau_k}(\pi_{\theta_k}))\|^2$$

$$+ \frac{19\alpha_k}{2}\|\nabla_\theta J_{\tau_k}(\pi_{\theta_k}, \phi_{\tau_k}(\pi_{\theta_k})) - \nabla_\theta J_{\tau_k}(\pi_{\theta_k}, \phi_{\psi_k})\|^2 + \frac{4(\tau_k - \tau_{k+1})}{1-\gamma}(\log|\mathcal{A}| + \log|\mathcal{B}|)$$

$$\leq 3\delta_k^{\pi} + (1 - \frac{3(1-\gamma)\beta_k\tau_k\rho_{\min}^2 c^2}{8|\mathcal{S}|})\delta_k^{\phi} - \frac{3\alpha_k(1-\gamma)\tau_k\rho_{\min}^2 c^2}{32|\mathcal{S}|}\delta_k^{\pi}$$

$$+ \frac{19\log(2)L_0^2\alpha_k}{\tau_k\rho_{\min}}\delta_k^{\phi} + \frac{4(\tau_k - \tau_{k+1}))}{1-\gamma}(\log|\mathcal{A}| + \log|\mathcal{B}|)$$

$$= 3(1 - \frac{(1-\gamma)\alpha_k\tau_k\rho_{\min}^2 c^2}{32|\mathcal{S}|})\delta_k^{\pi} + (1 - \frac{3(1-\gamma)\beta_k\tau_k\rho_{\min}^2 c^2}{8|\mathcal{S}|} + \frac{19\log(2)L_0^2\alpha_k}{\tau_k\rho_{\min}})\delta_k^{\phi}$$

$$+ \frac{4(\tau_k - \tau_{k+1})}{1-\gamma}(\log|\mathcal{A}| + \log|\mathcal{B}|). \tag{51}$$

With the step size rule $\frac{\alpha_0}{\beta_0} \leq \min\{\frac{(1-\gamma)\tau_0^2\rho_{\min}^3 c^2}{152\log(2)L_0^2|\mathcal{S}|}, 1\}$, we can simplify (51),

$$3\delta_{k+1}^{\pi} + \delta_{k+1}^{\phi} \leq 3(1 - \frac{(1-\gamma)\alpha_k\tau_k\rho_{\min}^2 c^2}{32|\mathcal{S}|})\delta_k^{\pi} + (1 - \frac{3(1-\gamma)\beta_k\tau_k\rho_{\min}^2 c^2}{8|\mathcal{S}|} + \frac{19\log(2)L_0^2\alpha_k}{\tau_k\rho_{\min}})\delta_k^{\phi}$$

$$+ \frac{4(\tau_k - \tau_{k+1})}{1-\gamma}(\log|\mathcal{A}| + \log|\mathcal{B}|)$$

$$\leq 3(1 - \frac{(1-\gamma)\alpha_k\tau_k\rho_{\min}^2 c^2}{32|\mathcal{S}|})\delta_k^{\pi}$$

$$+ (1 - \frac{3(1-\gamma)\beta_k\tau_k\rho_{\min}^2 c^2}{8|\mathcal{S}|} + \frac{19\log(2)L_0^2}{\tau_k\rho_{\min}}\frac{(1-\gamma)\rho_{\min}^3 c^2\tau_k^2\beta_k}{152\log(2)L_0^2|\mathcal{S}|})\delta_k^{\phi}$$

$$+ \frac{4(\tau_k - \tau_{k+1})}{1-\gamma}(\log|\mathcal{A}| + \log|\mathcal{B}|)$$

$$\leq 3(1 - \frac{(1-\gamma)\alpha_k\tau_k\rho_{\min}^2 c^2}{32|\mathcal{S}|})\delta_k^{\pi} + (1 - \frac{(1-\gamma)\beta_k\tau_k\rho_{\min}^2 c^2}{4|\mathcal{S}|})\delta_k^{\phi}$$

$$+ \frac{4(\tau_k - \tau_{k+1})}{1-\gamma}(\log|\mathcal{A}| + \log|\mathcal{B}|)$$

$$\leq (1 - \frac{(1-\gamma)\alpha_k\tau_k\rho_{\min}^2 c^2}{32|\mathcal{S}|})(3\delta_k^{\pi} + \delta_k^{\phi}) + \frac{4(\tau_k - \tau_{k+1})}{1-\gamma}(\log|\mathcal{A}| + \log|\mathcal{B}|)$$

$$\leq (1 - \frac{(1-\gamma)\rho_{\min}^2 c^2\alpha_0\tau_0}{32|\mathcal{S}|(k+h)})\frac{C_1}{(k+h)^{1/3}} + \frac{32\tau_0}{3(1-\gamma)(k+h)^{4/3}}(\log|\mathcal{A}| + \log|\mathcal{B}|),$$

where the last inequality follows from (37) and Lemma 7.

Letting $D_1 = \frac{(1-\gamma)\rho_{\min}^2 c^2}{32|\mathcal{S}|}$ and $D_2 = \frac{32}{3(1-\gamma)}(\log|\mathcal{A}| + \log|\mathcal{B}|)$,

$$3\delta_{k+1}^{\pi} + \delta_{k+1}^{\phi} \leq \left(1 - \frac{D_1\alpha_0\tau_0}{k+h}\right)\frac{C_1\tau_0}{(k+h)^{1/3}} + \frac{D_2\tau_0}{(k+1)^{4/3}}$$

$$= \left(k + h - D_1\alpha_0\tau_0 + \frac{D_2}{C_1}\right)\frac{C_1\tau_0}{(k+h)^{4/3}}.$$

By requiring

$$\tau_0 = \frac{65536\log(2)(\log|\mathcal{A}| + \log|\mathcal{B}|) + 96(1-\gamma)\rho_{\min}c^2}{3(1-\gamma)^2\rho_{\min}^3 c^4\alpha_0} = \frac{1}{D_1\alpha_0}(1 + \frac{D_2}{C_1}),$$

we have

$$3\delta_{k+1}^{\pi} + \delta_{k+1}^{\phi} \leq \left(k + h - D_1\alpha_0\tau_0 + \frac{D_2}{C_1}\right) \cdot \frac{C_1\tau_0}{(k+h)^{4/3}}$$

$$= \left(k + h - (1 + \frac{D_2}{C_1}) + \frac{D_2}{C_1}\right) \cdot \frac{C_1\tau_0}{(k+h)^{4/3}}$$

$$= \frac{C_1\tau_0(k - 1 + h)}{(k+h)^{4/3}},$$

Since $(k - 1 + h)^3(k + 1 + h) \leq (k + h)^4$ for all $k \geq 0$ and $h \geq 1$, we have

$$\frac{k - 1 + h}{(k+h)^{4/3}} = \frac{(k - 1 + h)(k + 1 + h)^{1/3}}{(k+1)^{4/3}(k + 1 + h)^{1/3}} \leq \frac{(k+h)^{4/3}}{(k+h)^{4/3}(k + 1 + h)^{1/3}} = \frac{1}{(k + 1 + h)^{1/3}},$$

which leads to

$$3\delta_{k+1}^{\pi} + \delta_{k+1}^{\phi} \leq \frac{C_1\tau_0(k - 1 + h)}{(k+h)^{4/3}} \leq \frac{C_1\tau_0}{(k + 1 + h)^{1/3}} = \frac{\rho_{\min}\tau_0 c^2}{64\log(2)(k + 1 + h)^{1/3}}.$$

This finishes our induction and implies that for all $k \geq 0$

$$J_{\tau_k}(\pi_{\tau_k}^{\star}, \phi_{\tau_k}^{\star}) - J_{\tau_k}(\pi_{\theta_k}, \phi_{\tau_k}(\pi_{\theta_k})) \leq \frac{C_1\tau_0}{3(k+h)^{1/3}},$$

$$J_{\tau_k}(\pi_{\theta_k}, \phi_{\psi_k}) - J_{\tau_k}(\pi_{\theta_k}, \phi_{\tau_k}(\pi_{\theta_k})) \leq \frac{C_1\tau_0}{(k+h)^{1/3}}.$$

**Bounding the difference between value functions with and without the regularization:**

Ultimately, we are interested in $J(\pi^{\star}, \phi^{\star}) - J(\pi_{\theta_k}, \phi_0(\pi_{\theta_k}))$ and $J(\pi_{\theta_k}, \phi_{\psi_k}) - J(\pi_{\theta_k}, \phi_0(\pi_{\theta_k}))$, which measure the performance of $\pi_{\theta_k}$ and $\phi_{\psi_k}$ in the original un-regularized Markov game.

By Lemma 3 Eq. (6), (7), and (8),

$$J_{\tau_k}(\pi_{\tau_k}^{\star}, \phi_{\tau_k}^{\star}) - J(\pi^{\star}, \phi^{\star}) \geq -\tau_k \log|\mathcal{B}|$$

$$J_{\tau_k}(\pi_{\theta_k}, \phi_{\tau_k}(\pi_{\theta_k})) - J(\pi_{\theta_k}, \phi_0(\pi_{\theta_k})) \leq \tau_k \log|\mathcal{A}|$$

$$J_{\tau_k}(\pi_{\theta_k}, \phi_{\psi_k}) - J(\pi_{\theta_k}, \phi_{\psi_k}) \geq -\frac{\tau_k}{1 - \gamma}\log|\mathcal{B}|.$$

Therefore,

$$J(\pi^{\star}, \phi^{\star}) - J(\pi_{\theta_k}, \phi_0(\pi_{\theta_k})) = J(\pi^{\star}, \phi^{\star}) - J_{\tau_k}(\pi_{\tau_k}^{\star}, \phi_{\tau_k}^{\star}) + J_{\tau_k}(\pi_{\tau_k}^{\star}, \phi_{\tau_k}^{\star}) - J_{\tau_k}(\pi_{\theta_k}, \phi_{\tau_k}(\pi_{\theta_k}))$$

$$+ J_{\tau_k}(\pi_{\theta_k}, \phi_{\tau_k}(\pi_{\theta_k})) - J(\pi_{\theta_k}, \phi_0(\pi_{\theta_k}))$$

$$\leq \tau_k \log|\mathcal{B}| + \frac{C_1\tau_0}{3(k+h)^{1/3}} + \tau_k \log|\mathcal{A}|$$

$$= \frac{C_1\tau_0 + 3(\log|\mathcal{A}| + \log|\mathcal{B}|)\tau_0}{3(k+h)^{1/3}},$$

and

$$J(\pi_{\theta_k}, \phi_{\psi_k}) - J(\pi_{\theta_k}, \phi_0(\pi_{\theta_k})) = J(\pi_{\theta_k}, \phi_{\psi_k}) - J_{\tau_k}(\pi_{\theta_k}, \phi_{\psi_k})$$

$$+ J_{\tau_k}(\pi_{\theta_k}, \phi_{\psi_k}) - J_{\tau_k}(\pi_{\theta_k}, \phi_{\tau_k}(\pi_{\theta_k}))$$

$$+ J_{\tau_k}(\pi_{\theta_k}, \phi_{\tau_k}(\pi_{\theta_k})) - J(\pi_{\theta_k}, \phi_0(\pi_{\theta_k}))$$

$$\leq \frac{\tau_k}{1 - \gamma}\log|\mathcal{B}| + \frac{C_1\tau_0}{(k+h)^{1/3}} + \tau_k \log|\mathcal{A}|$$

$$\leq \frac{(1 - \gamma)C_1\tau_0 + (\log|\mathcal{A}| + \log|\mathcal{B}|)\tau_0}{(1 - \gamma)(k+h)^{1/3}}.$$

**Remark 2.** *To select $\alpha_0$, $\beta_0$, $\tau_0$, and $h$, we first make $\tau_0 = \lambda h^{1/3}$ for some $\lambda > 0$ large enough. This choice guarantees the validity of (32) (we just need $\delta_0^\pi + \delta_0^\phi \le C_1 \lambda$). Viewing (33), it means*

$$\alpha_0 = \frac{65536 \log(2)(\log|\mathcal{A}| + \log|\mathcal{B}|) + 96(1-\gamma)\rho_{\min}c^2}{3(1-\gamma)^2 \rho_{\min}^3 c^4 \lambda h^{\frac{1}{3}}}.$$

*Now that $\lambda$ is fixed, to ensure (34), we choose $h$ large enough to observe*

$$\frac{65536 \log(2)(\log|\mathcal{A}| + \log|\mathcal{B}|) + 96(1-\gamma)\rho_{\min}c^2}{3(1-\gamma)^2 \rho_{\min}^3 c^4 \lambda h} = \frac{\alpha_0}{h^{\frac{2}{3}}}$$

$$\le (2L_\mathcal{H} + 4L_\mathcal{H}^2 C_2)\lambda + (L_\mathcal{H} + 4L_\mathcal{H}^2 C_2) + \frac{L_\mathcal{H}^2 C_2}{\lambda}.$$

*Once $\lambda$ and $h$ are chosen, $\alpha_0$, $\tau_0$, and $h$ are determined. Finally, since $\frac{(1-\gamma)\tau_0^2 \rho_{\min}^3 c^2}{152 \log(2)|\mathcal{S}|L_0^2} \le 1$, we just need to select $\beta_0 \in [\frac{152 \log(2)|\mathcal{S}|L_0^2 \alpha_0}{(1-\gamma)\tau_0^2 \rho_{\min}^3 c^2}, \frac{1}{L_0}]$. Recall that $L_0 = L_\mathcal{H}(2\tau_0 + 1)$, it can be easily seen that the lower bound $\frac{152 \log(2)|\mathcal{S}|L_0^2 \alpha_0}{(1-\gamma)\tau_0^2 \rho_{\min}^3 c^2} = \mathcal{O}(\frac{1}{\lambda^3 h^{1/3}})$, which is much smaller than the upper bound $\frac{1}{L_0} = \mathcal{O}(\frac{1}{\tau_0}) = \mathcal{O}(\frac{1}{\lambda h^{1/3}})$ since $\lambda$ was large enough.*

$\square$

# B  Proof of Lemmas

## B.1  Proof of Lemma 1

For a given $\phi$, let $\hat\pi \in \pi_\tau(\phi)$ (which is a possibly non-unique maximizer).

According to Mei et al. [2020][Lemma 26],

$$J_\tau(\hat\pi, \phi) - J_\tau(\pi, \phi) = \frac{\tau}{1-\gamma} \sum_{s \in \mathcal{S}} d_\rho^{\pi,\phi}(s) D_{KL}(\pi(\cdot \mid s) \| \hat\pi(\cdot \mid s)).$$

The Pinsker's inequality states that for any two probability distributions $p_1$ and $p_2$

$$D_{KL}(p_1 \| p_2) \ge \frac{1}{2 \log(2)} \| p_1 - p_2 \|_1^2.$$

Using this inequality,

$$\begin{aligned}
J_\tau(\hat\pi, \phi) - J_\tau(\pi, \phi) &= \frac{\tau}{1-\gamma} \sum_{s \in \mathcal{S}} d_\rho^{\pi,\phi}(s) D_{KL}(\pi(\cdot \mid s) \| \hat\pi(\cdot \mid s)) \\
&\ge \frac{\tau}{2 \log(2)(1-\gamma)} \sum_{s \in \mathcal{S}} d_\rho^{\pi,\phi}(s) \| \pi(\cdot \mid s) - \hat\pi(\cdot \mid s) \|_1^2 \\
&\ge \frac{\tau}{2 \log(2)(1-\gamma)} \sum_{s \in \mathcal{S}} (1-\gamma)\rho(s) \| \pi(\cdot \mid s) - \hat\pi(\cdot \mid s) \|_1^2 \\
&\ge \frac{\tau \min_{s \in \mathcal{S}} \rho(s)}{2 \log(2)} \sum_{s \in \mathcal{S}} \| \pi(\cdot \mid s) - \hat\pi(\cdot \mid s) \|_1^2 \\
&\ge \frac{\tau \min_{s \in \mathcal{S}} \rho(s)}{2 \log(2)} \| \pi - \hat\pi \|^2,
\end{aligned}$$

where the second inequality follows from the fact that $d_\rho^{\pi,\hat\phi}(s) \ge (1-\gamma)\rho(s)$ entry-wise. This inequality means that $\hat\pi \in \pi_\tau(\phi)$ has to be unique, as no other policy can achieve the same value function.

The same argument can be used to show Eq. (4).

$\square$

## B.2 Proof of Lemma 2

Let $(\pi_1, \phi_1)$, $(\pi_2, \phi_2)$ be optimal solution pairs to the maximin and minimax problem, respectively,

$$(\pi_1, \phi_1) \in \operatorname*{argmax}_{\pi \in \Delta_{\mathcal{A}}^{\mathcal{S}}} \operatorname*{argmin}_{\phi \in \Delta_{\mathcal{B}}^{\mathcal{S}}} J_\tau(\pi, \phi) \quad \text{and} \quad (\pi_2, \phi_2) \in \operatorname*{argmin}_{\phi \in \Delta_{\mathcal{B}}^{\mathcal{S}}} \operatorname*{argmax}_{\pi \in \Delta_{\mathcal{A}}^{\mathcal{S}}} J_\tau(\pi, \phi). \quad (52)$$

Since the policy simplex is a compact set, $(\pi_1, \phi_1)$ and $(\pi_2, \phi_2)$ exist and are well-defined. The following minimax inequality always holds

$$J_\tau(\pi_1, \phi_1) = \max_{\pi \in \Delta_{\mathcal{A}}^{\mathcal{S}}} \min_{\phi \in \Delta_{\mathcal{B}}^{\mathcal{S}}} J_\tau(\pi, \phi) \leq \min_{\phi \in \Delta_{\mathcal{B}}^{\mathcal{S}}} \max_{\pi \in \Delta_{\mathcal{A}}^{\mathcal{S}}} J_\tau(\pi, \phi) = J_\tau(\pi_2, \phi_2). \quad (53)$$

We first want to show that $\pi_1 = \pi_\tau(\phi_1)$ and $\phi_1 = \phi_\tau(\pi_1)$. Since

$$J_\tau(\pi_1, \phi_1) = \max_{\pi \in \Delta_{\mathcal{A}}^{\mathcal{S}}} \min_{\phi \in \Delta_{\mathcal{B}}^{\mathcal{S}}} J_\tau(\pi, \phi) = \min_{\phi \in \Delta_{\mathcal{B}}^{\mathcal{S}}} J_\tau(\pi_1, \phi) = J_\tau(\pi_1, \phi_\tau(\pi_1)),$$

we have $\phi_1 \in \phi_\tau(\pi_1)$, and Lemma 1 further implies $\phi_1 = \phi_\tau(\pi_1)$ is unique. In addition, we know that $\pi_1$ is the optimizer of $g_\tau$ defined in (2). Let $\theta_1$ be an softmax parameter for $\pi_1$ (e.g. $\theta_1(s, a) = \log \pi(a \mid s)$ for all $s, a$). Since $\pi_1$ is an optimizer of $g_\tau$ in policy space, $\theta_1$ must also be an (not necessarily unique) optimizer of $\tilde{g}_\tau(\theta) = \min_\phi J_\tau(\pi_\theta, \phi)$ in the parameter space. Therefore, we have $\forall \theta \in \mathbb{R}^{\mathcal{S} \times \mathcal{A}}$

$$0 \geq \langle \nabla_\theta g_\tau(\pi_{\theta_1}), \theta - \theta_1 \rangle = \langle \nabla_\theta J_\tau(\pi_{\theta_1}, \phi_1), \theta - \theta_1 \rangle, \quad (54)$$

where the first equality follows from Danskin's Theorem in (16). Since $\theta$ is not constrained, (54) means that

$$\nabla_\theta J_\tau(\pi_{\theta_1}, \phi_1) = 0,$$

implying that $\theta_1$ is a stationary point of

$$\max_\theta J_\tau(\pi_\theta, \phi_1).$$

By Lemma 4, every stationary point is also globally optimal. Therefore, we have $\pi_1 = \pi_{\theta_1} = \pi_\tau(\phi_1)$.

A consequence of $\pi_1 = \pi_\tau(\phi_1)$ and $\phi_1 = \phi_\tau(\pi_1)$ is that $(\pi_1, \phi_1)$ is the unique optimal solution pair to the maximin problem, i.e. there does not exist $(\widehat{\pi}_1, \widehat{\phi}_1) \neq (\pi_1, \phi_1)$ such that $(\widehat{\pi}_1, \widehat{\phi}_1) \in \operatorname{argmax}_{\pi \in \Delta_{\mathcal{A}}^{\mathcal{S}}} \operatorname{argmin}_{\phi \in \Delta_{\mathcal{B}}^{\mathcal{S}}} J_\tau(\pi, \phi)$. To see this, let us suppose that such a pair $(\widehat{\pi}_1, \widehat{\phi}_1)$ does exist. Then, the only possibility is $\widehat{\pi}_1 \neq \pi_1$ and $\widehat{\phi}_1 \neq \phi_1$ by Lemma 1. Since $\widehat{\pi}_1 \neq \pi_\tau(\phi_1)$ and $\phi_1 \neq \phi_\tau(\widehat{\pi}_1)$, we have

$$J_\tau(\widehat{\pi}_1, \phi_1) < J_\tau(\pi_1, \phi_1) = J_\tau(\widehat{\pi}_1, \widehat{\phi}_1) < J_\tau(\widehat{\pi}_1, \phi_1),$$

which creates a contradiction.

Similarly, it can be shown that

$$\pi_2 = \phi_\tau(\phi_2), \quad \text{and} \quad \phi_2 = \phi_\tau(\pi_2),$$

and that $(\pi_2, \phi_2)$ is the unique optimal solution pair to the minimax problem.

We now aim prove that $(\pi_1, \phi_1) = (\pi_2, \phi_2)$, i.e. the minimax and maximin problem have the same solution. Suppose $(\pi_1, \phi_1) \neq (\pi_2, \phi_2)$, which means that $\pi_1 \neq \pi_2$ and $\phi_1 \neq \phi_2$ have to hold due to Lemma 1. Since $\pi_2 \neq \pi_\tau(\phi_1)$ and $\phi_1 \neq \phi_\tau(\pi_2)$, we have from (53)

$$J_\tau(\pi_2, \phi_1) < J_\tau(\pi_1, \phi_1) \leq J_\tau(\pi_2, \phi_2) < J_\tau(\pi_2, \phi_1).$$

This is again a contradiction. Therefore, $(\pi_1, \phi_1) = (\pi_2, \phi_2)$ has to be true. Then, (53) leads to

$$\max_{\pi \in \Delta_{\mathcal{A}}^{\mathcal{S}}} \min_{\phi \in \Delta_{\mathcal{B}}^{\mathcal{S}}} J_\tau(\pi, \phi) = \max_{\pi \in \Delta_{\mathcal{A}}^{\mathcal{S}}} \min_{\phi \in \Delta_{\mathcal{B}}^{\mathcal{S}}} J_\tau(\pi, \phi).$$

We also know that the Nash equilibrium has to be unique in this case, as the maximin and minimax problems both have a unique solution pair that agrees with each other.

$$\square$$

## B.3 Proof of Lemma 3

By the definition of the value function,

$$J_\tau(\pi, \phi) - J_{\tau'}(\pi, \phi)$$

$$= \mathbb{E}\left[\sum_{k=0}^\infty \gamma^k \Big(r(s_k, a_k, b_k) - \tau \log \pi(a_k \mid s_k) + \tau \log \phi(b_k \mid s_k)\Big) \mid s_0 \sim \rho\right]$$

$$- \mathbb{E}\left[\sum_{k=0}^\infty \gamma^k \Big(r(s_k, a_k, b_k) - \tau' \log \pi(a_k \mid s_k) + \tau' \log \phi(b_k \mid s_k)\Big) \mid s_0 \sim \rho\right]$$

$$= \mathbb{E}\left[\sum_{k=0}^\infty \gamma^k \Big((\tau - \tau') \log \pi(a_k \mid s_k) + (\tau - \tau') \log \phi(b_k \mid s_k)\Big) \mid s_0 \sim \rho\right]$$

$$= \frac{\tau - \tau'}{1 - \gamma} \mathbb{E}_{s' \sim d_\rho^{\pi,\phi}, a \sim \pi(\cdot \mid s'), b \sim \phi(\cdot \mid s')}\left[-\log \pi(a \mid s') + \log \phi(b \mid s')\right]$$

$$= \frac{\tau - \tau'}{1 - \gamma} \mathbb{E}_{s' \sim d_\rho^{\pi,\phi}}[H(\pi(\cdot \mid s')) - H(\phi(\cdot \mid s'))],$$

where $H$ denotes the entropy and is defined in (15).

We have the following upper and lower bound on the entropy

$$0 \le H(\pi(\cdot \mid s')) \le \log |\mathcal{A}|, \quad 0 \le H(\phi(\cdot \mid s')) \le \log |\mathcal{B}|.$$

Therefore, if $\tau \ge \tau' \ge 0$,

$$-\frac{\tau - \tau'}{1 - \gamma} \log |\mathcal{B}| \le J_\tau(\pi, \phi) - J_{\tau'}(\pi, \phi) \le \frac{\tau - \tau'}{1 - \gamma} \log |\mathcal{A}|.$$

For any $\tau \ge \tau' \ge 0$,

$$J_\tau(\pi_\tau^\star, \phi_\tau^\star) - J_{\tau'}(\pi_{\tau'}^\star, \phi_{\tau'}^\star)$$
$$= \max_\pi \min_\phi J_\tau(\pi, \phi) - \min_\phi J_{\tau'}(\pi_{\tau'}^\star, \phi)$$
$$\ge \min_\phi J_\tau(\pi_{\tau'}^\star, \phi) - \min_\phi J_{\tau'}(\pi_{\tau'}^\star, \phi)$$
$$= \min_\phi \left(J_{\tau'}(\pi_{\tau'}^\star, \phi) + (\tau - \tau')\mathcal{H}_\pi(\rho, \pi_{\tau'}^\star, \phi) - (\tau - \tau')\mathcal{H}_\phi(\rho, \pi_{\tau'}^\star, \phi)\right) - \min_\phi J_{\tau'}(\pi_{\tau'}^\star, \phi)$$
$$\ge \min_\phi J_{\tau'}(\pi_{\tau'}^\star, \phi) + (\tau-\tau') \min_\phi \mathcal{H}_\pi(\rho, \pi_{\tau'}^\star, \phi) + (\tau - \tau') \min_\phi -\mathcal{H}_\phi(\rho, \pi_{\tau'}^\star, \phi) - \min_\phi J_{\tau'}(\pi_{\tau'}^\star, \phi)$$
$$= (\tau - \tau') \left(\min_\phi \mathcal{H}_\pi(\rho, \pi_{\tau'}^\star, \phi) - \max_\phi \mathcal{H}_\phi(\rho, \pi_{\tau'}^\star, \phi)\right)$$
$$\ge (\tau - \tau')(0 - \log |\mathcal{B}|)$$
$$= -(\tau - \tau') \log |\mathcal{B}|,$$

where the second inequality comes from the fact that $\min_x f_1(x) + f_2(x) \ge \min_x f_1(x) + \min_x f_2(x)$ for any functions $f_1, f_2$ of the same domain.

It can be shown by a similar argument

$$J_\tau(\pi_\tau^\star, \phi_\tau^\star) - J_{\tau'}(\pi_{\tau'}^\star, \phi_{\tau'}^\star) \le (\tau - \tau') \log |\mathcal{A}|.$$

In addition, for any $\tau \ge \tau' \ge 0$ and any policy $\pi$,

$$J_\tau(\pi, \phi_\tau(\pi)) - J_{\tau'}(\pi, \phi_0(\pi))$$
$$= \min_\phi J_\tau(\pi, \phi) - \min_\phi J_{\tau'}(\pi, \phi)$$
$$= \min_\phi (J_{\tau'}(\pi, \phi) + (\tau - \tau')\mathcal{H}_\pi(\rho, \pi, \phi) - (\tau - \tau')\mathcal{H}_\phi(\rho, \pi, \phi)) - \min_\phi J_{\tau'}(\pi, \phi)$$
$$\le \left(\min_\phi J_{\tau'}(\pi, \phi) + (\tau - \tau') \max_\phi \mathcal{H}_\pi(\rho, \pi, \phi) + (\tau - \tau') \max_\phi(-\mathcal{H}_\phi(\rho, \pi, \phi))\right) - \min_\phi J_{\tau'}(\pi, \phi)$$

$$= (\tau - \tau') \left( \max_\phi \mathcal{H}_\pi(\rho, \pi, \phi) - \min_\phi \mathcal{H}_\phi(\rho, \pi, \phi) \right)$$
$$\leq (\tau - \tau') \log |\mathcal{A}|.$$

It can be shown by a similar argument

$$J_\tau(\pi, \phi_\tau(\pi)) - J_{\tau'}(\pi, \phi_0(\pi)) \geq -(\tau - \tau') \log |\mathcal{B}|.$$

$\square$

### B.4 Proof of Lemma 4

Adapting Mei et al. [2020][Lemma 15], we have for any $\theta \in \mathbb{R}^{\mathcal{S} \times \mathcal{A}}$ and $\psi \in \mathbb{R}^{\mathcal{S} \times \mathcal{B}}$

$$\|\nabla_\theta J_\tau(\pi_\theta, \phi_\psi)\|^2 \geq \frac{2\tau \rho_{\min}}{|\mathcal{S}|} \left( \min_{s,a} \pi_\theta(a \mid s) \right)^2 \left\| \frac{d_\rho^{\pi_\tau(\phi_\psi), \phi_\psi}}{d_\rho^{\pi_\theta, \phi_\psi}} \right\|_\infty^{-1} (J_\tau(\pi_\tau(\phi_\psi), \phi_\psi) - J_\tau(\pi_\theta, \phi_\psi)),$$

$$\|\nabla_\psi J_\tau(\pi_\theta, \phi_\psi)\|^2 \geq \frac{2\tau \rho_{\min}}{|\mathcal{S}|} \left( \min_{s,b} \phi_\psi(b \mid s) \right)^2 \left\| \frac{d_\rho^{\pi_\theta, \phi_\tau(\pi_\theta)}}{d_\rho^{\pi_\theta, \phi_\psi}} \right\|_\infty^{-1} (J_\tau(\pi_\theta, \phi_\psi) - J_\tau(\pi_\theta, \phi_\tau(\pi_\theta))).$$

Then, the first inequality follows from $d_\rho^{\pi_\tau(\phi_\psi), \phi_\psi}(s) \leq 1$ and $d_\rho^{\pi_\theta, \phi_\psi}(s) \geq (1-\gamma)\rho(s) \geq (1-\gamma)\rho_{\min}$ for all $s \in \mathcal{S}$, and the second inequality from $d_\rho^{\pi_\theta, \phi_\tau(\pi_\theta)} \leq 1$ and $d_\rho^{\pi_\theta, \phi_\psi} \geq (1-\gamma)\rho_{\min}$ for all $s \in \mathcal{S}$.

$\square$

### B.5 Proof of Lemma 5

Mei et al. [2020][Lemma 7, Lemma 14] establishes the smoothness condition of the value function and the regularization entropy with respect to one player's policy, i.e.

$$\|\nabla_\theta J(\pi_{\theta_1}, \phi_{\psi_1}) - \nabla_\theta J(\pi_{\theta_2}, \phi_{\psi_1})\| \leq L_V \|\theta_1 - \theta_2\|,$$
$$\|\nabla_\psi J(\pi_{\theta_1}, \phi_{\psi_1}) - \nabla_\psi J(\pi_{\theta_1}, \phi_{\psi_2})\| \leq L_V \|\psi_1 - \psi_2\|.$$

Therefore, we only need to show

$$\|\nabla_\theta J(\pi_{\theta_1}, \phi_{\psi_1}) - \nabla_\theta J(\pi_{\theta_1}, \phi_{\psi_2})\| \leq L_V \|\psi_1 - \psi_2\|,$$
$$\|\nabla_\psi J(\pi_{\theta_1}, \phi_\psi) - \nabla_\psi J(\pi_{\theta_2}, \phi_\psi)\| \leq L_V \|\theta_1 - \theta_2\|.$$

Given a fixed $\theta$ and $\psi$, with arbitrary vectors $u$ and $v$ such that $\|u\|_2 = \|v\|_2 = 1$, we define the shorthand notation

$$\pi_{\alpha,u} = \pi_{\theta+\alpha u}, \quad \phi_{\beta,v} = \pi_{\psi+\beta v}.$$

According to Zeng et al. [2021a][Lemma B.5],

$$\sum_a \left| \frac{d\pi_{\alpha,u}(a \mid s)}{d\alpha} \right| \leq 2, \quad \sum_b \left| \frac{d\phi_{\beta,v}(b \mid s)}{d\beta} \right| \leq 2,$$

$$\sum_{a,b} \left| \frac{d\pi_\alpha(a \mid s)}{d\alpha} \frac{d\phi_{\beta,v}(b \mid s)}{d\beta} \right| \leq \left( \sum_a \left| \frac{d\pi_\alpha(a \mid s)}{d\alpha} \right| \right) \left( \sum_b \left| \frac{d\phi_\beta(b \mid s)}{d\beta} \right| \right) \leq 4.$$

Let $P(\alpha, \beta, u, v) \in \mathbb{R}^{|\mathcal{S}||\mathcal{A}||\mathcal{B}| \times |\mathcal{S}||\mathcal{A}||\mathcal{B}|}$ denote the state-action transition matrix induced by the policy pair $(\pi_{\alpha,u}, \phi_{\beta,v})$

$$P(\alpha, \beta, u, v)_{(s,a,b) \to (s',a',b')} = \mathcal{P}(s' \mid s, a, b)\pi_{\alpha,u}(a' \mid s')\phi_{\beta,v}(b' \mid s').$$

Differentiating with respect to $\alpha$ and $\beta$,

$$\left[ \frac{d^2 P(\alpha, \beta, u, v)}{d\alpha d\beta} \right]_{(s,a,b) \to (s',a',b')} = \frac{d\pi_{\alpha,u}(a' \mid s')}{d\alpha} \frac{d\phi_{\beta,v}(b' \mid s')}{d\beta} \mathcal{P}(s' \mid s, a, b),$$

which implies for any vector $x$

$$\left[\frac{d^2 P(\alpha, \beta, u, v)}{d\alpha d\beta} x\right]_{s,a,b} = \sum_{s',a',b'} \frac{d\pi_\alpha(a' \mid s')}{d\alpha} \frac{d\phi_{\beta,v}(b' \mid s')}{d\beta} \mathcal{P}(s' \mid s, a, b) x_{s',a',b'}.$$

The $\ell_\infty$ norm of this quantity can be upper bounded

$$\max_{\|u\|_2=\|v\|_2=1} \left\|\frac{d^2 P(\alpha, \beta, u, v)}{d\alpha d\beta} x\right\|_\infty$$

$$= \max_{s,a,b} \max_{\|u\|_2=\|v\|_2=1} \left|\left[\frac{d^2 P(\alpha, \beta, u, v)}{d\alpha d\beta} x\right]_{s,a,b}\right|$$

$$= \max_{s,a,b} \max_{\|u\|_2=\|v\|_2=1} \left|\sum_{s',a',b'} \frac{d\pi_\alpha(a' \mid s')}{d\alpha} \frac{d\phi_{\beta,v}(b' \mid s')}{d\beta} \mathcal{P}(s' \mid s, a, b) x_{s',a',b'}\right|$$

$$\leq \max_{s,a,b} \sum_{s'} \mathcal{P}(s' \mid s, a, b) \|x\|_\infty \max_{\|u\|_2=\|v\|_2=1} \sum_{a',b'} \left|\frac{d\pi_\alpha(a' \mid s')}{d\alpha} \frac{d\phi_{\beta,v}(b' \mid s')}{d\beta}\right|$$

$$\leq 4\|x\|_\infty. \tag{55}$$

Using an identical argument, we can show that

$$\max_{\|u\|_2=\|v\|_2=1} \left\|\frac{dP(\alpha, \beta, u, v)}{d\alpha} x\right\|_\infty \leq \sum_a \left|\frac{d\pi_{\alpha,u}(a \mid s)}{d\alpha}\right| \|x\|_\infty \leq 2\|x\|_\infty, \tag{56}$$

$$\max_{\|u\|_2=\|v\|_2=1} \left\|\frac{dP(\alpha, \beta, u, v)}{d\beta} x\right\|_\infty \leq \sum_b \left|\frac{d\pi_{\beta,v}(b \mid s)}{d\beta}\right| \|x\|_\infty \leq 2\|x\|_\infty. \tag{57}$$

With $M(\alpha, \beta, u, v) = (I - \gamma P(\alpha, \beta, u, v))^{-1}$ and $r = [r(s_0, a_0, b_0), \cdots, r(s_{|\mathcal{S}|}, a_{|\mathcal{A}|}, b_{|\mathcal{B}|})]$,

$$Q^{\pi_{\alpha,u}, \phi_{\beta,v}}(s, a, b) = e_{s,a,b}^\top M(\alpha, \beta, u, v) r.$$

Taking the derivatives,

$$\frac{dQ^{\pi_{\alpha,u}, \phi_{\beta,v}}(s, a, b)}{d\alpha} = \gamma e_{s,a,b}^\top M(\alpha, \beta, u, v) \frac{dP(\alpha, \beta, u, v)}{d\alpha} M(\alpha, \beta, u, v) r,$$

$$\frac{dQ^{\pi_{\alpha,u}, \phi_{\beta,v}}(s, a, b)}{d\beta} = \gamma e_{s,a,b}^\top M(\alpha, \beta, u, v) \frac{dP(\alpha, \beta, u, v)}{d\beta} M(\alpha, \beta, u, v) r.$$

Taking the second-order derivative,

$$\frac{d^2 Q^{\pi_{\alpha,u}, \phi_{\beta,v}}(s, a, b)}{d\alpha d\beta}$$

$$= \gamma^2 e_{s,a,b}^\top M(\alpha, \beta, u, v) \frac{dP(\alpha, \beta, u, v)}{d\alpha} M(\alpha, \beta, u, v) \frac{dP(\alpha, \beta, u, v)}{d\beta} M(\alpha, \beta, u, v) r$$

$$+ \gamma^2 e_{s,a,b}^\top M(\alpha, \beta, u, v) \frac{dP(\alpha, \beta, u, v)}{d\beta} M(\alpha, \beta, u, v) \frac{dP(\alpha, \beta, u, v)}{d\alpha} M(\alpha, \beta, u, v) r$$

$$+ \gamma e_{s,a,b}^\top M(\alpha, \beta, u, v) \frac{d^2 P(\alpha, \beta, u, v)}{d\alpha d\beta} M(\alpha, \beta, u, v) r$$

Using $M(\alpha, \beta, u, v)\mathbf{1} = (I - \gamma P(\alpha, \beta, u, v))^{-1}\mathbf{1} = \frac{1}{1-\gamma}\mathbf{1}$ and inequalities (55) and (57), we have

$$\max_{\|u\|_2=\|v\|_2=1} \left|\frac{dQ^{\pi_{\alpha,u}, \phi_{\beta,v}}(s, a, b)}{d\alpha}\right| \leq \left\|\gamma M(\alpha, \beta, u, v) \frac{dP(\alpha, \beta, u, v)}{d\alpha} M(\alpha, \beta, u, v) r\right\|_\infty \leq \frac{2\gamma}{(1-\gamma)^2},$$

$$\max_{\|u\|_2=\|v\|_2=1} \left|\frac{dQ^{\pi_{\alpha,u}, \phi_{\beta,v}}(s, a, b)}{d\beta}\right| \leq \left\|\gamma M(\alpha, \beta, u, v) \frac{dP(\alpha, \beta, u, v)}{d\beta} M(\alpha, \beta, u, v) r\right\|_\infty \leq \frac{2\gamma}{(1-\gamma)^2},$$

and

$$\max_{\|u\|_2=\|v\|_2=1} \left| \frac{d^2 Q^{\pi_{\alpha,u},\phi_{\beta,v}}(s,a,b)}{d\alpha d\beta} \right|$$

$$\leq \left\| \gamma^2 M(\alpha,\beta,u,v) \frac{dP(\alpha,\beta,u,v)}{d\alpha} M(\alpha,\beta,u,v) \frac{dP(\alpha,\beta,u,v)}{d\beta} M(\alpha,\beta,u,v)r \right\|_\infty$$

$$+ \left\| \gamma^2 M(\alpha,\beta,u,v) \frac{dP(\alpha,\beta,u,v)}{d\beta} M(\alpha,\beta,u,v) \frac{dP(\alpha,\beta,u,v)}{d\alpha} M(\alpha,\beta,u,v)r \right\|_\infty$$

$$+ \left\| \gamma M(\alpha,\beta,u,v) \frac{d^2 P(\alpha,\beta,u,v)}{d\alpha d\beta} M(\alpha,\beta,u,v)r \right\|_\infty$$

$$\leq \frac{2\gamma^2}{(1-\gamma)^3} + \frac{4\gamma}{(1-\gamma)^2}.$$

Since $V^{\pi_{\alpha,u},\phi_{\beta,v}}(s) = \sum_{a,b} \pi_{\alpha,u}(a \mid s)\phi_{\beta,v}(b \mid s)Q^{\pi_{\alpha,u},\phi_{\beta,v}}(s,a,b)$,

$$\frac{d^2 V^{\pi_{\alpha,u},\phi_{\beta,v}}(s)}{d\alpha d\beta} = \sum_{a,b} \frac{d\pi_{\alpha,u}(a \mid s)}{d\alpha} \frac{d\phi_{\beta,v}(b \mid s)}{d\beta} Q^{\pi_{\alpha,u},\phi_{\beta,v}}(s,a,b)$$

$$+ \sum_{a,b} \pi_{\alpha,u}(a \mid s)\phi_{\beta,v}(b \mid s) \frac{d^2 Q^{\pi_{\alpha,u},\phi_{\beta,v}}(s,a,b)}{d\alpha d\beta}$$

$$+ \sum_{a,b} \frac{d\pi_{\alpha,u}(a \mid s)}{d\alpha}\phi_{\beta,v}(b \mid s) \frac{dQ^{\pi_{\alpha,u},\phi_{\beta,v}}(s,a,b)}{d\beta}$$

$$+ \sum_{a,b} \pi_{\alpha,u}(a \mid s) \frac{d\phi_{\beta,v}(b \mid s)}{d\beta} \frac{dQ^{\pi_{\alpha,u},\phi_{\beta,v}}(s,a,b)}{d\alpha}.$$

Therefore,

$$\max_{\|u\|_2=\|v\|_2=1} \left| \frac{dV^{\pi_{\alpha,u},\phi_{\beta,v}}(s)}{d\alpha d\beta} \right| \leq \frac{4}{1-\gamma} + \left( \frac{2\gamma^2}{(1-\gamma)^3} + \frac{4\gamma}{(1-\gamma)^2} \right) + 2\frac{4\gamma}{(1-\gamma)^2} \leq \frac{8}{(1-\gamma)^3},$$

which implies

$$\|\nabla_\theta J(\pi_\theta, \phi_{\psi_1}) - \nabla_\theta J(\pi_\theta, \phi_{\psi_2})\| \leq \frac{8}{(1-\gamma)^3}\|\psi_1 - \psi_2\|.$$

Similarly, it follows by the same argument that

$$\|\nabla_\psi J(\pi_{\theta_1}, \phi_\psi) - \nabla_\psi J(\pi_{\theta_2}, \phi_\psi)\| \leq \frac{8}{(1-\gamma)^3}\|\theta_1 - \theta_2\|.$$

Zeng et al. [2021a][Lemma B.5] implies

$$\|J(\pi_{\theta_1}, \phi_{\psi_1}) - J(\pi_{\theta_2}, \phi_{\psi_2})\| \leq \frac{2}{(1-\gamma)^2}(\|\theta_1 - \theta_2\| + \|\psi_1 - \psi_2\|), \tag{58}$$

and we simply use $\frac{2}{(1-\gamma)^2} \leq L_V$.

$\square$

## B.6  Proof of Lemma 6

We will prove the first two inequalities on the Lipschitz gradient of $\mathcal{H}_\pi$. The next two inequalities are completely symmetric and can be derived using an identical argument.

Mei et al. [2020][Lemma 14] implies

$$\|\nabla_\theta \mathcal{H}_\pi(s, \pi_{\theta_1}, \phi_{\psi_1}) - \nabla_\theta \mathcal{H}_\pi(s, \pi_{\theta_2}, \phi_{\psi_1})\| \leq L_\mathcal{H}\|\theta_1 - \theta_2\|,$$

so we just need to show

$$
\begin{aligned}
\|\nabla_\theta \mathcal{H}_\pi(s, \pi_{\theta_1}, \phi_{\psi_1}) - \nabla_\theta \mathcal{H}_\pi(s, \pi_{\theta_1}, \phi_{\psi_2})\| &\le L_{\mathcal{H}} \|\psi_1 - \psi_2\|, \\
\|\nabla_\psi \mathcal{H}_\pi(s, \pi_{\theta_1}, \phi_{\psi_1}) - \nabla_\psi \mathcal{H}_\pi(s, \pi_{\theta_2}, \phi_{\psi_1})\| &\le L_{\mathcal{H}} \|\theta_1 - \theta_2\|, \\
\|\nabla_\psi \mathcal{H}_\pi(s, \pi_{\theta_1}, \phi_{\psi_1}) - \nabla_\psi \mathcal{H}_\pi(s, \pi_{\theta_1}, \phi_{\psi_2})\| &\le L_{\mathcal{H}} \|\psi_1 - \psi_2\|.
\end{aligned} \tag{59}
$$

Given a fixed $\theta$ and $\psi$, with arbitrary vectors $u$ and $v$ such that $\|u\|_2 = \|v\|_2 = 1$, we define the shorthand notation

$$
\pi_{\alpha,u} = \pi_{\theta + \alpha u}, \quad \phi_{\beta,v} = \pi_{\psi + \beta v}.
$$

Note that to show (59), it suffices to show for any $u, v$

$$
\left| \frac{d^2 \mathcal{H}_\pi(s, \pi_{\alpha,u}, \phi_{\beta,v})}{d\alpha d\beta} \right| \le L_{\mathcal{H}}, \quad \left| \frac{d^2 \mathcal{H}_\pi(s, \pi_{\alpha,u}, \phi_{\beta,v})}{d\beta^2} \right| \le L_{\mathcal{H}}.
$$

We define the state transition matrix $P \in \mathbb{R}^{|\mathcal{S}| \times |\mathcal{S}|}$ such that

$$
P(\alpha, \beta, u, v)_{s \to s'} = \sum_{a,b} \mathcal{P}(s' \mid s, a, b) \pi_{\alpha,u}(a \mid s) \phi_{\beta,v}(b \mid s).
$$

Let $M(\alpha, \beta, u, v) = (I - \gamma P(\alpha, \beta, u, v))^{-1}$. Then, we can re-write $\mathcal{H}_\pi(s, \pi, \phi)$ in the matrix form

$$
\mathcal{H}_\pi(s, \pi, \phi) = e_s^\top M(\alpha, \beta, u, v) h_{\alpha,u},
$$

where $h_{\alpha,u} = [h_{\alpha,u}(s_0), \cdots, h_{\alpha,u}(s_{|\mathcal{S}|})] \in \mathbb{R}^{|\mathcal{S}|}$ is a vector with

$$
h_{\alpha,u}(s) = -\sum_a \pi_{\alpha,u}(a \mid s) \log \pi_{\alpha,u}(a \mid s).
$$

According to Mei et al. [2020][Lemma 14],

$$
\left\| \frac{dh_{\alpha,u}}{d\alpha} \right\|_\infty \le 2 \log |\mathcal{A}| \|u\|_2 = 2 \log |\mathcal{A}|.
$$

Taking the derivatives of $\mathcal{H}_\pi(s, \pi, \phi)$,

$$
\begin{aligned}
&\frac{d\mathcal{H}_\pi(s, \pi_{\alpha,u}, \phi_{\beta,v})}{d\alpha} \\
&= \gamma e_s^\top M(\alpha, \beta, u, v) \frac{dP(\alpha, \beta, u, v)}{d\alpha} M(\alpha, \beta, u, v) h_{\alpha,u} + e_s^\top M(\alpha, \beta, u, v) \frac{dh_{\alpha,u}}{d\alpha},
\end{aligned}
$$

and taking second order derivative

$$
\begin{aligned}
&\frac{d^2 \mathcal{H}_\pi(s, \pi_{\alpha,u}, \phi_{\beta,v})}{d\alpha d\beta} \\
&= \gamma^2 e_s^\top M(\alpha, \beta, u, v) \frac{dP(\alpha, \beta, u, v)}{d\alpha} M(\alpha, \beta, u, v) \frac{dP(\alpha, \beta, u, v)}{d\beta} M(\alpha, \beta, u, v) h_{\alpha,u} \\
&\quad + \gamma^2 e_s^\top M(\alpha, \beta, u, v) \frac{dP(\alpha, \beta, u, v)}{d\beta} M(\alpha, \beta, u, v) \frac{dP(\alpha, \beta, u, v)}{d\alpha} M(\alpha, \beta, u, v) h_{\alpha,u} \\
&\quad + \gamma e_s^\top M(\alpha, \beta, u, v) \frac{d^2 P(\alpha, \beta, u, v)}{d\alpha d\beta} M(\alpha, \beta, u, v) h_{\alpha,u} \\
&\quad + \gamma e_s^\top M(\alpha, \beta, u, v) \frac{dP(\alpha, \beta, u, v)}{d\beta} M(\alpha, \beta, u, v) \frac{dh_{\alpha,u}}{d\alpha}.
\end{aligned}
$$

Using a similar line of argument to Mei et al. [2020][Eq. (192)-(195)] and analysis in Lemma 5 of our work, we can show that for any vector $x$

$$
\left\| \frac{dP(\alpha, \beta, u, v)}{d\alpha} x \right\|_\infty \le 2\|x\|_\infty, \quad \left\| \frac{dP(\alpha, \beta, u, v)}{d\beta} \right\|_\infty \le 2\|x\|_\infty, \quad \left\| \frac{d^2 P(\alpha, \beta, u, v)}{d\alpha d\beta} \right\|_\infty \le 4\|x\|_\infty.
$$

From the fact that $\|M(\alpha, \beta, u, v)x\|_\infty \leq \frac{1}{1-\gamma}\|x\|_\infty$, we have for any vectors $u, v$

$$\left| \frac{d^2 \mathcal{H}_\pi(s, \pi_{\alpha,u}, \phi_{\beta,v})}{d\alpha d\beta} \right|$$

$$\leq \gamma^2 \left\| M(\alpha, \beta, u, v) \frac{dP(\alpha, \beta, u, v)}{d\alpha} M(\alpha, \beta, u, v) \frac{dP(\alpha, \beta, u, v)}{d\beta} M(\alpha, \beta, u, v) h_{\alpha,u} \right\|$$

$$+ \gamma^2 \left\| M(\alpha, \beta, u, v) \frac{dP(\alpha, \beta, u, v)}{d\beta} M(\alpha, \beta, u, v) \frac{dP(\alpha, \beta, u, v)}{d\alpha} M(\alpha, \beta, u, v) h_{\alpha,u} \right\|$$

$$+ \gamma \left\| M(\alpha, \beta, u, v) \frac{d^2 P(\alpha, \beta, u, v)}{d\alpha d\beta} M(\alpha, \beta, u, v) h_{\alpha,u} \right\|$$

$$+ \gamma \left\| M(\alpha, \beta, u, v) \frac{dP(\alpha, \beta, u, v)}{d\beta} M(\alpha, \beta, u, v) \frac{dh_{\alpha,u}}{d\alpha} \right\|$$

$$\leq \frac{4\gamma^2 \log |\mathcal{A}|}{(1-\gamma)^3} + \frac{4\gamma^2 \log |\mathcal{A}|}{(1-\gamma)^3} + \frac{4\gamma \log |\mathcal{A}|}{(1-\gamma)^2} + \frac{2\gamma}{(1-\gamma)^2} \cdot 2\log |\mathcal{A}|$$

$$\leq \frac{8 \log |\mathcal{A}|}{(1-\gamma)^3}.$$

Now it remains to be shown

$$\left| \frac{d^2 \mathcal{H}_\pi(s, \pi_{\alpha,u}, \phi_{\beta,v})}{d\beta^2} \right| \leq L_\mathcal{H}.$$

From the eye of the second player, $\mathcal{H}_\pi(s, \pi_\theta, \phi_\psi)$ is simply the value function of a regular MDP with itself as the only agent (the first player's policy combines with $\mathcal{P}$) with the reward function $r(s, b) = -\sum_{a \in \mathcal{A}} \pi_\theta(a \mid s) \log \pi_\theta(a \mid s) \in [0, \log |\mathcal{A}|]$. Therefore, by Lemma 5 which is derived with reward bounded between 0 and 1, we know

$$\left| \frac{d^2 \mathcal{H}_\pi(s, \pi_{\alpha,u}, \phi_{\beta,v})}{d\beta^2} \right| \leq \log |\mathcal{A}| L_V \leq L_\mathcal{H}.$$

To show the Lipschitz continuity, we note that

$$\left| \frac{d\mathcal{H}_\pi(s, \pi_{\alpha,u}, \phi_{\beta,v})}{d\alpha} \right|$$

$$= \left| \gamma e_s^\top M(\alpha, \beta, u, v) \frac{dP(\alpha, \beta, u, v)}{d\alpha} M(\alpha, \beta, u, v) h_{\alpha,u} + e_s^\top M(\alpha, \beta, u, v) \frac{dh_{\alpha,u}}{d\alpha} \right|$$

$$\leq \gamma \| M(\alpha, \beta, u, v) \frac{dP(\alpha, \beta, u, v)}{d\alpha} M(\alpha, \beta, u, v) h_{\alpha,u} \| + \| M(\alpha, \beta, u, v) \frac{dh_{\alpha,u}}{d\alpha} \|$$

$$\leq \frac{4\gamma \log |\mathcal{A}|}{(1-\gamma)^2} + \frac{2 \log |\mathcal{A}|}{1-\gamma} \leq L_\mathcal{H}.$$

To show the Lipschitz continuity of $\mathcal{H}_\pi$ with respect to $\psi$, we use the same argument as above and note that from the eye of the second player, $\mathcal{H}_\pi(s, \pi_\theta, \phi_\psi)$ is simply the value function of a regular MDP with itself as the only agent (the first player's policy combines with $\mathcal{P}$) with the reward function $r(s, b) = -\sum_{a \in \mathcal{A}} \pi_\theta(a \mid s) \log \pi_\theta(a \mid s) \in [0, \log |\mathcal{A}|]$. Adapting (58), we have

$$\left| \frac{d\mathcal{H}_\pi(s, \pi_{\alpha,u}, \phi_{\beta,v})}{d\beta} \right| \leq \frac{2}{(1-\gamma)^2} \cdot \log |\mathcal{A}| \leq L_\mathcal{H}.$$

$\square$

### B.7 Proof of Lemma 7

We first show that for any $\tilde{k} > 0$, we have $\frac{1}{\tilde{k}^a} - \frac{1}{(\tilde{k}+1)^a} \leq \frac{8}{3(\tilde{k}+1)^{a+1}}$.

Since the integer $\tilde{k}$ is positive, it can be lower bound by $\frac{\tilde{k}+1}{2}$.

$$\frac{1}{\tilde{k}^a} - \frac{1}{(\tilde{k}+1)^a}$$

$$= \frac{(\tilde{k}+1)^a - \tilde{k}^a}{\tilde{k}^a(\tilde{k}+1)^a} \leq \frac{2((\tilde{k}+1)^a - \tilde{k}^a)}{(\tilde{k}+1)^{2a}} = \frac{2((\tilde{k}+1)^a - \tilde{k}^a)\left((\tilde{k}+1)^{1-a} + \tilde{k}^{1-a}\right)}{(\tilde{k}+1)^{2a}\left((\tilde{k}+1)^{1-a} + \tilde{k}^{1-a}\right)}$$

$$\leq \frac{2((\tilde{k}+1)^a - \tilde{k}^a)\left((\tilde{k}+1)^{1-a} + \tilde{k}^{1-a}\right)}{(\tilde{k}+1)^{2a}\left((\tilde{k}+1)^{1-a} + \frac{1}{2}(\tilde{k}+1)^{1-a}\right)} = \frac{4((\tilde{k}+1)^a - \tilde{k}^a)\left((\tilde{k}+1)^{1-a} + \tilde{k}^{1-a}\right)}{3(\tilde{k}+1)^{a+1}}$$

$$= \frac{4\left((\tilde{k}+1) - \tilde{k}^a(\tilde{k}+1)^{1-a} + \tilde{k}^{1-a}(\tilde{k}+1)^a - \tilde{k}\right)}{3(\tilde{k}+1)^{a+1}}$$

$$= \frac{4\left(1 - \tilde{k}^a(\tilde{k}+1)^{1-a} + \tilde{k}^{1-a}(\tilde{k}+1)^a\right)}{3(\tilde{k}+1)^{a+1}} \leq \frac{8}{3(\tilde{k}+1)^{a+1}},$$

where the last inequality follows from

$$\tilde{k}^{1-a}(\tilde{k}+1)^a - \tilde{k}^a(\tilde{k}+1)^{1-a} \leq (\tilde{k}+1)^{1-a}(\tilde{k}+1)^a - \tilde{k}^a\tilde{k}^{1-a} = \tilde{k}+1 - \tilde{k} = 1.$$

Choosing $\tilde{k} = k + h$ yields

$$\frac{1}{(k+h)^a} - \frac{1}{(k+1+h)^a} \leq \frac{8}{3(k+1+h)^{a+1}} \leq \frac{8}{3(k+h)^{a+1}}.$$

$$\square$$

## B.8 Proof of Lemma 8

The property of the min and max function implies that

$$\max_{s,a}(\pi_\tau^\star(a \mid s) - \pi_{\theta_k}(a \mid s)) + \min_{s,a} \pi_{\theta_k}(a \mid s) \geq \min_{s,a} \pi_\tau^\star(a \mid s).$$

Since the three terms are all non-negative, the inequality holds after taking the square

$$(\min_{s,a} \pi_\tau^\star(a \mid s))^2 \leq (\max_{s,a}(\pi_\tau^\star(a \mid s) - \pi_{\theta_k}(a \mid s)) + \min_{s,a} \pi_{\theta_k}(a \mid s))^2$$

$$\leq \frac{4}{3}(\min_{s,a} \pi_{\theta_k}(a \mid s))^2 + 4(\max_{s,a}(\pi_\tau^\star(a \mid s) - \pi_{\theta_k}(a \mid s)))^2.$$

Re-arranging the terms,

$$-\left(\min_{s,a} \pi_{\theta_k}(a \mid s)\right)^2 \leq -\frac{3}{4}\left(\min_{s,a} \pi_\tau^\star(a \mid s)\right)^2 + 3\left(\max_{s,a} \pi_\tau^\star(a \mid s) - \pi_{\phi_k}(a \mid s)\right)^2$$

$$\leq -\frac{3}{4}\left(\min_{s,a} \pi_\tau^\star(a \mid s)\right)^2 + 3\|\pi_\tau^\star - \pi_{\phi_k}\|^2$$

From Lemma 1,

$$-\left(\min_{s,a} \pi_{\theta_k}(a \mid s)\right)^2 \leq -\frac{3}{4}\left(\min_{s,a} \pi_\tau^\star(a \mid s)\right)^2 + 3\|\pi_\tau^\star - \pi_{\phi_k}\|^2$$

$$\leq -\frac{3}{4}\left(\min_{s,a} \pi_\tau^\star(a \mid s)\right)^2 + \frac{6\log(2)}{\tau\rho_{\min}}(J_\tau(\pi_\tau^\star, \phi_\tau^\star) - J_\tau(\pi_{\theta_k}, \phi_\tau^\star))$$

$$\leq -\frac{3}{4}\left(\min_{s,a} \pi_\tau^\star(a \mid s)\right)^2 + \frac{6\log(2)}{\tau\rho_{\min}}(J_\tau(\pi_\tau^\star, \phi_\tau^\star) - J_\tau(\pi_{\theta_k}, \phi_\tau(\pi_{\theta_k})))$$

$$= -\frac{3}{4}\left(\min_{s,a} \pi_\tau^\star(a \mid s)\right)^2 + \frac{6\log(2)}{\tau\rho_{\min}}\delta_k^\pi \tag{60}$$

Since $3\delta_k^\pi + \delta_k^\phi \le (1 - \frac{\alpha(1-\gamma)\tau\rho_{\min}^2 c^2}{32|\mathcal{S}|})^k(3\delta_0^\pi + \delta_0^\phi) \le 3\delta_0^\pi + \delta_0^\phi \le \frac{\rho_{\min} c^2}{64\log(2)}$, we have $\delta_k^\pi \le \frac{\rho_{\min} c^2}{64\log(2)}$.
Then, (60) implies

$$-\left(\min_{s,a} \pi_{\theta_k}(a \mid s)\right)^2 \le -\frac{3}{4}\left(\min_{s,a} \pi_\tau^\star(a \mid s)\right)^2 + \frac{6\log(2)}{\tau\rho_{\min}}\delta_k^\pi \le -\frac{3c^2}{4} + \frac{3c^2}{32} \le -\frac{3c^2}{8}.$$

Similarly, the property of the min and max function implies that

$$\max_{s,b}(\phi_\tau^\star(b \mid s) - \phi_{\psi_k}(b \mid s)) + \min_{s,b} \phi_{\psi_k}(b \mid s) \ge \min_{s,b} \phi_\tau^\star(b \mid s).$$

Again, all three terms are non-negative, which means that the inequality is preserved after taking the square

$$(\min_{s,b} \phi_\tau^\star(b \mid s))^2 \le (\min_{s,b} \phi_{\psi_k}(b \mid s) + \max_{s,b}(\phi_\tau^\star(b \mid s) - \phi_{\psi_k}(b \mid s)))^2$$

$$\le \frac{4}{3}(\min_{s,b} \phi_{\psi_k}(b \mid s))^2 + 4(\max_{s,b}(\phi_\tau^\star(b \mid s) - \phi_{\psi_k}(b \mid s)))^2,$$

which leads to

$$-(\min_{s,b} \phi_{\psi_k}(b \mid s))^2 \le -\frac{3}{4}(\min_{s,b} \phi_\tau^\star(b \mid s))^2 + 3(\max_{s,b}(\phi_\tau^\star(b \mid s) - \phi_{\psi_k}(b \mid s)))^2$$

$$\le -\frac{3}{4}(\min_{s,b} \phi_\tau^\star(b \mid s))^2 + 3\|\phi_\tau^\star - \phi_{\psi_k}\|^2$$

$$\le -\frac{3}{4}(\min_{s,b} \phi_\tau^\star(b \mid s))^2 + 6\|\phi_\tau(\pi_{\theta_k}) - \phi_{\psi_k}\|^2 + 6\|\phi_\tau^\star - \phi_\tau(\pi_{\theta_k})\|^2. \quad (61)$$

From Lemma 1,

$$\|\phi_\tau(\pi_{\theta_k}) - \phi_{\psi_k}\|^2 \le \frac{2\log(2)}{\tau\rho_{\min}}\left(J_\tau(\pi_{\theta_k}, \phi_{\psi_k}) - J_\tau(\pi_{\theta_k}, \phi_\tau(\pi_{\theta_k}))\right) = \frac{2\log(2)}{\tau\rho_{\min}}\delta_k^\phi, \quad (62)$$

and

$$\|\phi_\tau^\star - \phi_\tau(\pi_{\theta_k})\|^2 \le \frac{2\log(2)}{\tau\rho_{\min}}\left(J_\tau(\pi_{\theta_k}, \phi_\tau^\star) - J_\tau(\pi_{\theta_k}, \phi_\tau(\pi_{\theta_k}))\right)$$

$$\le \frac{2\log(2)}{\tau\rho_{\min}}\left(J_\tau(\pi_\tau^\star, \phi_\tau^\star) - J_\tau(\pi_{\theta_k}, \phi_\tau(\pi_{\theta_k}))\right)$$

$$= \frac{2\log(2)}{\tau\rho_{\min}}\delta_k^\pi, \quad (63)$$

Using (62) and (63) in (61),

$$-(\min_{s,b} \phi_{\psi_k}(b \mid s))^2 \le -\frac{3}{4}(\min_{s,b} \phi_\tau^\star(b \mid s))^2 + 6\|\phi_\tau(\pi_{\theta_k}) - \phi_{\psi_k}\|^2 + 6\|\phi_\tau^\star - \phi_\tau(\pi_{\theta_k})\|^2$$

$$\le -\frac{3}{4}(\min_{s,b} \phi_\tau^\star(b \mid s))^2 + \frac{12\log(2)}{\tau\rho_{\min}}\delta_k^\phi + \frac{12\log(2)}{\tau\rho_{\min}}\delta_k^\pi$$

$$= -\frac{3}{4}(\min_{s,b} \phi_\tau^\star(b \mid s))^2 + \frac{12\log(2)}{\tau\rho_{\min}}(\delta_k^\pi + \delta_k^\phi).$$

$3\delta_k^\pi + \delta_k^\phi \le (1 - \frac{\alpha(1-\gamma)\tau\rho_{\min}^2 c^2}{32|\mathcal{S}|})^k(3\delta_0^\pi + \delta_0^\phi) \le 3\delta_0^\pi + \delta_0^\phi \le \frac{\rho_{\min} c^2}{64\log(2)}$ guarantees $\delta_k^\pi + \delta_k^\phi \le \frac{\rho_{\min} c^2}{32\log(2)}$.
Using this in the inequality above, we have

$$-(\min_{s,b} \phi_{\psi_k}(b \mid s))^2 \le -\frac{3}{4}(\min_{s,b} \phi_\tau^\star(b \mid s))^2 + \frac{12\log(2)}{\tau\rho_{\min}}(\delta_k^\pi + \delta_k^\phi) \le -\frac{3c^2}{4} + \frac{3c^2}{8} \le -\frac{3c^2}{8}.$$

$$\square$$

## B.9 Proof of Lemma 9

From Lemma 4, for any $\psi \in \mathbb{R}^{|\mathcal{S}| \times |\mathcal{B}|}$

$$J_\tau(\pi_{\theta_2}, \phi_\psi) - J_\tau(\pi_{\theta_2}, \phi_\tau(\pi_{\theta_2})) \leq \frac{|\mathcal{S}|}{2\tau \rho_{\min} (\min_{s,a} \phi_\psi(a \mid s))^2} \left\| \frac{d_\rho^{\pi_{\theta_2}, \phi_\tau(\pi_{\theta_2})}}{d_\rho^{\pi_{\theta_2}, \phi_\psi}} \right\|_\infty \|\nabla_\psi J_\tau(\pi_{\theta_2}, \phi_\psi)\|^2$$

$$\leq \frac{|\mathcal{S}|}{2\tau(1-\gamma) (\min_{s,a} \phi_\psi(a \mid s))^2} \|\nabla_\psi J_\tau(\pi_{\theta_2}, \phi_\psi)\|^2,$$

where the second inequality follows by an argument similar to (48). Letting $\psi$ be the parameter that parameterizes $\phi_\tau(\pi_{\theta_1})$, we have

$$J_\tau(\pi_{\theta_2}, \phi_\tau(\pi_{\theta_1})) - J_\tau(\pi_{\theta_2}, \phi_\tau(\pi_{\theta_2}))$$

$$\leq \frac{|\mathcal{S}|}{2\tau(1-\gamma) (\min_{s,a} \phi_\tau(\pi_{\theta_1})(a \mid s))^2} \|\nabla_\psi J_\tau(\pi_{\theta_2}, \phi_\tau(\pi_{\theta_1}))\|^2$$

$$= \frac{|\mathcal{S}|}{2\tau(1-\gamma) (\min_{s,a} \phi_\tau(\pi_{\theta_1})(a \mid s))^2} \|\nabla_\psi J_\tau(\pi_{\theta_2}, \psi_{\rho,\tau}^\star(\pi_{\theta_1})) - \nabla_\psi J_\tau(\pi_{\theta_1}, \psi_{\rho,\tau}^\star(\pi_{\theta_1}))\|^2$$

$$\leq \frac{L^2 |\mathcal{S}|}{2\tau(1-\gamma) (\min_{s,a} \phi_\tau(\pi_{\theta_1})(a \mid s))^2} \|\theta_1 - \theta_2\|^2,$$

where the last inequality follows from the fact that for any $\theta_1, \theta_2 \in \mathbb{R}^{|\mathcal{S}| \times |\mathcal{A}|}$, $\psi_1, \psi_2 \in \mathbb{R}^{|\mathcal{S}| \times |\mathcal{B}|}$

$$\|\nabla_\psi J_\tau(\pi_{\theta_1}, \phi_{\psi_1}) - \nabla_\psi J_\tau(\pi_{\theta_2}, \phi_{\psi_2})\| \leq \|\nabla_\psi J(\pi_{\theta_1}, \phi_{\psi_1}) - \nabla_\psi J(\pi_{\theta_2}, \phi_{\psi_2})\|$$
$$+ \tau \|\nabla_\psi \mathcal{H}_\pi(s, \pi_{\theta_1}, \phi_{\psi_1}) - \nabla_\psi \mathcal{H}_\pi(s, \pi_{\theta_2}, \phi_{\psi_2})\|$$
$$+ \tau \|\nabla_\psi \mathcal{H}_\phi(s, \pi_{\theta_1}, \phi_{\psi_1}) - \nabla_\psi \mathcal{H}_\phi(s, \pi_{\theta_2}, \phi_{\psi_2})\|$$
$$\leq L(\|\theta_1 - \theta_2\| + \|\psi_1 - \psi_2\|), \tag{64}$$

which is a result of Lemmas 5 and 6.

By Lemma 1, we also have

$$J_\tau(\pi_{\theta_2}, \phi_\tau(\pi_{\theta_1})) - J_\tau(\pi_{\theta_2}, \phi_\tau(\pi_{\theta_2})) \geq \frac{\tau \rho_{\min}}{2 \log(2)} \|\phi_\tau(\pi_{\theta_1}) - \phi_\tau(\pi_{\theta_2})\|^2.$$

Combining the two inequalities and re-arranging the terms, we have

$$\|\phi_\tau(\pi_{\theta_1}) - \phi_\tau(\pi_{\theta_2})\| \leq \frac{\sqrt{|\mathcal{S}| \log(2)} L}{\sqrt{(1-\gamma)\rho_{\min}} \tau (\min_{s,a} \phi_\tau(\pi_{\theta_1})(a \mid s))} \|\theta_1 - \theta_2\|. \tag{65}$$

Therefore, by (17),

$$\|\nabla_\theta J_\tau(\pi_{\theta_k}, \phi_\tau(\pi_{\theta_k})) - \nabla_\theta J_\tau(\pi_{\theta_{k+1}}, \phi_\tau(\pi_{\theta_{k+1}}))\|$$
$$\leq L\|\theta_k - \theta_{k+1}\| + L\|\phi_\tau(\pi_{\theta_k}) - \phi_\tau(\pi_{\theta_{k+1}})\|$$
$$\leq L \left( 1 + \frac{\sqrt{|\mathcal{S}| \log(2)} L}{\sqrt{(1-\gamma)\rho_{\min}} \tau (\min_{s,a} \phi_\tau(\pi_{\theta_k})(a \mid s))} \right) \|\theta_k - \theta_{k+1}\|$$

Due to the Danskin's Theorem (16), this implies that we can perform the expansion

$$J_\tau(\pi_{\theta_k}, \phi_\tau(\pi_{\theta_k})) - J_\tau(\pi_{\theta_{k+1}}, \phi_\tau(\pi_{\theta_{k+1}}))$$
$$\leq -\langle \nabla_\theta J_\tau(\pi_{\theta_k}, \phi_\tau(\pi_{\theta_k})), \theta_{k+1} - \theta_k \rangle$$
$$+ \frac{L}{2} \left( 1 + \frac{\sqrt{|\mathcal{S}| \log(2)} L}{\sqrt{(1-\gamma)\rho_{\min}} \tau (\min_{s,a} \phi_\tau(\pi_{\theta_k})(a \mid s))} \right) \|\theta_{k+1} - \theta_k\|^2$$
$$\leq -\alpha_k \langle \nabla_\theta J_\tau(\pi_{\theta_k}, \phi_\tau(\pi_{\theta_k})), \nabla_\theta J_\tau(\pi_{\theta_k}, \phi_{\psi_k}) \rangle$$
$$+ \frac{L\alpha_k^2}{2} \left( 1 + \frac{\sqrt{|\mathcal{S}| \log(2)} L}{\sqrt{(1-\gamma)\rho_{\min}} \tau (\min_{s,a} \phi_\tau(\pi_{\theta_k})(a \mid s))} \right) \|\nabla_\theta J_\tau(\pi_{\theta_k}, \phi_{\psi_k})\|^2. \tag{66}$$

Note that by the property of the min function

$$\min_{s,a} \phi_\tau(\pi_{\theta_k})(a \mid s) \geq \min_{s,a} \phi_\tau^\star(a \mid s) - \max_{s,a}(\phi_\tau^\star(a \mid s) - \phi_\tau(\pi_{\theta_k})(a \mid s))$$

$$\geq \min_{s,a} \phi_\tau^\star(a \mid s) - \|\phi_\tau^\star - \phi_\tau(\pi_{\theta_k})\|$$

$$\geq c - \sqrt{\frac{2\log(2)}{\tau\rho_{\min}}(\delta_k^\pi + \delta_k^\phi)}, \tag{67}$$

where the last inequality uses the same argument as in (72). Since (37) implies $\delta_k^\pi + \delta_k^\phi \leq \frac{\rho_{\min}c^2\tau}{64\log(2)(k+1)^{1/3}}$, we further have

$$\min_{s,a} \phi_\tau(\pi_{\theta_k})(a \mid s) \geq c - \sqrt{\frac{2\log(2)}{\tau\rho_{\min}}(\delta_k^\pi + \delta_k^\phi)} \geq c(1 - \sqrt{\frac{1}{32}}) \geq \frac{c\sqrt{\log(2)}}{2}.$$

Using this bound in (66),

$$J_\tau(\pi_{\theta_k}, \phi_\tau(\pi_{\theta_k})) - J_\tau(\pi_{\theta_{k+1}}, \phi_\tau(\pi_{\theta_{k+1}}))$$
$$\leq -\alpha_k \langle \nabla_\theta J_\tau(\pi_{\theta_k}, \phi_\tau(\pi_{\theta_k})), \nabla_\theta J_\tau(\pi_{\theta_k}, \phi_{\psi_k}) \rangle$$
$$+ \frac{L\alpha_k^2}{2}\left(1 + \frac{\sqrt{|\mathcal{S}|\log(2)}L}{\sqrt{(1-\gamma)\rho_{\min}}\tau\,(\min_{s,a}\phi_\tau(\pi_{\theta_1})(a \mid s))}\right)\|\nabla_\theta J_\tau(\pi_{\theta_k}, \phi_{\psi_k})\|^2$$
$$\leq -\alpha_k \langle \nabla_\theta J_\tau(\pi_{\theta_k}, \phi_\tau(\pi_{\theta_k})), \nabla_\theta J_\tau(\pi_{\theta_k}, \phi_{\psi_k}) \rangle$$
$$+ \frac{L\alpha_k^2}{2}\left(1 + \frac{2\sqrt{|\mathcal{S}|}L}{\sqrt{(1-\gamma)\rho_{\min}}\tau c}\right)\|\nabla_\theta J_\tau(\pi_{\theta_k}, \phi_{\psi_k})\|^2, \tag{68}$$

With the step size choice $\alpha_k \leq \left(L + \frac{2\sqrt{|\mathcal{S}|}L^2}{\sqrt{(1-\gamma)\rho_{\min}}\tau c}\right)^{-1}$, we get

$$J_\tau(\pi_{\theta_k}, \phi_\tau(\pi_{\theta_k})) - J_\tau(\pi_{\theta_{k+1}}, \phi_\tau(\pi_{\theta_{k+1}}))$$
$$\leq -\alpha_k \langle \nabla_\theta J_\tau(\pi_{\theta_k}, \phi_\tau(\pi_{\theta_k})), \nabla_\theta J_\tau(\pi_{\theta_k}, \phi_{\psi_k}) \rangle$$
$$+ \frac{L\alpha_k^2}{2}\left(1 + \frac{2\sqrt{|\mathcal{S}|}L}{\sqrt{(1-\gamma)\rho_{\min}}\tau c}\right)\|\nabla_\theta J_\tau(\pi_{\theta_k}, \phi_{\psi_k})\|^2$$
$$\leq -\alpha_k \langle \nabla_\theta J_\tau(\pi_{\theta_k}, \phi_\tau(\pi_{\theta_k})), \nabla_\theta J_\tau(\pi_{\theta_k}, \phi_{\psi_k}) \rangle$$
$$+ \frac{\alpha_k}{2}\|\nabla_\theta J_\tau(\pi_{\theta_k}, \phi_{\psi_k})\|^2$$
$$= \frac{\alpha_k}{2}\|\nabla_\theta J_\tau(\pi_{\theta_k}, \phi_\tau(\pi_{\theta_k})) - \nabla_\theta J_\tau(\pi_{\theta_k}, \phi_{\psi_k})\|^2 - \|\nabla_\theta J_\tau(\pi_{\theta_k}, \phi_\tau(\pi_{\theta_k}))\|^2.$$

$$\square$$

### B.10 Proof of Lemma 10

The property of the min and max function implies that

$$\max_{s,a}(\pi_{\tau_k}^\star(a \mid s) - \pi_{\theta_k}(a \mid s)) + \min_{s,a} \pi_{\theta_k}(a \mid s) \geq \min_{s,a} \pi_{\tau_k}^\star(a \mid s).$$

Since the three terms are all non-negative, the inequality holds after taking the square

$$(\min_{s,a} \pi_{\tau_k}^\star(a \mid s))^2 \leq (\max_{s,a}(\pi_{\tau_k}^\star(a \mid s) - \pi_{\theta_k}(a \mid s)) + \min_{s,a} \pi_{\theta_k}(a \mid s))^2$$

$$\leq \frac{4}{3}(\min_{s,a} \pi_{\theta_k}(a \mid s))^2 + 4(\max_{s,a}(\pi_{\tau_k}^\star(a \mid s) - \pi_{\theta_k}(a \mid s)))^2.$$

Re-arranging the terms,

$$-\left(\min_{s,a} \pi_{\theta_k}(a \mid s)\right)^2 \leq -\frac{3}{4}\left(\min_{s,a} \pi_{\tau_k}^\star(a \mid s)\right)^2 + 3\left(\max_{s,a} \pi_{\tau_k}^\star(a \mid s) - \pi_{\phi_k}(a \mid s)\right)^2$$

$$\leq -\frac{3}{4}\left(\min_{s,a}\pi^\star_{\tau_k}(a\mid s)\right)^2 + 3\|\pi^\star_{\tau_k} - \pi_{\phi_k}\|^2$$

From Lemma 1,

$$-\left(\min_{s,a}\pi_{\theta_k}(a\mid s)\right)^2 \leq -\frac{3}{4}\left(\min_{s,a}\pi^\star_{\tau_k}(a\mid s)\right)^2 + 3\|\pi^\star_{\tau_k} - \pi_{\phi_k}\|^2$$

$$\leq -\frac{3}{4}\left(\min_{s,a}\pi^\star_{\tau_k}(a\mid s)\right)^2 + \frac{6\log(2)}{\tau_k\rho_{\min}}(J_{\tau_k}(\pi^\star_{\tau_k},\phi^\star_{\tau_k}) - J_{\tau_k}(\pi_{\theta_k},\phi^\star_{\tau_k}))$$

$$\leq -\frac{3}{4}\left(\min_{s,a}\pi^\star_{\tau_k}(a\mid s)\right)^2 + \frac{6\log(2)}{\tau_k\rho_{\min}}(J_{\tau_k}(\pi^\star_{\tau_k},\phi^\star_{\tau_k}) - J_{\tau_k}(\pi_{\theta_k},\phi_{\tau_k}(\pi_{\theta_k})))$$

$$= -\frac{3}{4}\left(\min_{s,a}\pi^\star_{\tau_k}(a\mid s)\right)^2 + \frac{6\log(2)}{\tau_k\rho_{\min}}\delta^\pi_k, \tag{69}$$

Since $3\delta^\pi_k + \delta^\phi_k \leq \frac{\rho\tau_k c^2}{64\log(2)}$, we have $\delta^\pi_k \leq \frac{\rho\tau_k c^2}{64\log(2)}$, which along with (69) implies

$$-\left(\min_{s,a}\pi_{\theta_k}(a\mid s)\right)^2 \leq -\frac{3}{4}\left(\min_{s,a}\pi^\star_{\tau_k}(a\mid s)\right)^2 + \frac{6\log(2)}{\tau_k\rho_{\min}}\delta^\pi_k \leq -\frac{3c^2}{4} + \frac{3c^2}{32} \leq -\frac{3c^2}{8}.$$

Similarly, the property of the min and max function implies that

$$\max_{s,b}(\phi^\star_{\tau_k}(b\mid s) - \phi_{\psi_k}(b\mid s)) + \min_{s,b}\phi_{\psi_k}(b\mid s) \geq \min_{s,b}\phi^\star_{\tau_k}(b\mid s).$$

Again, all three terms are non-negative, which means that the inequality is preserved after taking the square

$$(\min_{s,b}\phi^\star_{\tau_k}(b\mid s))^2 \leq (\min_{s,b}\phi_{\psi_k}(b\mid s) + \max_{s,b}(\phi^\star_{\tau_k}(b\mid s) - \phi_{\psi_k}(b\mid s)))^2$$

$$\leq \frac{4}{3}(\min_{s,b}\phi_{\psi_k}(b\mid s))^2 + 4(\max_{s,b}(\phi^\star_{\tau_k}(b\mid s) - \phi_{\psi_k}(b\mid s)))^2,$$

which leads to

$$-(\min_{s,b}\phi_{\psi_k}(b\mid s))^2 \leq -\frac{3}{4}(\min_{s,b}\phi^\star_{\tau_k}(b\mid s))^2 + 3(\max_{s,b}(\phi^\star_{\tau_k}(b\mid s) - \phi_{\psi_k}(b\mid s)))^2$$

$$\leq -\frac{3}{4}(\min_{s,b}\phi^\star_{\tau_k}(b\mid s))^2 + 3\|\phi^\star_{\tau_k} - \phi_{\psi_k}\|^2$$

$$\leq -\frac{3}{4}(\min_{s,b}\phi^\star_{\tau_k}(b\mid s))^2 + 6\|\phi_{\tau_k}(\pi_{\theta_k}) - \phi_{\psi_k}\|^2 + 6\|\phi^\star_{\tau_k} - \phi_{\tau_k}(\pi_{\theta_k})\|^2. \tag{70}$$

From Lemma 1,

$$\|\phi_{\tau_k}(\pi_{\theta_k}) - \phi_{\psi_k}\|^2 \leq \frac{2\log(2)}{\tau_k\rho_{\min}}\left(J_{\tau_k}(\pi_{\theta_k},\phi_{\psi_k}) - J_{\tau_k}(\pi_{\theta_k},\phi_{\tau_k}(\pi_{\theta_k}))\right) = \frac{2\log(2)}{\tau_k\rho_{\min}}\delta^\phi_k, \tag{71}$$

and

$$\|\phi^\star_{\tau_k} - \phi_{\tau_k}(\pi_{\theta_k})\|^2$$

$$\leq \frac{2\log(2)}{\tau_k\rho_{\min}}\left(J_{\tau_k}(\pi_{\theta_k},\phi^\star_{\tau_k}) - J_{\tau_k}(\pi_{\theta_k},\phi_{\tau_k}(\pi_{\theta_k}))\right)$$

$$\leq \frac{2\log(2)}{\tau_k\rho_{\min}}\left(\left(J_{\tau_k}(\pi_{\theta_k},\phi^\star_{\tau_k}) - J_{\tau_k}(\pi_{\theta_k},\phi_{\psi_k})\right) + \underbrace{\left(J_{\tau_k}(\pi_{\theta_k},\phi_{\psi_k}) - J_{\tau_k}(\pi_{\theta_k},\phi_{\tau_k}(\pi_{\theta_k}))\right)}_{\delta^\phi_k}\right)$$

$$= \frac{2\log(2)}{\tau_k\rho_{\min}}\left(\left(J_{\tau_k}(\pi_{\theta_k},\phi^\star_{\tau_k}) - J_{\tau_k}(\pi_{\theta_k},\phi_{\tau_k}(\pi_{\theta_k}))\right) + \left(J_{\tau_k}(\pi_{\theta_k},\phi_{\tau_k}(\pi_{\theta_k})) - J_{\tau_k}(\pi_{\theta_k},\phi_{\psi_k})\right) + \delta^\phi_k\right)$$

$$\leq \frac{2\log(2)}{\tau_k \rho_{\min}} \left( J_{\tau_k}(\pi_{\theta_k}, \phi_{\tau_k}^\star) - J_{\tau_k}(\pi_{\theta_k}, \phi_{\tau_k}(\pi_{\theta_k})) + \delta_k^\phi \right)$$

$$\leq \frac{2\log(2)}{\tau_k \rho_{\min}} \left( J_{\tau_k}(\pi_{\tau_k}^\star, \phi_{\tau_k}^\star) - J_{\tau_k}(\pi_{\theta_k}, \phi_{\tau_k}(\pi_{\theta_k})) + \delta_k^\phi \right)$$

$$= \frac{2\log(2)}{\tau_k \rho_{\min}} \left( \delta_k^\pi + \delta_k^\phi \right), \tag{72}$$

where the third inequality follows from $J_{\tau_k}(\pi_{\theta_k}, \phi_{\tau_k}(\pi_{\theta_k})) - J_{\tau_k}(\pi_{\theta_k}, \phi_{\psi_k}) \leq 0$.

Using (71) and (72) in (70),

$$-(\min_{s,b} \phi_{\psi_k}(b \mid s))^2 \leq -\frac{3}{4}(\min_{s,b} \phi_{\tau_k}^\star(b \mid s))^2 + 6\|\phi_{\tau_k}(\pi_{\theta_k}) - \phi_{\psi_k}\|^2 + 6\|\phi_{\tau_k}^\star - \phi_{\tau_k}(\pi_{\theta_k})\|^2$$

$$\leq -\frac{3}{4}(\min_{s,b} \phi_{\tau_k}^\star(b \mid s))^2 + \frac{12\log(2)}{\tau_k \rho_{\min}} \delta_k^\phi + \frac{12\log(2)}{\tau_k \rho_{\min}}(\delta_k^\pi + \delta_k^\phi)$$

$$= -\frac{3}{4}(\min_{s,b} \phi_{\tau_k}^\star(b \mid s))^2 + \frac{12\log(2)}{\tau_k \rho_{\min}}(\delta_k^\pi + 2\delta_k^\phi).$$

$3\delta_k^\pi + \delta_k^\phi \leq \frac{\rho\tau_k c^2}{64\log(2)}$ implies that $\delta_k^\pi + 2\delta_k^\phi \leq \frac{\rho\tau_k c^2}{32\log(2)}$. Using this in the inequality above,

$$-(\min_{s,b} \phi_{\psi_k}(b \mid s))^2 \leq -\frac{3}{4}(\min_{s,b} \phi_{\tau_k}^\star(b \mid s))^2 + \frac{12\log(2)}{\tau_k \rho_{\min}}(\delta_k^\pi + 2\delta_k^\phi) \leq -\frac{3c^2}{4} + \frac{12c^2}{32} \leq -\frac{3c^2}{8}.$$

$\square$

## B.11 Proof of Lemma 11

From Lemma 4, for any $\psi \in \mathbb{R}^{|\mathcal{S}| \times |\mathcal{B}|}$

$$J_{\tau_k}(\pi_{\theta_2}, \phi_\psi) - J_{\tau_k}(\pi_{\theta_2}, \phi_{\tau_k}(\pi_{\theta_2}))$$

$$\leq \frac{|\mathcal{S}|}{2\tau_k \rho_{\min} (\min_{s,a} \phi_\psi(a \mid s))^2} \left\| \frac{d_\rho^{\pi_{\theta_2}, \phi_{\tau_k}(\pi_{\theta_2})}}{d_\rho^{\pi_{\theta_2}, \phi_\psi}} \right\|_\infty \|\nabla_\psi J_{\tau_k}(\pi_{\theta_2}, \phi_\psi)\|^2$$

$$\leq \frac{|\mathcal{S}|}{2\tau_k (1-\gamma) (\min_{s,a} \phi_\psi(a \mid s))^2} \|\nabla_\psi J_{\tau_k}(\pi_{\theta_2}, \phi_\psi)\|^2,$$

where the second inequality follows by an argument similar to (48). Letting $\psi$ be the parameter that parameterizes $\phi_{\tau_k}(\pi_{\theta_1})$ and defining $L_k = L_{\mathcal{H}}(2\tau_k + 1)$, we have

$$J_{\tau_k}(\pi_{\theta_2}, \phi_{\tau_k}(\pi_{\theta_1})) - J_{\tau_k}(\pi_{\theta_2}, \phi_{\tau_k}(\pi_{\theta_2}))$$

$$\leq \frac{|\mathcal{S}|}{2\tau_k (1-\gamma) (\min_{s,a} \phi_{\tau_k}(\pi_{\theta_1})(a \mid s))^2} \|\nabla_\psi J_{\tau_k}(\pi_{\theta_2}, \phi_{\tau_k}(\pi_{\theta_1}))\|^2$$

$$= \frac{|\mathcal{S}|}{2\tau_k (1-\gamma) (\min_{s,a} \phi_{\tau_k}(\pi_{\theta_1})(a \mid s))^2} \|\nabla_\psi J_{\tau_k}(\pi_{\theta_2}, \psi_{\rho,\tau_k}^\star(\pi_{\theta_1})) - \nabla_\psi J_{\tau_k}(\pi_{\theta_1}, \psi_{\rho,\tau_k}^\star(\pi_{\theta_1}))\|^2$$

$$\leq \frac{L_k^2 |\mathcal{S}|}{2\tau_k (1-\gamma) (\min_{s,a} \phi_{\tau_k}(\pi_{\theta_1})(a \mid s))^2} \|\theta_1 - \theta_2\|^2,$$

where the last inequality uses the same argument as (64).

By Lemma 1, we also have

$$J_{\tau_k}(\pi_{\theta_2}, \phi_{\tau_k}(\pi_{\theta_1})) - J_{\tau_k}(\pi_{\theta_2}, \phi_{\tau_k}(\pi_{\theta_2})) \geq \frac{\tau_k \rho_{\min}}{2\log(2)} \|\phi_{\tau_k}(\pi_{\theta_1}) - \phi_{\tau_k}(\pi_{\theta_2})\|^2.$$

Combining the two inequalities and re-arranging the terms, we have

$$\|\phi_{\tau_k}(\pi_{\theta_1}) - \phi_{\tau_k}(\pi_{\theta_2})\| \leq \frac{\sqrt{|\mathcal{S}| \log(2)} L_k}{\sqrt{(1-\gamma)\rho_{\min}} \tau_k (\min_{s,a} \phi_{\tau_k}(\pi_{\theta_1})(a \mid s))} \|\theta_1 - \theta_2\|. \tag{73}$$

Therefore, by (17),

$$\|\nabla_\theta J_{\tau_k}(\pi_{\theta_k}, \phi_{\tau_k}(\pi_{\theta_k})) - \nabla_\theta J_{\tau_k}(\pi_{\theta_{k+1}}, \phi_{\tau_k}(\pi_{\theta_{k+1}}))\|$$
$$\leq L_k \|\theta_k - \theta_{k+1}\| + L_k \|\phi_{\tau_k}(\pi_{\theta_k}) - \phi_{\tau_k}(\pi_{\theta_{k+1}})\|$$
$$\leq L_k \left(1 + \frac{\sqrt{|\mathcal{S}| \log(2)} L_k}{\sqrt{(1-\gamma)\rho_{\min}} \tau_k \left(\min_{s,a} \phi_{\tau_k}(\pi_{\theta_k})(a \mid s)\right)}\right) \|\theta_k - \theta_{k+1}\|$$

Due to the Danskin's Theorem (16), this implies that we can perform the expansion

$$J_{\tau_k}(\pi_{\theta_k}, \phi_{\tau_k}(\pi_{\theta_k})) - J_{\tau_k}(\pi_{\theta_{k+1}}, \phi_{\tau_k}(\pi_{\theta_{k+1}}))$$
$$\leq -\langle \nabla_\theta J_{\tau_k}(\pi_{\theta_k}, \phi_{\tau_k}(\pi_{\theta_k})), \theta_{k+1} - \theta_k \rangle$$
$$\quad + \frac{L_k}{2} \left(1 + \frac{\sqrt{|\mathcal{S}| \log(2)} L_k}{\sqrt{(1-\gamma)\rho_{\min}} \tau_k \left(\min_{s,a} \phi_{\tau_k}(\pi_{\theta_k})(a \mid s)\right)}\right) \|\theta_{k+1} - \theta_k\|^2$$
$$\leq -\alpha_k \langle \nabla_\theta J_{\tau_k}(\pi_{\theta_k}, \phi_{\tau_k}(\pi_{\theta_k})), \nabla_\theta J_{\tau_k}(\pi_{\theta_k}, \phi_{\psi_k}) \rangle$$
$$\quad + \frac{L_k \alpha_k^2}{2} \left(1 + \frac{\sqrt{|\mathcal{S}| \log(2)} L_k}{\sqrt{(1-\gamma)\rho_{\min}} \tau_k \left(\min_{s,a} \phi_{\tau_k}(\pi_{\theta_k})(a \mid s)\right)}\right) \|\nabla_\theta J_{\tau_k}(\pi_{\theta_k}, \phi_{\psi_k})\|^2. \quad (74)$$

Note that by the property of the min function

$$\min_{s,a} \phi_{\tau_k}(\pi_{\theta_k})(a \mid s) \geq \min_{s,a} \phi^\star_{\tau_k}(a \mid s) - \max_{s,a}(\phi^\star_{\tau_k}(a \mid s) - \phi_{\tau_k}(\pi_{\theta_k})(a \mid s))$$
$$\geq \min_{s,a} \phi^\star_{\tau_k}(a \mid s) - \|\phi^\star_{\tau_k} - \phi_{\tau_k}(\pi_{\theta_k})\|$$
$$\geq c - \sqrt{\frac{2\log(2)}{\tau_k \rho_{\min}}(\delta_k^\pi + \delta_k^\phi)}, \quad (75)$$

where the last inequality uses the same argument as in (72). Since (37) implies $\delta_k^\pi + \delta_k^\phi \leq \frac{\rho_{\min} c^2 \tau_0}{64 \log(2)(k+1)^{1/3}}$, we further have

$$\min_{s,a} \phi_{\tau_k}(\pi_{\theta_k})(a \mid s) \geq c - \sqrt{\frac{2\log(2)}{\tau_k \rho_{\min}}(\delta_k^\pi + \delta_k^\phi)} \geq c\left(1 - \sqrt{\frac{1}{32}}\right) \geq \frac{c\sqrt{\log(2)}}{2}.$$

Using this bound in (74),

$$J_{\tau_k}(\pi_{\theta_k}, \phi_{\tau_k}(\pi_{\theta_k})) - J_{\tau_k}^{\pi_{\theta_{k+1}}, \phi_{\tau_k}(\pi_{\theta_{k+1}})}(\rho)$$
$$\leq -\alpha_k \langle \nabla_\theta J_{\tau_k}(\pi_{\theta_k}, \phi_{\tau_k}(\pi_{\theta_k})), \nabla_\theta J_{\tau_k}(\pi_{\theta_k}, \phi_{\psi_k}) \rangle$$
$$\quad + \frac{L_k \alpha_k^2}{2} \left(1 + \frac{\sqrt{|\mathcal{S}| \log(2)} L_k}{\sqrt{(1-\gamma)\rho_{\min}} \tau_k \left(\min_{s,a} \phi_{\tau_k}(\pi_{\theta_1})(a \mid s)\right)}\right) \|\nabla_\theta J_{\tau_k}(\pi_{\theta_k}, \phi_{\psi_k})\|^2$$
$$\leq -\alpha_k \langle \nabla_\theta J_{\tau_k}(\pi_{\theta_k}, \phi_{\tau_k}(\pi_{\theta_k})), \nabla_\theta J_{\tau_k}(\pi_{\theta_k}, \phi_{\psi_k}) \rangle$$
$$\quad + \frac{L_k \alpha_k^2}{2} \left(1 + \frac{2\sqrt{|\mathcal{S}|} L_k}{\sqrt{(1-\gamma)\rho_{\min}} \tau_k c}\right) \|\nabla_\theta J_{\tau_k}(\pi_{\theta_k}, \phi_{\psi_k})\|^2$$
$$\leq -\alpha_k \langle \nabla_\theta J_{\tau_k}(\pi_{\theta_k}, \phi_{\tau_k}(\pi_{\theta_k})), \nabla_\theta J_{\tau_k}(\pi_{\theta_k}, \phi_{\psi_k}) \rangle + \frac{\alpha_k^2}{2}\left(L_k + \frac{C_2 L_k^2}{\tau_k}\right) \|\nabla_\theta J_{\tau_k}(\pi_{\theta_k}, \phi_{\psi_k})\|^2.$$
$$(76)$$

The condition on $h$, which is $\frac{\alpha_0}{h^{2/3}} \leq (2L_\mathcal{H} + 4L_\mathcal{H}^2 C_2)\frac{\tau_0}{h^{1/3}} + (L_\mathcal{H} + 4L_\mathcal{H}^2 C_2) + \frac{L_\mathcal{H}^2 C_2 h^{1/3}}{\tau_0}$, can be equivalently expressed as $\alpha_0 \left(L_0 + \frac{C_2 L_0^2}{\tau_0}\right) \leq 1$. Since $\alpha_k$ decays faster than $\tau_k$, this guarantees for all $k \geq 0$

$$\alpha_k \left(L_k + \frac{C_2 L_k^2}{\tau_k}\right) \leq 1.$$

Using this inequality in (76), we get

$$J_{\tau_k}(\pi_{\theta_k}, \phi_{\tau_k}(\pi_{\theta_k})) - J_{\tau_k}(\pi_{\theta_{k+1}}, \phi_{\tau_k}(\pi_{\theta_{k+1}}))$$

$$\leq -\alpha_k \langle \nabla_\theta J_{\tau_k}(\pi_{\theta_k}, \phi_{\tau_k}(\pi_{\theta_k})), \nabla_\theta J_{\tau_k}(\pi_{\theta_k}, \phi_{\psi_k}) \rangle + \frac{\alpha_k}{2} \|\nabla_\theta J_{\tau_k}(\pi_{\theta_k}, \phi_{\psi_k})\|^2$$

$$= \frac{\alpha_k}{2} \left( \|\nabla_\theta J_{\tau_k}(\pi_{\theta_k}, \phi_{\tau_k}(\pi_{\theta_k})) - \nabla_\theta J_{\tau_k}(\pi_{\theta_k}, \phi_{\psi_k})\|^2 - \|\nabla_\theta J_{\tau_k}(\pi_{\theta_k}, \phi_{\tau_k}(\pi_{\theta_k}))\|^2 \right).$$

$\square$

## C  Discussion on the Initial Condition for Corollary 1 and Theorem 2

We show that as $\tau_0 \to \infty$, both $\delta_0^\pi / \tau_0$ and $\delta_0^\phi / \tau_0$ approach 0. Decomposing $\delta_0^\pi / \tau_0$, we have

$$\frac{\delta_0^\pi}{\tau_0} = \frac{1}{\tau_0} \left( J(\pi_{\tau_0}^\star, \phi_{\tau_0}^\star) - J(\pi_{\theta_0}, \phi_{\tau_0}(\pi_{\theta_0})) \right) \tag{77}$$

$$+ \left( \mathcal{H}_\pi(\rho, \pi_{\tau_0}^\star, \phi_{\tau_0}^\star) - \mathcal{H}_\pi(\rho, \pi_{\theta_0}, \phi_{\tau_0}(\pi_{\theta_0})) \right) + \left( \mathcal{H}_\phi(\rho, \pi_{\tau_0}^\star, \phi_{\tau_0}^\star) - \mathcal{H}_\phi(\rho, \pi_{\theta_0}, \phi_{\tau_0}(\pi_{\theta_0})) \right) \tag{78}$$

The original value functions are bounded within $[0, \frac{1}{1-\gamma}]$, which implies that the term (77) decays inversely with $\tau_0$ in the worst case. When $\tau_0 \to \infty$, the Nash equilibrium policy pair $\pi_{\tau_0}^\star$ and $\phi_{\tau_0}^\star$ both approach the uniform distribution, and so does $\phi_{\tau_0}(\pi_{\theta_0})$. This means that (78) approaches 0. Therefore, as the sum of (77) and (78), $\delta_k^\pi / \tau_0$ decays to 0 as $\tau_0 \to \infty$. A similar argument can be used for $\delta_0^\phi / \tau_0$.

## D  Experiment Details

We first discuss the design of the completely mixed Markov game. The dimension of state space is 2, and so is the dimension of the action spaces of both players. Using $s_1$, $s_2$ to denote the two states, we can essentially describe $\mathcal{P}$ as a $2 \times 2 \times 2 \times 2$ tensor where $\mathcal{P}(s' \mid s, \cdot, \cdot)$ is a $2 \times 2$ matrix for any $s, s' \in \mathcal{S}$ with rows corresponding to the action of the first player and columns corresponding to the second player

$$\mathcal{P}(s_1 \mid s_1, \cdot, \cdot) = \begin{bmatrix} 0.2 & 0.5 \\ 0.5 & 0.1 \end{bmatrix}, \quad \mathcal{P}(s_2 \mid s_1, \cdot, \cdot) = \begin{bmatrix} 0.8 & 0.5 \\ 0.5 & 0.9 \end{bmatrix},$$

$$\mathcal{P}(s_1 \mid s_2, \cdot, \cdot) = \begin{bmatrix} 0.3 & 0.2 \\ 0.6 & 0.2 \end{bmatrix}, \quad \mathcal{P}(s_2 \mid s_2, \cdot, \cdot) = \begin{bmatrix} 0.7 & 0.8 \\ 0.4 & 0.8 \end{bmatrix}.$$

Similarly, the reward function can be described by a $2 \times 2 \times 2$ tensor where $r(s, \cdot, \cdot)$ is a $2 \times 2$ matrix for any $s \in \mathcal{S}$ with rows corresponding to the action of the first player and columns corresponding to the second player

$$r(s_1, \cdot, \cdot) = \begin{bmatrix} 1 & 2 \\ 2 & 1 \end{bmatrix}, \quad r(s_2, \cdot, \cdot) = \begin{bmatrix} 6 & 4 \\ 3 & 10 \end{bmatrix}.$$

Under the initial distribution $\rho = [0.5, 0.5]^\top$ and discount factor $\gamma = 0.9$, the (approximate) Nash equilibrium of this Markov game is

$$\pi^\star(\cdot \mid s_1) = [0.812, 0.188], \quad \pi^\star(\cdot \mid s_2) = [0.837, 0.163],$$
$$\phi^\star(\cdot \mid s_1) = [0.880, 0.120], \quad \phi^\star(\cdot \mid s_2) = [0.597, 0.403].$$

To design the Markov game that does not observe Assumption 2, we use the same transition probability matrices as in the completely mixed Markov game case. The reward function is

$$r(s_1, \cdot, \cdot) = \begin{bmatrix} 1 & 2 \\ 3 & 4 \end{bmatrix}, \quad r(s_2, \cdot, \cdot) = \begin{bmatrix} 1 & 2 \\ 3 & 4 \end{bmatrix}.$$

Under the initial distribution $\rho = [0.5, 0.5]^\top$ and discount factor $\gamma = 0.9$, it can be easily seen that the Nash equilibrium of this Markov game is unique and is

$$\pi^\star(\cdot \mid s_1) = [0, 1], \quad \pi^\star(\cdot \mid s_2) = [0, 1],$$
$$\phi^\star(\cdot \mid s_1) = [1, 0], \quad \phi^\star(\cdot \mid s_2) = [1, 0].$$

Since the Nash equilibrium consists of a pair of deterministic policies, Assumption 2 is not satisfied in this case.