# OpenReview forum: "Regularized Gradient Descent Ascent for Two-Player Zero-Sum Markov Games"
_NeurIPS.cc/2022/Conference — NeurIPS 2022 Accept_

### Official Review · Reviewer_j9tp · 2022-07-05

**Rating:** 7
**Confidence:** 4
**Soundness:** 4 excellent
**Presentation:** 4 excellent
**Contribution:** 3 good

**Summary:**

This paper shows last iterate convergence for regularized gradient descent-ascent on markov games with finite state and action space under the assumption that the smallest coefficients of the support of the regularized markov games are uniformly (uniformly lower bounded over the regularization parameter.) Additionally, the authors assume that the initial state distribution has a full support.

**Questions:**

- After (1), you mention that J is non-concave with respect to the policy of the first player. But it seems to me that J is actually linear (with respect to the policy) since we have $$V^{\lambda \pi_1 + (1-\lambda)\pi_2,\phi} = \lambda V^{\pi_1,\phi}  + (1-\lambda) V^{\pi_2,\phi}$$ (playing according to the mixed policy is equivalent to tossing a coin and playing according to either the first or the second policy). Can you comment on that point?
- What is your opinion on the optimal convergence rate? Are there lower bounds ? (since the problem is nonconcave-nonconvex in the parameters pre-softmax the standard convex-concave lower bounds do not hold)
- It seems that Assumption 2 could be connected to the Maximum entropy Nash (MaxEnt). (the lower bound should be on the support of the MaxEnt. One way to prove such a result could be by saying that the MaxEnt is the limit of the regularized Nash. Since the Regularized Nash always has full support (the regularization goes to +\infty for non-full support solutions) and its limit (the MaxEnt for no regularization and the uniform distribution for infinite regularization) also has full support, there exists a (uniform) lower bound on the support of any regularized Nash. However, it does not show that this lower bound is the lower bound on the MaxEnt (which seems intuitively true). Any comment on that?


I will be eager to increase my score if the authors answer my questions and are able to show that Assumption 2 can be relaxed.

**Limitations:**

- This work is limited to the full batch case which is not practical.
- The proposed rate may be suboptimal. It could be interesting to investigate experimentally different decays ($1/t$, $1/\sqrt{t}$).
- Also a log-log scale for the plots could help to visualize the experimental convergence rate.


**Strengths And Weaknesses:**


strengths:
- To the extent of my knowledge, the result seems novel and relevant to the community.
- This paper is well written.

weaknesses:
- Assumption 2 is strong and does not seem necessary.
- There is no discussion on the optimality of the rate proposed rate.
- Algorithm is not practical since one needs access to the (full) gradient of J.
- The authors could do more experiments to investigate these questions left open.

---

> ### Author Response · Authors · 2022-08-02
> **Response to Reviewer j9tp**
>
> a) In response to the reviewer's question on the non-concavity of the value function, we note that the linear relation only holds in matrix games (single-state Markov game). For general Markov games, the reward collected in the current state obeys this linear relation, but $\pi_1$ and $\pi_2$ can result in different transitions to the future states and the expected future reward is non-linear. In Lemma 3.1 of [Agarwal 2021], counter-examples show that the objective function is not convex or concave even in a single-agent MDP.
>
> b) Regarding the question on the tightness of our upper bound, we do not have any lower bound yet, and we are not aware of any result in the literature that we can invoke with modest changes to derive a lower bound. In fact, we do believe that our rate is likely sub-optimal. As we mention in the third bullet in our response to reviewer KhY2, value-based methods and policy optimization methods are two main classes of algorithms that achieve comparable convergence guarantees in single-agent MDP. Value-based methods for Markov games and MDPs have been shown to have similar convergence rates, but our rate $\mathcal{O}(k^{-1/3})$ is far from the optimal convergence rate of the policy gradient algorithm for MDPs.
>
> c) We thank the reviewer for the thoughtful comment on the connection of Assumption 2 to the MaxEnt Nash. We have investigated the connection similar to what the reviewer suggests. In a single-agent MDP, we can show that the smallest entry of the MaxEnt optimal policy is a lower bound on the optimal policy of regularized MDPs for any $\tau>0$. But in a two-player Markov game, this is not the case, and we have examples where the smallest entry of the MaxEnt Nash is larger than the the smallest entry of the regularized Nash under some small $\tau>0$. We do hope to formally translate Assumption 2 to a condition on the MaxEnt Nash. At the same time, as we discuss in the second bullet in our response to all reviewers, we are refining the PL-type condition derived in Lemma 4 to completely remove Assumption 2.
>
> d) On the inaccessibility the exact gradient in practice, we discuss in the first bullet in our response to all reviewers how sample-based extensions of our algorithm can be made with some extra technical but straightforward analysis.
>
> References
>
>
> Agarwal, Alekh, et al. "On the Theory of Policy Gradient Methods: Optimality, Approximation, and Distribution Shift." J. Mach. Learn. Res. 22.98 (2021): 1-76.

---

> > ### Comment · Reviewer_j9tp · 2022-08-08
> > **Thank you for your answer**
> >
> > a) thank you for your answer now I understand your point better.
> >
> > Here is mine:  The non-convexity depends on how you define the convex-combination of $\pi_1$ and $\pi_2$. By using the definition of [Agarwal 2021] $\pi_3 = .5(\pi_1+\pi_2)$ is defined as  $\pi_3(a|s) = .5\pi_1(a|s)+.5*\pi_2(a|s)$. In that case, I agree that $V^\pi$ may be non-linear.
> >
> > However, if you define the convex combination of $\pi_1$ and $\pi_2$ as the policy where you decide *at the beginning* which policy you will follow and then follow this policy *all along the trajectory*, then it seems that $V^\pi$ will be linear. More precisely $\pi_3 = .5(\pi_1+\pi_2)$ is defined as the policy that samples a benouilli with probability $.5$ and follow either $\pi_1$ or $\pi_2$ depending on the result on that bernouilli.
> >
> > Does it make sense?

---

> > > ### Author Response · Authors · 2022-08-08
> > > **Thanks for your follow-up.**
> > >
> > > We thank the reviewer for the follow-up. We agree that if a Bernoulli random variable with probability .5 is sampled and depending on the outcome $\pi_1$ or $\pi_2$ is used to generate the entire trajectory, then we get a value function of $.5(V^{\pi_1}+V^{\pi_2})$. In fact, this argument holds for any arbitrary function $V$ (even if it is not the objective function of a policy optimization problem).
> > >
> > > To show $V$ is linear, we want to show $\alpha V^{\pi_1}+(1-\alpha)V^{\pi_2} = V^{\alpha\pi_1+(1-\alpha)\pi_2}$. The $\pi_3$ suggested by the reviewer is actually not a linear combination of $\pi_1$ and $\pi_2$ but rather by construction the policy that produces the LHS of the equation, and the argument shows that the LHS equals the LHS and does not touch on the RHS.

---

### Official Review · Reviewer_7QJ7 · 2022-07-09

**Rating:** 6
**Confidence:** 4
**Soundness:** 3 good
**Presentation:** 3 good
**Contribution:** 2 fair

**Summary:**

This paper develops policy gradient algorithms for entropy-regularized zero-sum games. Under some assumptions, the authors show that the entropy-regularized objective function obeys a Polyak-Łojasiewicz type condition, which allows linear convergence of policy-gradient-based GDA to the equilibrium point of the regularized game. Then, the authors propose two scheduling schemes for the regularization parameter and analyze their convergence and complexities.

**Questions:**

Overall, the paper is well written, and the references are properly cited. I have the following questions/comments.

1. Entropy-regularized zero-sum game has been studied in the existing literature (Cen 2021), and I think this paper provides a policy-gradient-based solution and complexity analysis.

2. The convergence metric used in this paper is different from the duality gap metric used in Cen 2021, is there a connection between these two metrics? How do your convergence rate and complexity compare with those established in Cen 2021?

3. Comparing with the analysis of Cen 2021, the authors stated that they discuss and leverage structure in the regularized Markov
game that was previously unknown. While I agree that the policy gradient approach and PL condition of the Entropy-regularized zero-sum game were not discussed in Cen 2021, their analysis actually leverages certain similar structures implicitly. Specifically, Cen 2021 follows the standard value iteration algorithm that has a linear convergence rate. In the inner loop, they need to solve an entropy-regularized bilinear matrix game, which satisfies a strongly-convex-strongly-concave type geometry due to the entropy regularization (w.r.t to the $\ell_1$ norm). In their convergence proof, this geometry helps them achieve a linear convergence.

**Limitations:**

Yes

**Strengths And Weaknesses:**

Strength: comprehensive theoretical result.

Weakness: It is still not clear why Assumption 2 is needed and justifiable.

---

> ### Author Response · Authors · 2022-08-02
> **Response to Reviewer 7QJ7**
>
> a) On the connection of the work to [Cen 2021], we note that as the reviewer described, [Cen 2021] is essentially a value-based method which uses value iteration in the outer loop and runs an extragradient algorithm only in the inner loop to solve a regularized bimatrix game. [Perolat 2015] is one of the first works to extend value-based methods from single-agent MDP to two-player Markov games. Since then, the basic techniques for analyzing value-based methods are known. In fact, the key to the linear convergence established in [Cen 2021] is a contraction property that their value-based method takes advantage of. On the other hand, policy optimization algorithms are much less understood in Markov games, but this is an important subject to study due to the wide use of such algorithms in practice. Our analysis treats the Markov game purely from the optimization perspective and is very different from the analysis in [Cen 2021] in nature, though a small portion of the steps may look similar as both our work and [Cen 2021] need to take advantage of the structure of the game. In single-agent MDPs, value-based methods and policy optimization methods have comparable convergence guarantees today, and our work aims to narrow the gap between the understanding of these two classes of algorithms in two-player Markov games. An additional difference in the algorithm structure is that our algorithm is a single-loop algorithm which may be more convenient to implement in practice than nested-loop algorithms.
>
> b) On the difference in convergence metric compared to [Cen 2021], both our metric and the duality gap considered in [Cen 2021] are common metrics in the literature. The duality gap metric arises more naturally when value-based algorithms are considered. Our metric is more connected to the bi-level/Stackelberg viewpoint where the game is treated as a bi-level optimization problem. Our convergence metric being 0 for a policy pair implies that the policy pair is the Nash equilibrium. The same is true under the duality gap metric.

---

### Official Review · Reviewer_KhY2 · 2022-07-12

**Rating:** 4
**Confidence:** 3
**Soundness:** 3 good
**Presentation:** 3 good
**Contribution:** 2 fair

**Summary:**

This paper provided a new algorithm to find the Nash equilibrium for a two-player zero-sum Markov game. The new algorithm is based on a gradient descent with KL regularization. For completely mixed Markov games, the authors prove that the algorithm can find Nash equilibrium efficiently. They also conduct a simple experiment for tabular Markov games with low dimensions. The experiments support their main theoretical results and show that their algorithm may also work for the Markov games that are not completely mixed.

**Questions:**

1. How can we access the gradient $\nabla J$?

2. Can this algorithm be applied to Markov games with more complex structures?

3. What's the advantage of the new algorithm compared with other algorithms that are designed for tabular settings such as [4]?


[4] Bai, Yu, and Chi Jin. "Provable self-play algorithms for competitive reinforcement learning." International conference on machine learning. PMLR, 2020.

**Limitations:**

The authors have addressed their work's limitations and potential negative social impact.

**Strengths And Weaknesses:**

This paper is well organized, and the proofs seem sound to me. However, I still have several concerns about the clarity and significance of the main results.

1. The main result of this paper only works for "completely mixed Markov games." I would recommend the authors indicate that in the title. Otherwise, it would be kind of misleading. Moreover, completely mixed Markov games seem quite strong in the applications, which may limit the significance of the theoretical result.

2. The authors conduct experiments and show that their algorithm may work for Markov games without completely mixed assumptions. However, the experiment is too simple to support their claims seems it is a tubular Markov game with dimension 2.

3. One of the biggest advantages of using gradient descent to solve Markov games is that it can deal with complex models. However, this paper only conducted simple tubular Markov games by using a tabular softmax policy parameterization. Can the algorithm be applied to Markov games with more complex structures, such as with linear function approximation [1,2] or general function approximation [3]?

4. This paper assumes that we can access the expected value function during the training and doesn't consider the exploration problem.


[1] Xie, Qiaomin, et al. "Learning zero-sum simultaneous-move Markov games using function approximation and correlated equilibrium." Conference on learning theory. PMLR, 2020.

[2] Chen, Zixiang, Dongruo Zhou, and Quanquan Gu. "Almost optimal algorithms for two-player Markov games with linear function approximation." arXiv preprint arXiv:2102.07404 (2021).

[3] Huang, Baihe, et al. "Towards general function approximation in zero-sum Markov games." arXiv preprint arXiv:2107.14702 (2021).

---

> ### Author Response · Authors · 2022-08-02
> **Response to Reviewer KhY2**
>
> a) Regarding how to compute $\nabla J$, we note that the main bottlenecks are the discounted visitation distribution and the advantage value function. In small systems where we have explicit knowledge of the transition probability kernel, these quantities can be solved with matrix inversion and the exact gradient is accessible. When we do not explicitly know the transition kernel and can only take samples according to it, sample-based algorithms such as REINFORCE and actor-critic can be used to estimate $\nabla J$. In our first bullet in the response to all reviewers, we discuss how these sample-based extensions are straightforward extensions of the deterministic gradient method considered in this paper.
>
> b) On whether the algorithm can be applied with function approximations, we note that most of our analytical techniques should still be applicable, but a key necessary change is to replace Lemma 4 with a gradient domination condition derived for the specific function approximation. We agree with the reviewer that solving the problem beyond the tabular setting under linear or general function approximations is interesting, but we believe that our result stands on its own as the foundation and the analysis for function approximations can be built on the top.
>
> c) On the connection and novelty of our work compared to the suggested prior work, we note that [Yu 2020], along with [Cen 2021] mentioned by reviewer 7QJ7 and a number of other prior works [Xie 2020, Sayin 2022], is a value-based method at the core. [Yu 2020] considers a value iteration algorithm with confidence bounds. In [Cen 2021], the outer loop uses value iteration and only the inner loop runs an extragradient algorithm to solve a regularized bimatrix game. [Perolat 2015] is one of the first works to extend value-based methods from single-agent MDP to two-player Markov games. Since then, the basic techniques for analyzing value-based methods are known. On the other hand, policy optimization algorithms are much less understood in Markov games, but this is an important subject to study due to the wide use of such algorithms in practice. Our analysis treats the Markov game purely from the optimization perspective and is very different from the analysis in [Yu 2020, Cen 2021]. In single-agent MDPs, value-based methods and policy optimization methods have comparable convergence guarantees today, and our work aims to narrow the gap between the understanding of these two classes of algorithms in two-player Markov games.
>
> References
>
> Bai, Yu, and Chi Jin. "Provable self-play algorithms for competitive reinforcement learning." International conference on machine learning. PMLR, 2020.
>
> Cen, Shicong, Yuting Wei, and Yuejie Chi. "Fast policy extragradient methods for competitive games with entropy regularization." Advances in Neural Information Processing Systems 34 (2021): 27952-27964.
>
> Perolat, Julien, et al. "Approximate dynamic programming for two-player zero-sum Markov games." International Conference on Machine Learning. PMLR, 2015.
>
> Sayin, Muhammed O., Francesca Parise, and Asuman Ozdaglar. "Fictitious play in zero-sum stochastic games." SIAM Journal on Control and Optimization 60.4 (2022): 2095-2114.
>
> Xie, Qiaomin, et al. "Learning zero-sum simultaneous-move markov games using function approximation and correlated equilibrium." Conference on learning theory. PMLR, 2020.

---

> > ### Comment · Reviewer_KhY2 · 2022-08-09
> > **Response Follow Up**
> >
> > Thank the author's detailed response for $\nabla J$. However, most of my concerns remain unsolved.
> >
> > 1. I agree with the author that completely mixed Markov games are considered a hard problem to solve. Therefore I still recommend the authors include the "completely mixed Markov games" in the title, which is more rigorous and gives a better summarization of this work.
> >
> > 2. Nowadays, policy optimization algorithms are often applied with NN or features, but I don't know how to extend this paper's result to practical settings, e.g., Lemma 4 will no longer hold for the model with function approximations.
> >
> > 3. The author said in response c) that policy optimization algorithms are "an important subject to study due to the wide use of such algorithms in practice." Therefore, the authors should add more practical experiments to show their algorithm indeed works in practice. I believe this is important since there is no theoretical advantage of the author's algorithm compared with [Yu 2020].

---

> > > ### Author Response · Authors · 2022-08-09
> > > **Follow-Up Response to Reviewer KhY2**
> > >
> > > We thank the reviewer for the follow-up feedback.
> > >
> > > We very much agree with the reviewer that the investigating the convergence of the algorithm under NN is an interesting future direction. However, it is worth noting that the understanding of reinforcement learning algorithms for both MDP and Markov game needs to start from the tabular case which lays the foundation (for example, see [Bai 2019, Bhandari 2019, Cen 2021]). The extension of the result to the NN function approximation is certainly non-trivial, and so is the extension to linear function approximation and/or stochastic gradients. Exactly for this reason, we think it is more proper to leave NN as a future work and present the paper as it is, as it may be relevant to the researchers in the community seeking different extensions.
> > >
> > > With regards to the experiments, the sole purpose of the experiment in the paper is to suggest that the algorithm may work for non-completely mixed games and to motivate the future work in this direction. On the comparison to [Yu 2020], we note that [Yu 2020] derives a regret bound in the number of episodes (and a large number of parameter updates are carried out in each episode) and our result is a convergence rate in the number of parameter updates. Such differences make the direction comparison of the results inapplicable. In addition, the regret bound in [Yu 2020] does not imply a last-iterate guarantee, while our algorithm guarantee the convergence of the last iterate.
> > >
> > > Reference
> > >
> > > Bai, Yu, and Chi Jin. "Provable self-play algorithms for competitive reinforcement learning." International conference on machine learning. PMLR, 2020.
> > >
> > > Bhandari, Jalaj, and Daniel Russo. "Global optimality guarantees for policy gradient methods." arXiv preprint arXiv:1906.01786 (2019).
> > >
> > > Cen, Shicong, Yuting Wei, and Yuejie Chi. "Fast policy extragradient methods for competitive games with entropy regularization." Advances in Neural Information Processing Systems 34 (2021): 27952-27964.

---

> > > ### Author Response · Authors · 2022-08-09
> > > **Addressing comment on function approximation**
> > >
> > > We want to point out that using function approximation, such as neural networks (NNs), in the context of game theory is much more challenging than a single MDP problem. Indeed, in minimax problem considered in this paper, if we use NN, the objective function is in general nonconvex and nonconcave. In this case, the notion of a stationary point is not well defined in the literature. Even if we can adopt some definition of a stationary point in the existing work, it is also unclear how can we relate this point to the Nash equilibrium of the game. Moreover, the gradient descent ascent algorithm can also diverge in this general setting. On the other hand, in a single MDP problem, the existing literature always shows that gradient descent converges to a stationary point when using function approximation. Even in this simple setting, it is still challenging to characterize the quality of this stationary point. Thus, we do not think that an extension to the case of function approximation would be trivial in the context of game. Of course, one can use function approximation and the concept of our algorithm will remain the same. However, theoretical guarantees are largely open.
> > >
> > > It is also worth noting that even in the tabular setting, the theoretical understanding of gradient descent ascent is very sparse. Indeed, as mentioned our results significantly improve the existing results by using different techniques.

---

> > > ### Author Response · Authors · 2022-08-09
> > > **Comments on the relevant work in Yu 2020**
> > >
> > > We do not understand why the reviewer mentioned: "there is no theoretical advantage of the author's algorithm compared with [Yu 2020]"? The work in [Yu 2020] is the value-based method and provides regret analysis. On the other hand, our paper is about policy gradient descent ascent and study finite-time analysis. The nature of these two studies is very different, and both are important areas in machine learning literature. Directly comparing our work with [Yu 2020], in our opinion, is inappropriate, like comparing apple vs orange.

---

### Official Review · Reviewer_C5Kb · 2022-07-28

**Rating:** 6
**Confidence:** 4
**Soundness:** 2 fair
**Presentation:** 3 good
**Contribution:** 2 fair

**Summary:**

This work studies the so-called two-player zero-sum Markov game. They provides theoretical analysis of the convergence (to the Nash equilibrium) of practical gradient descent ascent (GDA) algorithms with some new numerical tools. In particular, they introduce an entropy regularization to the value function of the original game, and shows the linear convergence of GDA to the equilibrium point of the regularized game.

In addition, in order to derive practical algorithms to solve the original problem, they also studies a critical parameter, the weight parameter of the regularizer. Since there is a clear trade-off between improving convergence rate and minimizing the gap between the regularized game and the original one, they provide some practical schemes to adjust the regularization weight. The hope is that when the regularization weight is iteratively reduced to zero, the GDA algorithm will converge to the Nash equilibrium of the original game and hence the original game will be solved.

**Questions:**

The questions are listed above.

**Limitations:**

no.

**Strengths And Weaknesses:**

Strength:
1. The paper is well-written, self-contained and organized.
2. This work extends the current research on finding the Nash equilibrium of two-player zero-sum Markov game. They provide significant contributions to convergence analysis of practical gradient-based algorithms.
3. They also derive some practical algorithms to apply their regularization technique to solve the original game.

Weakness:
1. The description of the numerical experiments part is too simplified. I would like to see more details, for example, more information about the game environment, why the current setting of the synthetic game is chosen, more analysis on the results.  Although some of the details have been provided in appendix, but I think that is not enough, and some details should be put into the main paper.
2. The role of Assumption 2 played in the main results of this work is not clear. As the paper itself mentioned, some of the experimental results show that the Algorithm 2 still converges correctly when it is applied to a Markov game that does not observe Assumption 2.
3. In some of experimental results,  the pure GDA approach without regularization also has a last-iterate convergence and does not exhibit the oscillation behavior. Did the authors choose an appropriate game environment to valid their contributions?
4. The idea of introducing an entropy regularizer to improve the structure of the game is straightforward.

---

> ### Author Response · Authors · 2022-08-02
> **Response to Reviewer C5Kb**
>
> a) We thank for the reviewer for suggesting more discussion on the experiment design. The two games considered in Section 5 are minimal scale examples of completely mixed games and deterministic Nash games, respectively. To create a completely mixed game with $|\mathcal{A}|=|\mathcal{B}|=2$, we simply need to choose the reward function such that $r(s,\cdot,\cdot)$ as a 2x2 matrix is diagonal dominant or sub-diagonal dominant for any state $s$, and we can use an arbitrary transition probability kernel. Regarding why the last iterate of GDA without regularization also converges in Fig.2 (in the game with deterministic Nash equilibrium), we note that finding the Nash equilibrium in a deterministic game is often considered easier since the gradient of each player has the same sign regardless of the action of the other player, even if no regularization is used at all. Please see the second bullet in our response to all reviewers for more discussion. We will also consider adding experimental details to the main text in the next revision.
>
> b) Also in the second bullet in our response to all reviewers, we discuss how Assumption 2 comes up, why we believe it is an artifact of the analysis, and how we may remove the assumption in the future.

---

### Author Response · Authors · 2022-08-02
**Response to All Reviewers**

We greatly appreciate the feedback from the reviewers. We will carefully consider and incorporate the suggestions in the next revision of the paper.

The major questions for the paper are on the inaccessibility of the exact gradient $\nabla J$ and Assumption 2.

a) We agree with the reviewers that the exact gradient is difficult to compute in large systems where we do not have explicit knowledge of the transition probability kernel. However, it is important to note that our deterministic gradient algorithm lays the mathematical foundation and its sample-based extensions are technical but straightforward. By introducing a critic variable that tracks the Q function, we can show that the purely sample-based online (non-batch) actor-critic version of the regularized GDA converges under additional standard assumptions on the critic. Or if we assume having access to an unbiased, bounded-variance stochastic estimate of the true gradient returned by some sampling procedure like REINFORCE, our exact analysis can be used to guarantee convergence, with small changes to the choice of the step sizes. We will add discussion on these data-driven extensions in the next revision of the paper.

b) Assumption 2 is indeed a restrictive assumption that does not seem necessary but rather arises as an artifact of the current analysis. When we apply the weaker PL-type condition (Lemma 4) in the analysis, the entries of the iterates $\pi_{\theta_k},\phi_{\theta_k}$ need to be uniformly lower bounded, which is difficult to establish using the game structure. We come up with an innovative induction approach to quantify the connection between $\min_{s,a}\pi_{\theta_k}(a\mid s),\min_{s,b}\phi_{\psi_k}(b\mid s)$ and the optimality gaps $\delta_k^{\pi},\delta_k^{\phi}$. This approach allows us to transform the uniform lower bound requirement on $\pi_{\theta_k},\phi_{\psi_k}$ to that on the Nash equilibrium, leading to the uniformly completely mixed game assumption.

However, completely mixed Markov games are considered harder to solve in many places than games with deterministic Nash equilibrium, for which we do not have convergence guarantees yet. [Mertikopoulos 2018, Vlatakis-Gkaragkounis 2020] show that a classic algorithm fails only when the game is completely mixed.
In some sense, the difficulty of analyzing GDA for non-convex non-concave optimization is directly related to the mixed nature of the Nash equilibrium, which causes the gradient of one player to constantly change signs as the other player updates and results in the oscillation observed in first two plots of Fig.1. In games with only deterministic Nash equilibria, the gradient of one player does not change sign regardless of the policy of the other player, which makes convergence easier and explains why the last iterate of GDA without regularization converges in Fig.2.

We believe that we can remove Assumption 2 in the future, and the key is to use an enhanced version of Lemma 4. By refining the use of performance difference lemma and KL divergence, we may be able to establish Lemma 4 with $\min_{s,a}\pi_{\theta_k}(a\mid s),\min_{s,b}\phi_{\psi_k}(b\mid s)$ replaced by $\min_{s,a:\pi^{\star}(s,a)\neq 0}\pi_{\theta_k}(a\mid s),\min_{s,b:\phi^{\star}(s,b)\neq 0}\phi_{\psi_k}(b\mid s)$, which combined with careful mathematical treatment may remove Assumption 2. But this will likely require more sophisticated analysis beyond that in the current paper.
Finally, we note that the completely mixed game assumption is not uncommon in the literature when entropy-flavored regularization is used (see for example [Mertikopoulos 2016] and Theorem 6.1 of [Perolat 2021]).

References

Mertikopoulos, Panayotis, and William H. Sandholm. "Learning in games via reinforcement and regularization." Mathematics of Operations Research 41.4 (2016): 1297-1324.

Mertikopoulos, Panayotis, Christos Papadimitriou, and Georgios Piliouras. "Cycles in adversarial regularized learning." Proceedings of the Twenty-Ninth Annual ACM-SIAM Symposium on Discrete Algorithms. Society for Industrial and Applied Mathematics, 2018.

Perolat, Julien, et al. "From Poincaré recurrence to convergence in imperfect information games: Finding equilibrium via regularization." International Conference on Machine Learning. PMLR, 2021.

Vlatakis-Gkaragkounis, Emmanouil-Vasileios, et al. "No-regret learning and mixed nash equilibria: They do not mix." Advances in Neural Information Processing Systems 33 (2020): 1380-1391.

---

### Meta-Review · Area_Chair_6mca · 2022-09-05

**Recommendation:** Accept
**Confidence:** Certain

**Metareview:**

The paper studies the problem of finding the Nash equilibrium of a two-player zero-sum Markov game. Despite nonconvexity, this min-max optimization satisfies the PL condition and hence it is "easy" to solve. The authors rigorously studied the iteration complexity of this problem. The reviewers found the paper well organized and self-contained. The reviewers also found the contributions of the work solid. I recommend acceptance of the work. It would be great if the authors can address the reviewer's concerns and particularly the following concerns in the final version of the paper (most of which were answered in the discussion forum, but not in the paper):
- Please include discussions on various related works (such as[Cen 2021], [Perolat 2015], [Yu 2020] in the paper.) Please avoid having a "laundry list" type of citations, and do your best to discuss each of these works to the extent possible in the paper given one additional page (as done in the discussion forum). In addition, some of the references are closely related. For example, [Yang et al 2020]'s two-sided PL setting is closely related to the paper. Also, [Ostrovskii et al 2021] considers non-Euclidean geometries which is closely related to entropy regularization.
- Please include more details on the experiments (see the individual reviews)
- Please include the discussion on Assumption 2 (see the individual reviews)


**Award:**

No

---

### Decision · Program_Chairs · 2022-09-14

Accept